# Dynamics of stratospheric wave reflection over the North Pacific

Michael K. Schutte[1], Alice Portal[2], Simon H. Lee[3], and Gabriele Messori[1,4,5]

[1]Department of Earth Sciences, Uppsala University, Uppsala, Sweden
[2]Institute of Geography, Oeschger Centre for Climate Change Research, University of Bern, Bern, Switzerland
[3]School of Earth and Environmental Sciences, University of St Andrews, St Andrews, UK
[4]Swedish Centre for Impacts of Climate Extremes (climes), Uppsala University, Uppsala, Sweden
[5]Department of Meteorology and Bolin Centre for Climate Research, Stockholm University, Stockholm, Sweden

**Correspondence:** Michael Schutte (michael.schutte@geo.uu.se)

**Abstract.** Stratospheric wave reflection events involve the upward propagation of planetary waves, which are subsequently reflected downward by the stratospheric polar vortex. This phenomenon establishes a connection between the large-scale circulations in the troposphere and in the stratosphere. Here, we investigate a set of wave reflection events characterised by an enhanced difference between poleward eddy heat flux over the Northwest Pacific and equatorward eddy heat flux over Canada.
Previous research has pointed to a link between these events and anomalies in the tropospheric circulation over North America, with an associated abrupt continental-scale surface temperature decrease over the same region. In this study, we elucidate the dynamical mechanisms governing this chain of events.

We find that the evolution of meridional heat flux anomalies over the Northwest Pacific and Canada around reflection events is explained by a westward-propagating geopotential-height ridge and by the downstream development of a trough.
The trough advects colder-than-average air southwards in the lower troposphere over North America, leading to an abrupt temperature decrease close to the surface. The evolution of this large-scale pattern resembles the shift from a Pacific Trough to an Alaskan Ridge weather regime, with approximately one-third to one-half of such transitions associated with reflection events. Furthermore, stratospheric wave reflection events exert a far-reaching influence on the tropospheric circulation across the Northern mid- and high latitudes. For example, a few days after the reflection-driven temperature decrease across North
America the North Atlantic jet stream becomes unusually intense and zonal, favouring the occurrence of extreme winds over Europe.

# 1 Introduction

The variability of the stratospheric polar vortex plays a crucial role in shaping tropospheric circulation patterns and influencing weather extremes on subseasonal to seasonal timescales (Baldwin and Dunkerton, 2001; Thompson and Wallace, 2001; Domeisen and Butler, 2020; Kodera et al., 2013; Afargan-Gerstman et al., 2020). One notable example is that of sudden stratospheric warmings (SSWs), characterised by a rapid temperature increase in the stratosphere over the polar regions and a disruption of the stratospheric polar vortex (Scherhag, 1952; Matthewman et al., 2009; Charlton and Polvani, 2007). The occurrence of SSWs is favoured by the upward propagation of Rossby waves from the troposphere, which break in the stratospheric westerlies and weaken them (Matsuno, 1971; Sjoberg and Birner, 2012; Polvani and Waugh, 2004; Reichler and Jucker, 2022), although enhanced tropospheric wave activity is neither a necessary nor a sufficient condition for the occurrence of an SSW (de la Cámara et al., 2019; Birner and Albers, 2017; Albers and Birner, 2014). The study of SSWs has received particular attention due to their potential to modulate the tropospheric circulation (Charlton-Perez et al., 2018; Hitchcock and Simpson, 2014) and hence surface weather; for example, by favouring large-scale cold-air outbreaks over Eurasia (Cohen et al., 2007; Kretschmer et al., 2018; Kidston et al., 2015).

A related but distinct phenomenon, stratospheric wave reflection, has been linked to cold-air outbreaks over North America (Millin et al., 2022; Kodera et al., 2016; Kretschmer et al., 2018; Messori et al., 2022; Matthias and Kretschmer, 2020; Guan et al., 2020). Stratospheric wave reflection events are characterized by a local maximum of zonal winds in the mid-stratosphere (Perlwitz and Harnik, 2004; Messori et al., 2022) and imply the downward reflection of upward-propagating Rossby waves from the troposphere (Harnik and Lindzen, 2001; Perlwitz and Harnik, 2003; Shaw et al., 2010; Shaw and Perlwitz, 2013).

Stratospheric wave reflection events have been defined in many different ways. Some studies focus on daily wave activity fluxes (Kodera et al., 2008; Nath et al., 2014) or zonal-mean wave geometry diagnostics (Harnik and Lindzen, 2001; Shaw et al., 2010; Shaw and Perlwitz, 2013), which are insightful but may require a linear stationary wave model or data that are usually not a standard output of reanalyses or climate models (Matthias and Kretschmer, 2020). Another approach uses a reflection index based on the difference in zonal-mean zonal winds at 2 and 10 $\mathrm{hPa}$ (Perlwitz and Harnik, 2003; Harnik, 2009; Nath et al., 2016); however, this does not distinguish between SSWs and reflection events (Harnik, 2009). Alternatively, meridional eddy heat flux can be used to detect stratospheric wave reflection, since in a zonal mean framework it is proportional to the vertical component of the Eliassen-Palm (EP) flux or the vertical component of the wave activity flux (Takaya and Nakamura, 2001). Earlier studies have connected extremes of the zonal mean meridional eddy heat flux in the stratosphere to changes in stratospheric and tropospheric circulation (Dunn-Sigouin and Shaw, 2015; Shaw et al., 2014; Polvani and Waugh, 2004). However, due to the use of zonal averaging, these methods do not provide a spatial characterisation of wave reflection events.

To overcome this challenge, Matthias and Kretschmer (2020) proposed an index based on regional averages of the meridional eddy heat flux that is able to capture wave reflection events over the North Pacific. This new index, unlike more traditional reflection events based on zonal-mean diagnostics, builds on the climatological pattern of the meridional eddy heat flux that displays upward wave activity flux over the Bering strait and downward wave activity flux over North Canada. Previous research has highlighted that this definition of stratospheric wave reflection captures events leading to a transition from the

Pacific Trough (PT) to the Alaskan Ridge (AKR) North American weather regime and a statistically significant decrease in 2m air temperatures over North America (Messori et al., 2022; Matthias and Kretschmer, 2020; Cohen et al., 2022). However, the physical mechanisms linking such stratospheric wave reflection episodes to the tropospheric anomalies are not fully understood. In addition, reflection events and the subsequent decrease in surface temperatures over North America may exert remote effects

on the circulation and surface weather in the Euro-Atlantic sector, as suggested by previous work (e.g., Leeding et al., 2023a; Riboldi et al., 2023; Kodera et al., 2016).

In this work, we aim to provide a deeper insight into the evolution of the atmosphere during and after reflection events over the North Pacific, with a particular focus on weather regimes over North America. Specifically, we seek to answer the following questions:

1. How does the evolution of the meridional eddy heat flux relate to the tropospheric and stratospheric wave structure? Even though several studies have applied the definition of reflection events by Matthias and Kretschmer (2020), an analysis of the temporal evolution of meridional eddy heat flux at different vertical levels over East Siberia and North Canada is lacking. Exploring how meridional heat flux anomalies during stratospheric wave reflection events link to the wave structure can give additional insights on the vertical coupling between stratosphere and troposphere.

2. What is the mutual relation between stratospheric wave reflection events and the North American regime transition from Pacific Trough to Alaskan Ridge? Stratospheric downward reflection of Rossby waves has been shown to influence tropospheric dynamics, including weather regime transitions over North America (Messori et al., 2022; Kodera et al., 2008; Perlwitz and Harnik, 2003), which we would like to investigate further.

3. Do reflection events over the North Pacific affect the mid-latitude circulation beyond North America? Little is known on
these possible remote effects, and we explore this aspect here.

To address the above questions, we first analyse the evolution of reflection events with respect to meridional eddy heat flux and geopotential height anomalies. We further compare changes in geopotential height during reflection events to the shift from PT to AKR regimes. Next, we investigate changes in the mid-latitude Rossby wave activity through space-time spectral analysis. Finally, we study potential remote effects of reflection events on near-surface winds over Europe through the
modulation of the North Atlantic jet stream. Additional figures in Appendix A provide further context for and verification of our findings, but all core results are presented in the main body of the paper.

## 2  Methods

### 2.1  Data

The analysis is based on ERA5 reanalysis data (Hersbach et al., 2020), obtained at a spatial resolution of $1° \times 1°$ and 6 hourly temporal resolution, covering the extended winter season (DJFM) from 1 December 1979 to 31 March 2021. We analyse the daily mean of geopotential height, zonal and meridional wind, air temperature and the daily maximum of hourly 10-m wind speed. Daily space-time Rossby wave spectra are obtained from the meridional wind at $0.75° \times 0.75°$ and 6 hourly resolution. The wave activity flux (WAF) is calculated following Takaya and Nakamura (2001) using the NCL script provided in Nishii (2016). Unless stated differently, we deseasonalize all quantities shown in the following analysis by subtracting the daily seasonal cycle, computed with a 15-day centred running mean using data from 16 November to 15 April. Additionally, the temperature field is linearly detrended at each grid point. The meridional eddy heat flux, WAF and space-time Rossby wave spectra are computed from the full fields and deseasonalized afterwards.

### 2.2  Event definition

To identify reflection events, we follow the procedure outlined in Messori et al. (2022, 2024). Firstly, deviations from the zonal mean of meridional wind and temperature at 100 hPa are multiplied together to obtain the meridional eddy heat flux, $v'T'$. Next, we compute the regional cosine-latitude weighted averages of $v'T'$ over the Siberian domain (45-75°N, 140-200°E) and the Canadian domain (45-75°N, 230-280°E), shown in Fig. A1. We then apply a 15-day running mean, and compute anomalies with respect to the seasonal cycle. These anomalies are then divided by the standard deviation for each calendar day, which we denote by an asterisk. The reflection index is then given by:

$$RI = (v'T')^*_{Sib} - (v'T')^*_{Can} \tag{1}$$

We identify stratospheric reflection events when RI exceeds a value of 1 for a minimum of 10 consecutive days, yielding 45 events during the months of December to March (DJFM). This definition of reflection events differs from traditional approaches relying on zonal-mean metrics. The events can be aligned either at their onset or end date. Previous studies have investigated the sensitivity of this definition to parameter choice (e.g. Matthias and Kretschmer, 2020; Messori et al., 2022).

The relationship between wave reflection events and the large-scale circulation over North America is investigated using the weather regime data provided by Lee et al. (2023b). This dataset contains four weather regimes defined over North America (20-80°N, 180-330°E) through a $k$-means cluster analysis of 10-day low-pass filtered 500 hPa geopotential height fields. The four regimes are Pacific Trough (PT), Pacific Ridge (PR), Alaskan Ridge (AKR), and Greenland High (GH). Days where the 500 hPa geopotential height anomalies are closer to climatology than any of the four regimes are classified as No Regime (N). These are year-round regimes; although their occurrence frequency shows some seasonality, the reference spatial regime patterns are the same across all seasons. Further details of the regime classification can be found in Lee et al. (2023a). We then define North American weather regime 'events' as instances of more than 5 consecutive days with the same regime.

Applying this criterion, we identified 91 events for AKR and 107 events for PT, with the central date considered as the event date. We further define a PT-to-AKR transition when the PT regime is present on at least one of the 15 days preceding an AKR event. Using this definition, we identified 62 such transition events with day 0 set as the midpoint of the AKR event. The corresponding large-scale patterns are qualitatively independent of the choice of thresholds.

To verify whether the hemispheric-scale space-time Rossby wave decomposition reflects the dynamics of North Pacific reflection events, we define a regionalized signal of reflection events. We specifically select days where the cosine-latitude weighted spatial correlation of 100 hPa geopotential height anomalies over 45°N to 75°N and 140°E to 80°W (purple box in Fig. A1) with the composite anomalies during reflection events exceeds 0.95. We compute regionalized events separately for the onset and end dates of the reflection events using the Pearson correlation coefficient. Subsequently, we merge high-correlation days that are separated by less than 10 days to one regionalized event by selecting the day with the highest correlation coefficient at 100 hPa as the regionalized event date. This gives 63 regionalized events with geopotential height anomalies correlated to the onset of reflection events and 54 events correlated to the end of reflection events. With respect to a 69-day time window (the longest duration of a single reflection event), we found 39 of 54 instances with regionalized onset events being followed by regionalized end events. Computing the correlation on several vertical levels simultaneously to identify the high-correlation days results in qualitatively similar results (not shown).

## 2.3 Space-time Rossby wave spectra

In contrast to analyses of stratosphere-troposphere coupling focusing on circulation indices (Baldwin and Dunkerton, 2001; Thompson et al., 2006; Hall et al., 2021; Kolstad et al., 2022) or weather regimes (Charlton-Perez et al., 2018; Domeisen et al., 2020; Hall et al., 2023; Messori et al., 2022; Lee et al., 2022), space-time spectral analysis allows to study time-varying wave structures with different horizontal scales and phase speeds. Previous studies have already employed such a methodology to compare Rossby wave properties across data sets (Dell'Aquila et al., 2005) and time periods (Riboldi et al., 2020; Sussman et al., 2020). Furthermore, the method has proved useful to analyse circumpolar Rossby wave patterns during boreal winter (Riboldi et al., 2022) and the connection between the stratosphere and troposphere during strong and weak stratospheric polar vortex events (Schutte et al., 2024).

Space-time wave spectra decompose a time-varying wave structure (as illustrated, for example, in an Hovmöller diagram) into distinct harmonics with varying horizontal scales and phase speeds. We obtain the space-time spectra of Rossby waves following the procedure described in Riboldi et al. (2022); Schutte et al. (2024). Rossby waves, represented by meridional wind between 35°N and 75°N, are decomposed in time and in space along each latitude circle using a double Fourier transform:

$$V(\lambda, t; \phi) = \sum_{j=-N_T/2}^{N_T/2} \sum_{n=-N_L/2}^{N_L/2} \hat{V}(n, \omega_j; \phi) e^{i(n\lambda - \omega_j t)}, \tag{2}$$

where $\hat{V}(n, \omega_j; \phi)$ denotes the complex Fourier coefficient for a given wavenumber $n$, angular frequency $\omega_j$ and latitude $\phi$ of the meridional wind $V(\lambda, t; \phi)$ at longitude $\lambda$, time $t$ and latitude $\phi$. The number of grid points along a given latitude

circle is $N_L = 480$, and $\omega_j = 2\pi j/N_T$ is the angular frequency, with $N_T = 244$ six-hourly time steps over 61 days. For the decomposition in time, each day is treated as the midpoint of a sliding 61-day time frame, centred at 12 UTC. To minimize boundary effects, a double cosine tapering method reduces the weight of the first and last 12 days of each window. This yields periodograms $S_{\overline{V'V'}}(n, \omega_j; \phi) = 2\, Re(\hat{V}\hat{V}^*)$ as a function of frequency and wavenumber, where $\hat{V}^*$ is the complex conjugate of the corresponding coefficient $\hat{V}$ and the overline denotes the absolute value of the spectrum. Each periodogram undergoes ten iterations of smoothing in the frequency direction, employing a three-point window, as in Wheeler and Kiladis (1999). Subsequently, these periodograms are remapped to the phase speed domain along each latitude circle by interpolating the periodogram along lines of constant phase speed $c_p = \omega/k$, followed by re-scaling (Randel and Held, 1991). After averaging in the latitude band between 35°N and 75°N, one obtains the spectra $S_{\overline{V'V'}}(n, c_p)$. As for the other variables, the spectra are deseasonalized by subtracting the daily seasonal cycle, computed with a 15-day centred running mean using data from 16 November to 15 March. Further details can be found in the Supplement to Riboldi et al. (2022). Despite the extended time window used for the computation of Rossby wave space-time spectra, this metric can capture daily to weekly changes in Rossby wave behaviour (cf. Fig. 5 with A2, and see discussion in the end of section 3.1.3).

Based on the space-time spectra, two integral metrics can be defined to summarize the overall magnitude of Rossby wave activity and the direction of wave propagation. Firstly, the integrated spectral power (ISP) measures the overall magnitude of spectral density of Rossby waves across all wavenumbers $n$ and phase speeds $c_p$:

$$\mathrm{ISP} = \sum_{n=1}^{15} \sum_{c_p=-30}^{30} S(n, c_p) \tag{3}$$

The ISP of westward-(eastward-)propagating Rossby waves, which we denote as $\mathrm{ISP}_{west}$ ($\mathrm{ISP}_{east}$) is obtained by integrating the spectral density between phase speeds from -30 $ms^{-1}$ to 0 $ms^{-1}$ (0 $ms^{-1}$ and 30 $ms^{-1}$).

Secondly, the Rossby wave phase speed $\overline{c_p}$, defined in Riboldi et al. (2020), serves as a hemispherically-averaged estimate across the resolved harmonics and represents a weighted mean:

$$\overline{c_p} = \frac{\sum_{n=1}^{15} \sum_{c_p=-30}^{30} S_{\overline{V'V'}}(n, c_p) \cdot c_p}{\sum_{n=1}^{15} \sum_{c_p=-30}^{30} S_{\overline{V'V'}}(n, c_p)} = \frac{1}{\mathrm{ISP}} \sum_{n=1}^{15} \sum_{c_p=-30}^{30} S_{\overline{V'V'}}(n, c_p) \cdot c_p \tag{4}$$

The phase speed associated with each $(n, c_p)$ harmonic is weighted by the corresponding spectral energy density $S_{\overline{V'V'}}(n, c_p)$, ensuring that the most energetic harmonics contribute most to the $\overline{c_p}$. Both integral metrics, $ISP$ and $\overline{c_p}$, are computed from the full space-time spectra and deseasonalized afterwards.

## 2.4 Statistical significance assessment

The statistical significance of composite anomalies, such as those indicated by hatching in Fig. 1 is assessed using 10 000 bootstrapping iterations (i.e., random sampling with replacement). Following the approach by Wilks (2016), we account for multiple testing. Indeed, since we conduct a separate statistical test at each gridpoint in our fields, and the number of gridpoints

is large (e.g., $N_{tests} = N_{lat} \cdot N_{lon} \approx 20\,000$), the risk of erroneously rejecting the null hypothesis by chance is high. Following equation (3) in Wilks (2016), we look for the largest percentile $p^*_{FDR}$ that satisfies the condition:

170
$$p^*_{FDR} = max_{i=1...N}[p_i \; : \; p_i \leq (\frac{i}{N_{tests}}) \cdot \alpha_{FDR}], \qquad (5)$$

where $p_i$ represents the i-th entry in the sorted list of percentiles. Given the spatial and temporal correlation between data points, the chosen control level $\alpha_{FDR}$ is set to 0.10 (Wilks, 2016). Consequently, $p^*_{FDR}$, which in our analyses yields mostly values below 0.05, is selected as the new significance level.

# 3   Results

## 3.1   Evolution of stratospheric reflection events

### 3.1.1   Temporal and spatial patterns in meridional eddy heat flux

Following the definition of reflection events from Sect. 2.2, the difference between $v'T'$ anomalies in the Siberian domain and the Canadian domain is maximised during reflection events. This corresponds to positive $v'T'$ anomalies and enhanced upward-propagating Rossby waves over Siberia and negative $v'T'$ anomalies and enhanced downward-propagating Rossby waves over Canada.

The positive anomalies over Siberia during reflection events are preceded by negative anomalies with a distinct vertical structure. Both the negative and positive $v'T'$ anomalies reach the upper troposphere, although the positive anomalies are strongest in the stratosphere (Fig. 1 a). The lack of statistically significant anomalies below 300 hPa could arise from greater tropospheric variability during reflection events. Furthermore, when centred around the reflection onset date, statistically significant positive anomalies in $v'T'$ start four days earlier in the middle stratosphere (10 hPa) compared to the upper troposphere (200 hPa) (Fig. 1 a). This is consistent with a stratospheric onset of reflection events, connected to a local maximum of zonal wind speeds at 10 hPa, which favours wave reflection (cf. Fig. 2 in Messori et al. (2022)). Negative anomalies of Siberian $v'T'$ emerge following the events when centring at the end date. (Fig. 1 b).

The alternation of different signed $v'T'$ anomalies around reflection events is also evident in the Canadian domain (Fig. 1 c and d). Here, a more pronounced (albeit statistically non-significant) coupling to lower tropospheric levels is discernible. When centred around the end of reflection events, positive $v'T'$ anomalies emerge close to the surface 8 days prior, later extending to higher levels (Fig. 1 d). The emergence of low-level positive $v'T'$ anomalies is connected to the southward advection of colder-than-average air near the surface, while the negative upper-level anomalies weaken from higher stratospheric levels downward, consistent with decreasing wave reflection. This could in turn correspond to increased Rossby wave breaking or absorption in the middle stratosphere.

To better understand this vertical structure, we consider the spatio-temporal evolution of $v'T'$ around the end of the reflection events, designated as day 0. During the peak of reflection events, the 100 hPa anomalies remain mostly confined within the Siberian and Canadian domains. However, the negative 100-hPa $v'T'$ anomaly above Canada starts to propagate westward around day 0 (Fig. 2 a) and is replaced by a positive anomaly at +5 to +12 days. Additional positive anomalies of meridional eddy heat flux during reflection events over Asia and Europe suggest a connection between reflection events and hemispheric-scale circulation patterns. The anomalies at 250 hPa mirror the behaviour of the lower stratosphere, albeit with weaker anomalies (Fig. 2 b).

The picture at 850 hPa is very different compared to the upper levels, notably in the positive anomalies that emerge in the Canadian domain at negative lags of a few days (Fig. 2 c). These correspond to southward advection of colder-than-average air, associated with negative 2m air temperature anomalies (Fig. 2 c). This stands in stark contrast with the positive temperature anomalies around the onset of the reflection events, associated with positive $v'T'$ anomalies over the Northeast Pacific advecting

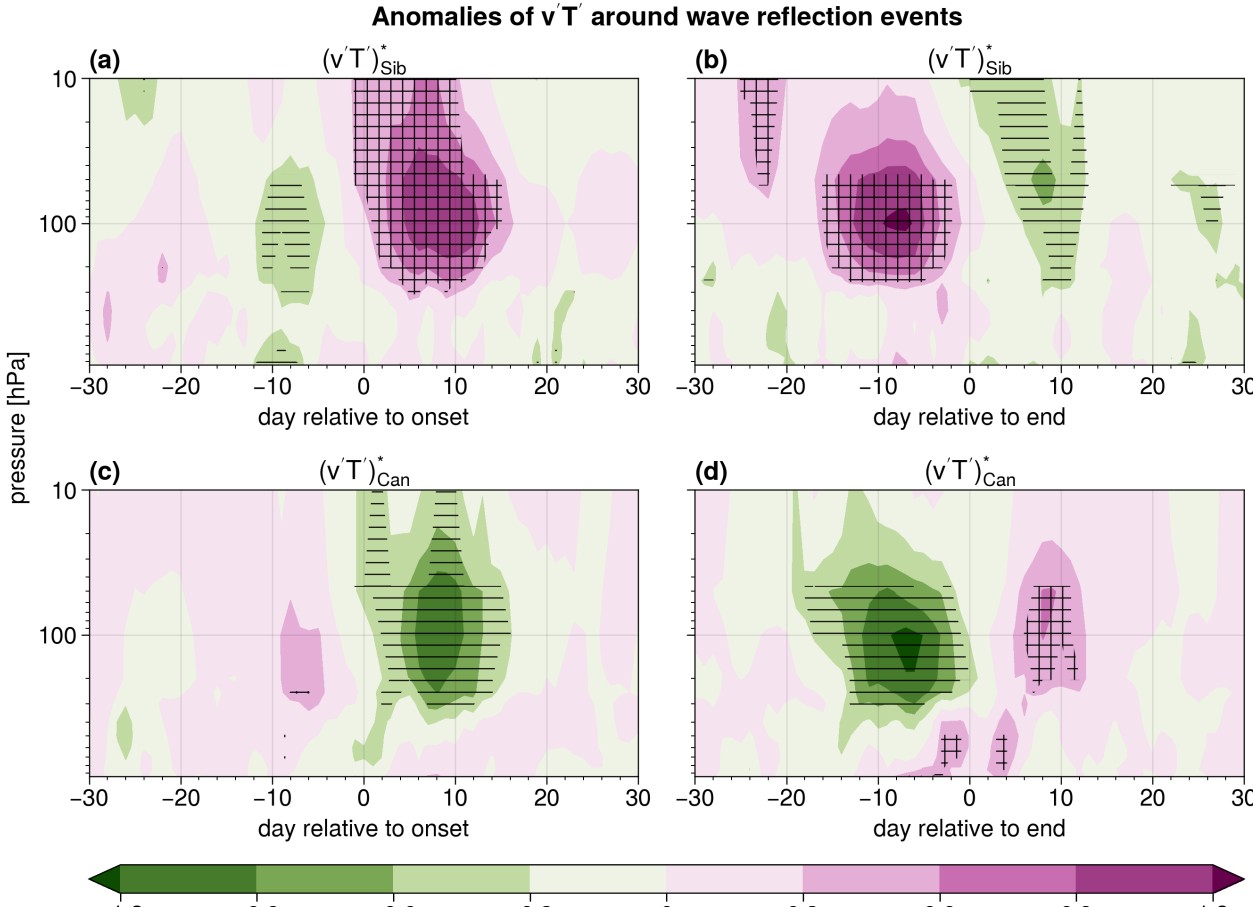

**Figure 1.** Vertical cross-section of standardized anomalies of composite meridional eddy heat flux $v'T'$ averaged over the Siberian domain (45-75°N, 140-200°E) around (a) onset date and (b) end date of reflection events. In panels (c) and (d) the same is shown for the Canadian domain (45-75°N, 230-280°E). Horizontal hatching marks statistically significant negative anomalies and cross-hatching statistically significant positive anomalies. Anomalies of v'T' are obtained by subtracting the daily seasonal cycle, computed with a 15-day centred running mean (section 2.1).

warm air northward. Investigating the evolution of air temperature anomalies confirms the difference between the upper levels and 850 hPa (Fig. A3).

In summary, the majority of statistically significant $v'T'$ anomalies are observed in the stratosphere and upper troposphere during reflection events. The oscillation of $v'T'$ anomalies in the stratosphere over the Siberian and Canadian domain originates from the westward propagation of negative $v'T'$ anomalies and the subsequent formation of new positive $v'T'$ anomalies downstream, and is mostly linked to changes in the meridional wind (Fig. A4). Furthermore, the contrasting $v'T'$ anomalies

210

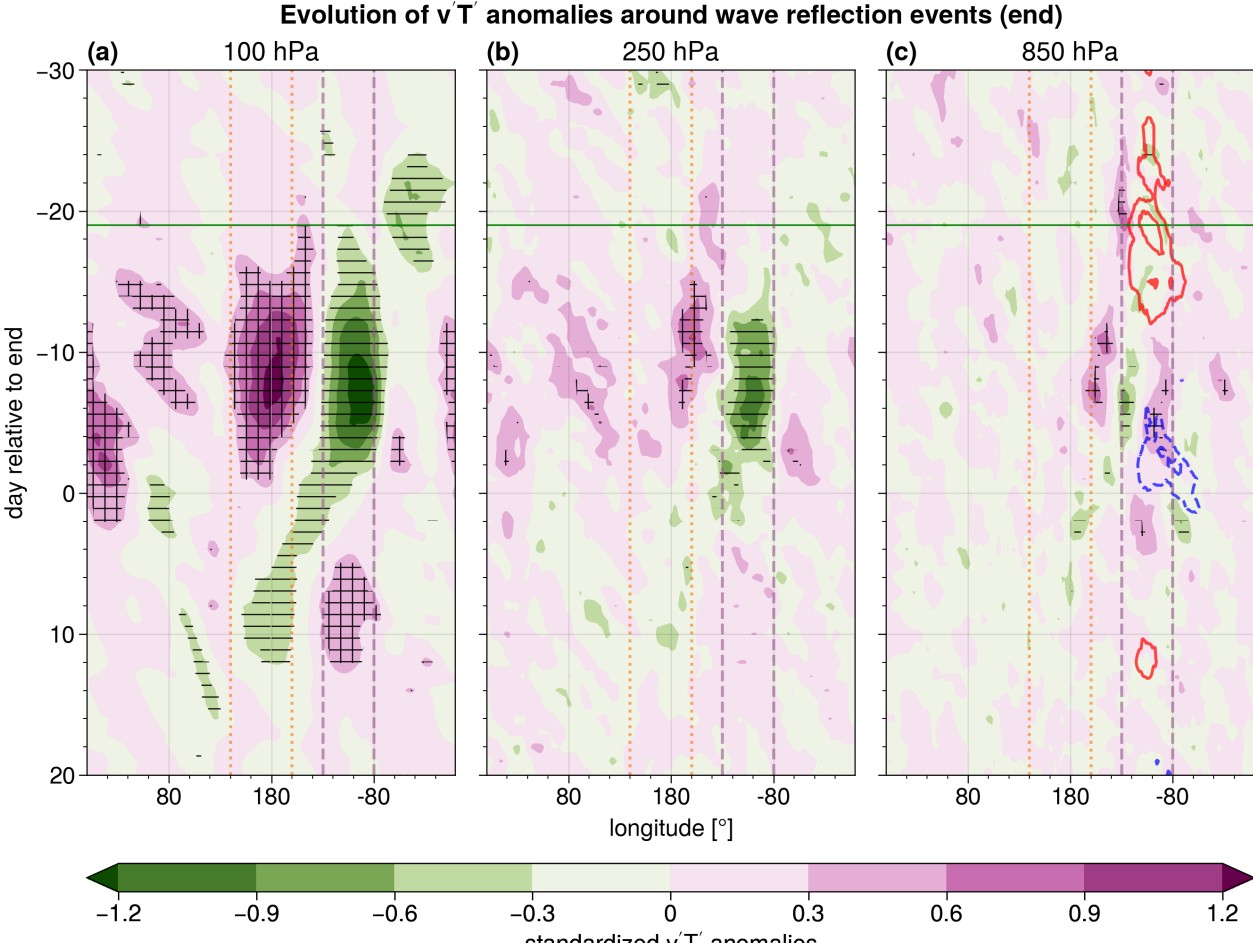

**Figure 2.** Hovmöller diagram of standardized anomalies of composite meridional eddy heat flux $v'T'$ averaged between $45°N - 75°N$ centred around the end date of reflection events at (a) 100 hPa, (b) 250 hPa and (c) 850 hPa. Horizontal hatching marks statistically significant negative anomalies and cross-hatching statistically significant positive anomalies. Continuous red and dashed blue contours in (c) show standardized positive and negative 2m air temperature anomalies, respectively, averaged between $40°N - 55°N$ ($1\ std \approx 3.8\ K$). The contours start at $\pm 0.5\ std$ with steps of $0.3\ std$. The continuous horizontal green line shows the median onset day of reflection events. The vertical lines mark longitudes of the Siberian (orange, dotted) and Canadian (purple, dashed) domains.

before the onset and after the end of reflection events relate to times with spatially more uniform, weakly positive meridional eddy heat fluxes over both regions, while the more-pronounced anomalies during reflection events can be interpreted as bursts in $v'T'$ (Fig. A5). In the Canadian domain, opposite anomalies between the upper levels and the lower troposphere appear a few days before the end of the reflection events. The positive low-level $v'T'$ anomalies are consistent with equatorward cold-air advection and the observed large-scale decrease in 2m air temperatures.

### 3.1.2 Evolution of geopotential height anomalies

After highlighting westward-propagating $v'T'$ anomalies around the end of reflection events, we now connect this behaviour to the evolution of geopotential height anomalies and vertical WAF.

A westward-propagating ridge stands out as a characteristic feature of reflection events. The positive geopotential height anomaly associated with the ridge forms during the onset at 100 hPa over the Canadian domain (Fig. 3 a; green horizontal line). This anomaly intensifies and exhibits westward propagation as the events progress, reaching Europe by day +6. Shortly after the formation of the ridge, a trough forms over Eurasia and persists throughout the duration of the reflection events. After the end of the event, a new trough develops over North America, downstream of the westward propagating ridge. West of the ridge, upward WAF anomalies match the positive $v'T'$ anomalies observed over the Siberian domain in the stratosphere (cf. Figs. 2 a and 3 a), while enhanced downward WAF east of the ridge is consistent with the negative anomalies of $v'T'$ over North America. The behaviour of geopotential height and WAF at 250 hPa closely resembles that observed at 100 hPa, albeit with slightly weaker anomalies (Fig. 3 b). While the statistically significant strengthening of the ridge at 250 hPa occurs a few days later than at 100 hPa, the negative geopotential height anomaly downstream of the ridge precedes that at 100 hPa by a few days (cf. Fig. 3 a and b).

As in the case of the $v'T'$ anomalies, the pattern of WAF anomalies is different at lower levels. Upward 850 hPa WAF anomalies over North America around the onset of reflection events are replaced, around day -8, by enhanced downward anomalies (Fig. 3 c). At the same time, the positive geopotential height anomaly propagates to the west and replaces the negative anomaly over the North Pacific, similarly to the upper levels. The downstream development of the negative 850 hPa geopotential height anomaly occurs at around -10 days, earlier than at upper levels.

The westward propagation of the ridge around the end of reflection events corresponds to increasing ISP for Rossby waves with westward phase speed (Fig. 3 d). In particular, suppressed values of $\mathrm{ISP}_{west}$ observed around the median onset day are followed by a roughly monotonic increase, peaking a few days before their end date. The spectral properties of Rossby waves are further analysed in Section 3.1.3.

The evolution of the spatial pattern of geopotential height anomalies highlights the hemispheric scale of the reflection events (Fig. 4). In addition to the westward-propagating ridge and subsequent development of a trough downstream (Fig. 3), we observe clear geopotential height anomalies over Eurasia (first three columns in Fig. 4). Specifically, a Rossby wave train pattern extending from the North Atlantic to East Siberia suggests a hemispheric-scale circulation pattern around the end of reflection events (Fig. 4 c and h). This pattern, as well as the shift of geopotential height anomalies over North America and the Pacific, are even more pronounced when comparing the onset and end of reflection events (Fig. 4 d and i). Furthermore, the vertical coherence in the evolution of geopotential height anomalies highlights the equivalent barotropic character of reflection events (Fig. A6).

The temperature anomalies associated with the ridge formation over the North Pacific align with the previously discussed $v'T'$ anomalies. Warm air is advected northward at higher levels over the North Pacific around the ridge and redirected southward over North America (Fig. 4 e). The temperature anomalies at lower levels exhibit a more distinct pattern than at 100

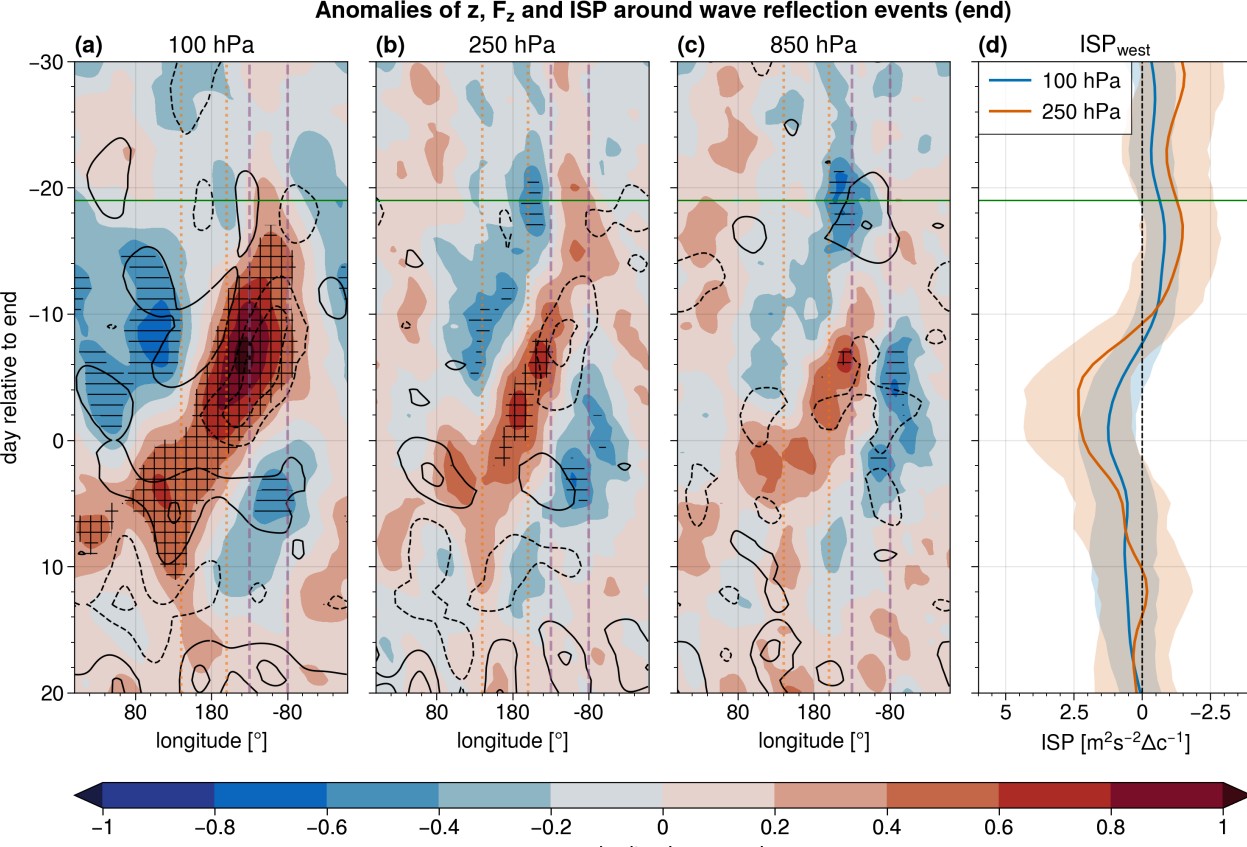

**Figure 3.** Hovmöller diagram of standardized anomalies of geopotential height averaged between $45°\text{N} - 75°\text{N}$ centred around the end date of reflection events at (a) 100 hPa, (b) 250 hPa and (c) 850 hPa in shading. Horizontal hatching marks statistically significant negative anomalies and cross-hatching statistically significant positive anomalies. Black lines indicate standardized anomalies of the vertical component of WAF (Takaya and Nakamura, 2001) filtered for wavenumbers-1 to 4, in steps of 0.1, excluding the zero-line. (d) Time series of ISP for westward-propagating Rossby waves and the 95% confidence interval (shaded area) at 100 hPa and 250 hPa with the x-axis being flipped, so that enhanced activity of westward-propagating Rossby waves lies to the left of the zero line. The continuous horizontal green line shows the median onset time of reflection events. The vertical lines mark longitudes of the Siberian (orange, dotted) and Canadian (purple, dashed) domains.

hPa, with warmer-than-average air predominantly directed poleward over Alaska around day -4 and colder-than-average air advected over Canada (Fig. 4 j). The resulting vertical sign-change of temperature anomalies over Canada links to the opposite sign of $v'T'$ anomalies between stratosphere and lower troposphere, as discussed in the previous section. Since geopotential
255  height anomalies are proportional to the vertical integral of the underlying temperature anomalies, we expect a local minimum in geopotential height in correspondence of the change in sign from low-level negative to upper-level positive temperature anomalies (Fig. A7 a; see also Harnik and Lindzen (2001); Randel and Stanford (1985)).

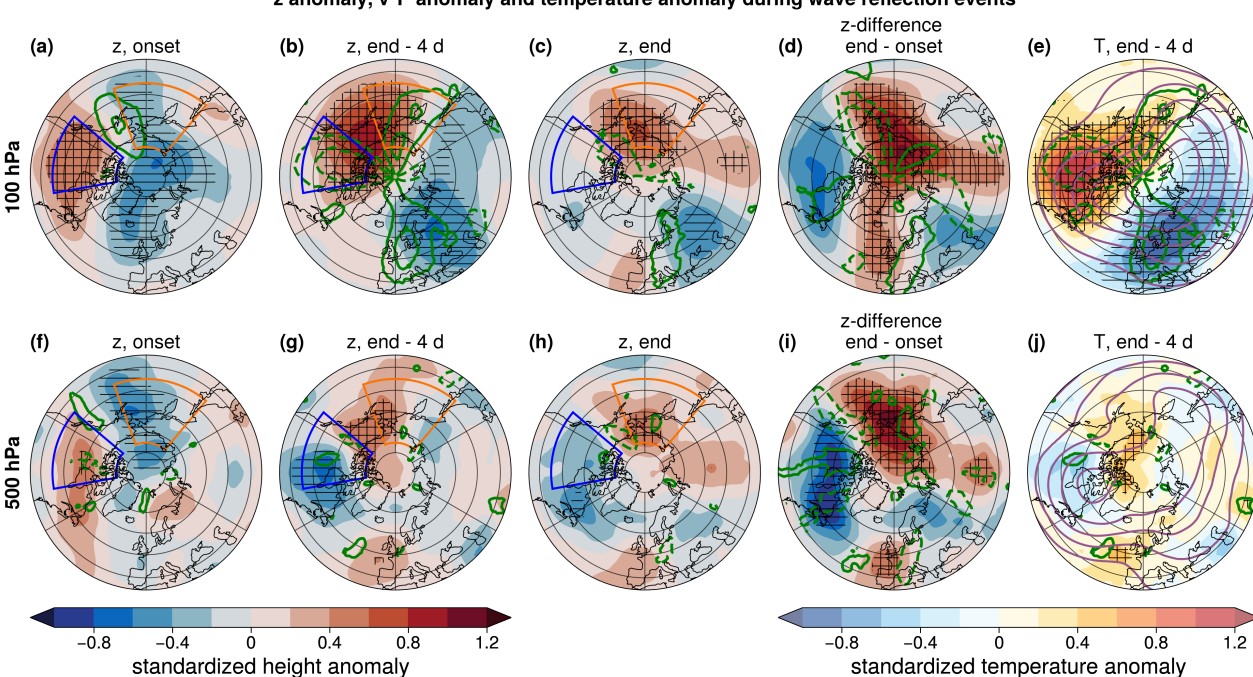

**Figure 4.** Geopotential height anomalies in shading and $v'T'$ anomalies in green contours (a, f) during onset, (b, g) 4 days before end, (c, h) at the end date and (d, i) difference between onset and end of reflection events at 100 hPa (first row) and 500 hPa (second row). Horizontal hatching marks statistically significant negative geopotential height anomalies and cross-hatching statistically significant positive anomalies. (e, j) Temperature anomalies in shading, geopotential height field in purple and $v'T'$ anomalies in green 4 days before end.

In summary, the onset of reflection events is characterized by a vertically coherent positive geopotential height anomaly at upper levels over North Canada. This propagates westward, reaching Siberia by the end of the events. At the same time, a negative geopotential height anomaly emerges downstream over eastern Canada, first at lower levels and later extending into the stratosphere. At upper levels enhanced upward WAF is observed over the Siberian domain west of the ridge, and enhanced downward WAF over Canada east of the ridge. Positive WAF anomalies close to the surface over North America emerge 10 days before the end of reflection events.

260

### 3.1.3 Rossby wave characteristics during reflection events

We next use the space-time spectral analysis described in Section 2.3 to investigate the Rossby wave behaviour at the onset and end of reflection events.

Rossby wave spectra indicate an enhancement of westward-propagating Rossby waves from the onset to the end of reflection. During the onset at 100 hPa, the strong activity for eastward-propagating waves 2 to 5 is likely related to the presence of an accelerated stratospheric flow, under the assumption that this affects all wavenumbers equally by shifting the whole spectrum towards higher phase speeds than climatology (Fig. 5 a). This is followed by enhanced activity for westward-propagating Rossby waves during the end of the events (Fig. 5 b), which agrees with the presence of westward-propagating anomalies. The deceleration of Rossby wave propagation during the evolution of the reflection events is characterized by a statistically significant enhancement of westward-propagating wave 1 , 3 and 4 and by a decreased activity for eastward-propagating waves 2 to 5 (Fig. 5 c). Although the decrease in activity of eastward-propagating waves 2 to 5 is not statistically significant, it has a magnitude comparable to the statistically significant enhancement of westward-propagating waves. The anomalies for the high-frequency waves lie in a phase-space region with low spectral power, hence they contribute less to the overall Rossby wave energy. At the end of reflection events, their enhanced westward activity is related to the presence of retrograding planetary-scale Rossby waves, which form the background where the high-frequency waves propagate. Westward-propagating Rossby wave activity increases also at higher stratospheric levels (Fig. A8) and in the upper troposphere, albeit the tropospheric change in westward-propagating wave-1 is weaker (Fig. 5 d to f).

The phase speed of Rossby waves in the vertical column between 850 hPa and 10 hPa thus decreases during the course of reflection events (Fig. 6 a). Higher phase speed $\overline{c_p}$ in the lower stratosphere and troposphere around the onset of reflection events is consistent with the faster zonal flow and with the enhanced activity of eastward-propagating Rossby waves. Around the end, the phase speed drops below average across the entire vertical column. Additionally, the phase speed stays below average after reflection events have ended, potentially indicating a weakening of the stratospheric circulation after reflection events. However, as noted by Messori et al. (2022), this weakening is not sufficient to cause a statistically significant drop in zonal-mean zonal winds, but could rather be connected to a stretched stratospheric polar vortex displaced towards Eurasia.

Anomalies in ISP confirm the change of anomalous Rossby wave behaviour during reflection events, but highlight differences in wave activity between stratosphere and troposphere. The overall wave activity is unusually high in the stratosphere before the onset of reflection events and increases in the troposphere during reflection events (Fig. 6 b). In the stratosphere, the overall wave activity increases again around the end of reflection events, which could indicate that waves propagate more into the upper stratosphere than during reflection events. Positive anomalies of $ISP_{east}$ align with periods of enhanced phase speed around the onset of reflection events (cf. Fig. 6 a and d). However, when focusing on $ISP_{west}$, an increase in Rossby wave activity is visible in the entire vertical column approximately one week prior to the end of reflection events (Fig. 6 c). Simultaneously, $ISP_{east}$ is suppressed in the lower troposphere (Fig. 6 d), while Rossby waves are more active in the stratosphere. The enhanced westward propagation of Rossby waves beginning in the lower troposphere coincides temporally with the previously-mentioned strong decrease of 2m air temperatures over North America and influences the upper levels only at a later stage. The integrated spectral

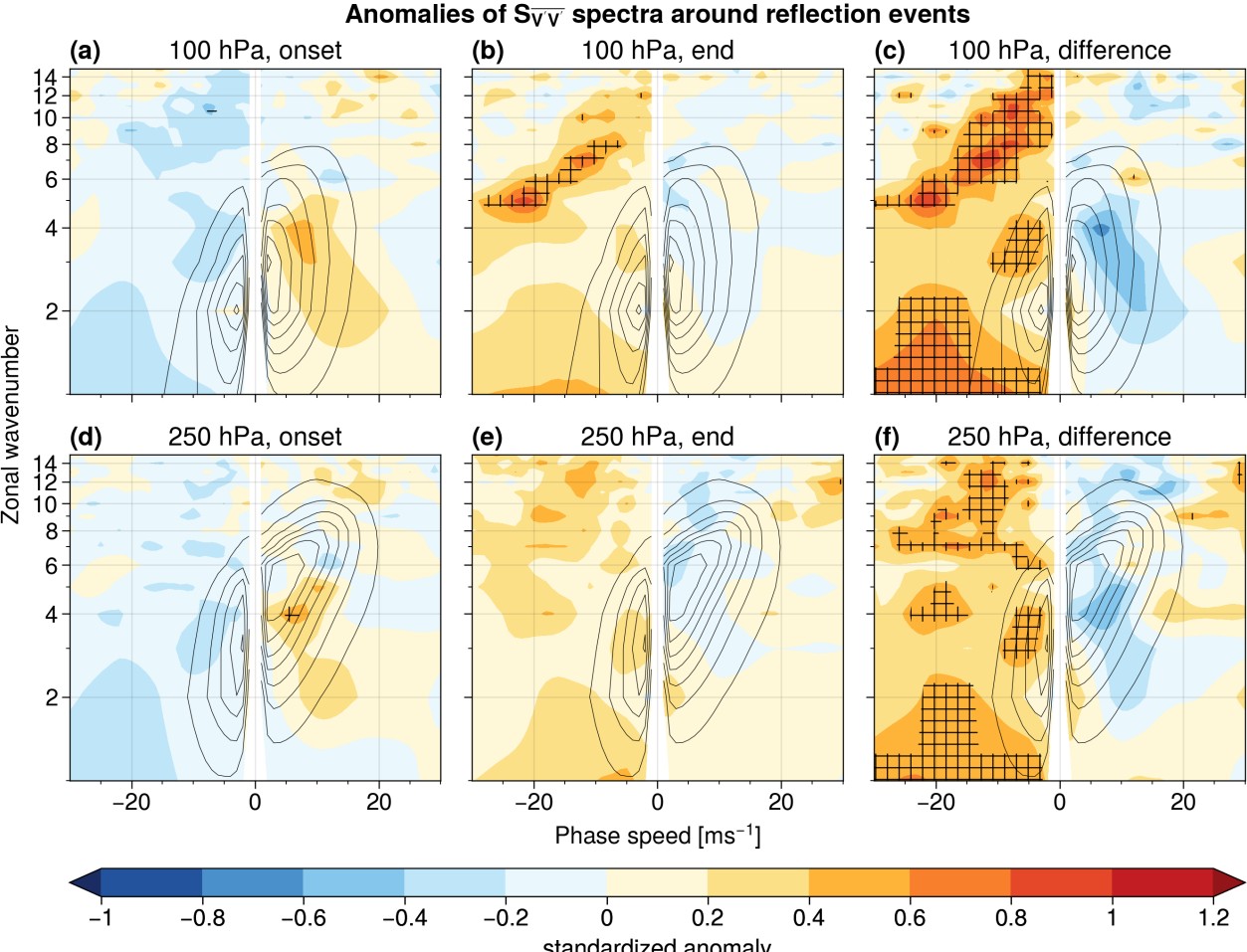

**Figure 5.** Standardized anomalies of spectral power $S_{\overline{V'V'}}$ of each harmonic at 100 hPa during (a) reflection event onset, (b) end and (c) difference between onset and end are represented in shading. Sub-panels (d) - (e) show the same, but for 250 hPa. Black contours show the DJFM mean for all years (100 hPa: steps of $0.05\,\mathrm{m^2\,s^{-2}\,\Delta c^{-1}}$ from $0.05\,\mathrm{m^2\,s^{-2}\,\Delta c^{-1}}$ to $0.35\,\mathrm{m^2\,s^{-2}\,\Delta c^{-1}}$; 250 hPa: steps of $0.1\,\mathrm{m^2\,s^{-2}\,\Delta c^{-1}}$ from $0.1\,\mathrm{m^2\,s^{-2}\,\Delta c^{-1}}$ to $0.7\,\mathrm{m^2\,s^{-2}\,\Delta c^{-1}}$). Horizontal hatching indicates statistically significant negative anomalies, cross-hatching indicates statistically significant positive anomalies.

metrics thus complement the space-time spectra and highlight relevant temporal scales and vertical variations of Rossby wave activity during reflection events.

300     To summarize, space-time spectral analysis highlights the enhancement of westward-propagating Rossby waves towards the end of reflection events. This is particularly pronounced in westward-propagating waves 1, 3 and 4, and coincides with a decrease in phase speed. The vertical structure of geopotential anomalies confirms the importance of waves 2 to 4 in the troposphere and wave 1 in the stratosphere (Fig. A7). The westward propagation of large-scale anomalies in the lower stratosphere

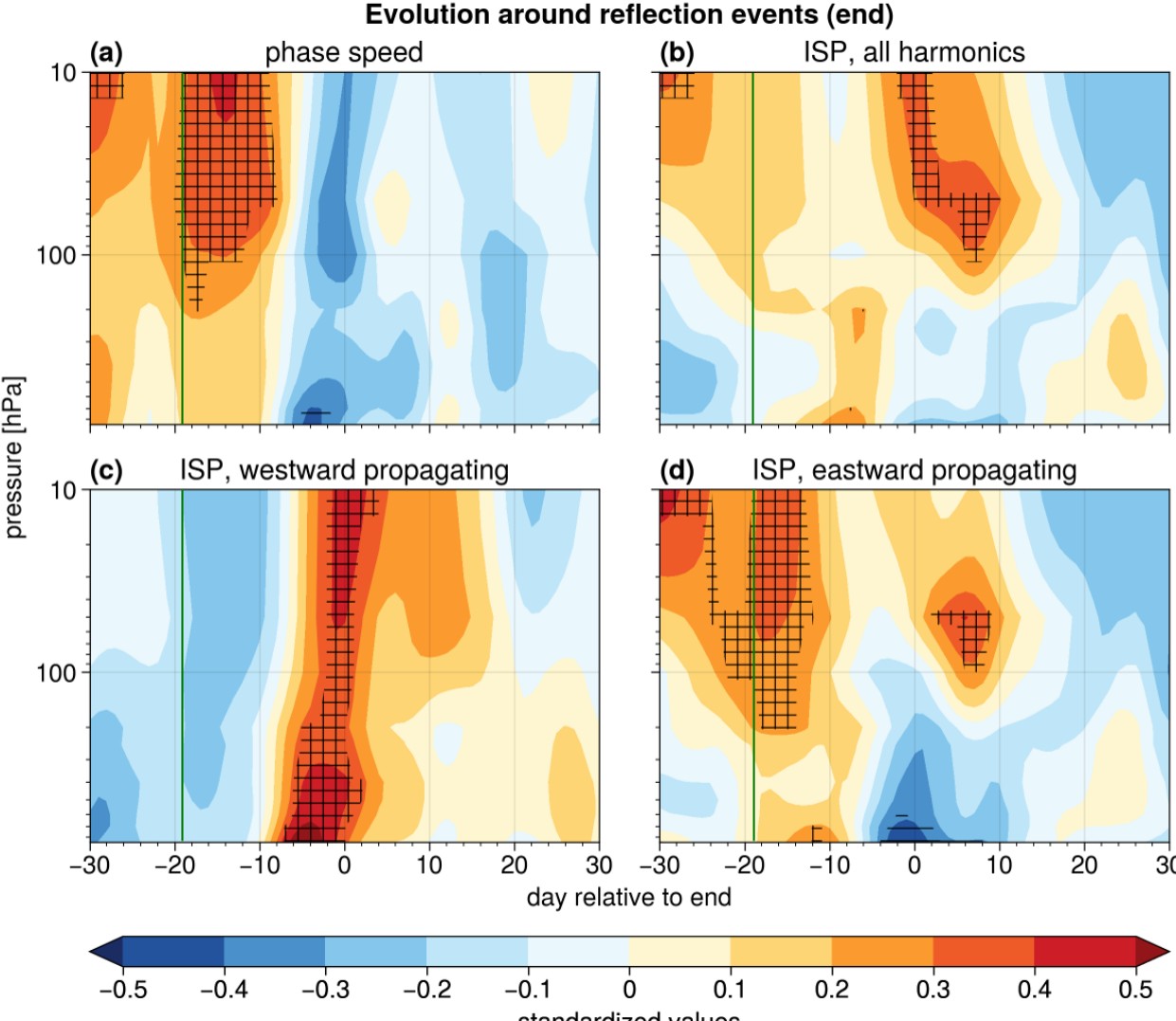

**Figure 6.** Vertical cross-section of standardized anomalies of (a) phase speed, (b) ISP, (c) ISP$_{west}$ and (d) ISP$_{east}$ derived from spectral power S$_{\overline{V'V'}}$ between 850 hPa and 10 hPa centred around the end date of reflection events. Horizontal hatching marks statistically significant negative anomalies, cross-hatching statistically significant positive anomalies. The continuous vertical green line shows the median onset time of reflection events.

and upper troposphere coincides with a shift of Rossby wave phase lines, that typically tilt westward with height, to a more vertical orientation over the North Pacific during stratospheric wave reflection (Fig. 12 in Messori et al., 2022). This change in vertical phase tilt is consistent with a more pronounced enhancement of eastward phase speeds at higher stratospheric levels compared to those below (Fig. 6 a). Additionally, westward-propagating Rossby waves become more active in the troposphere

and stratosphere around 1 week before the end of reflection events. This matches temporally with the strengthening of the blocking ridge over Alaska, which will be discussed in the next section.

A possible limitation of this analysis is that the spectra are computed over a full hemispheric domain and may present difficulties in resolving a regional signal. To connect the spectra to reflection events, we build a set of "regionalized events" based on the correlation of the 100 hPa geopotential height field over the North Pacific with reflection events (section 2.2). These events show a spectral shift to enhanced westward-propagating Rossby waves similar to reflection events, demonstrating the effectiveness of the spectral analysis in quantifying changes in Rossby waves during North Pacific reflection events (cf. Fig.

5 and A9). Even though geopotential height anomalies over Eurasia differ between regionalized and reflection events, the key feature of a westward propagating ridge over the North Pacific is well represented in regionalized events (cf. Fig. 3 and A10).

        A second potential limitation consists in the large time window (61 days) over which the space-time spectra are computed, which could smooth out short signals and prevent to effectively capture the reflection events' temporal evolution. However, only 37 days fully contribute to each day's space-time spectra due to tapering at the edges (Sect. 2.3). Choosing a shorter

time window could lead to the inability to measure the phase speeds of slow-propagating Rossby waves, which are crucial for understanding wave behaviour during reflection events. To support the ability of space-time spectra and the derived integrated metrics to effectively capture the temporal evolution of Rossby waves during reflection events, we highlight the correspondence between anomalies in wave activity in Fig. 5 and the presence and propagation of geopotential height anomalies in Fig. A2 during the onset and end of reflection events. The comparison between geopotential height anomalies and spectral power in

Figures 3, 7 and A11, also leads to a similar conclusion.

## 3.2 Connecting reflection events to tropospheric weather regime evolution

Messori et al. (2022) assessed the association between reflection events and transitions from the Pacific Trough (PT) to the Alaskan Ridge (AKR) weather regime. Given that the spatial patterns shown in Section 3.1.2 are consistent with this transition, we now discuss the characteristics of the regime shift and its similarity to reflection events.

During transitions from PT to AKR, the signal in geopotential height anomalies extends from the stratosphere to the lower troposphere, with an equivalent barotropic character similar to that during reflection events (Fig. 7 a–c). The transition begins about 12 days before the central date of AKR with a negative geopotential height anomaly centred at approximately 140 °W, i.e., the circulation anomaly characteristic of the PT. This is followed by a positive geopotential height anomaly, i.e. the AKR, which replaces the PT. As in the case of reflection events, both anomalies propagate westward (cf. Figs. 3 and 7). In parallel, a new negative anomaly downstream of the ridge forms at approximately 80°W during the AKR regime, and persists for some time afterwards (Fig. 7 a–c). The geopotential anomalies are most pronounced in the troposphere, likely due to the regimes being defined at 500 hPa (Lee et al., 2023a). Anomalies in WAF behave similarly during the regime transition compared to reflection events, i.e., downward WAF anomalies east of the geopotential height anomaly at 100 hPa before and during the peak intensity of the ridge (cf. Figs. 3 a and 7 a).

During the regime transition from PT to AKR, the evolution of $ISP_{west}$ anomalies is also comparable to that during reflection events. $ISP_{west}$ takes anomalously weak values during the PT regime and stronger-than-average values during AKR, resembling the change in Rossby wave evolution between the onset and end of reflection events (cf. Figs. 3 d and 7 d).

Changes in the Rossby wave spectrum from PT to AKR are also consistent with those observed during reflection events, except for the lack of statistically significant differences for wave-1 (Fig. 8 a, b). The difference between the AKR and PT spectra highlights the enhancement of spectral power for westward-propagating waves-3 and 4 (Fig. 8 c). Additional statistically significant anomalies occur in phase-space regions with climatologically low spectral power. As in the case of geopotential height anomalies, Rossby wave spectra anomalies of PT and AKR are more pronounced at 250 hPa than in the stratosphere (cf. Figs. 8 and A12).

In summary, the transition from a PT to an AKR weather regime resembles the atmospheric evolution during reflection events, confirming previous research (Messori et al., 2022). This includes the transition from a negative geopotential height anomaly over the Northeast Pacific around the onset to a ridge before the end of reflection events, which subsequently propagates westward and facilitates the formation of a new trough downstream over North America. About one-third of the transition events occur during reflection events (21 out of 62), and more than half of the reflection events (24 out of 45) include a PT-AKR regime transition if one includes transition events with the central date of AKR after the end of reflection events. We also note that PT-AKR weather regime transitions that do not occur during reflection events exhibit a different evolution of stratospheric geopotential height anomalies compared to those transitions happening during reflection events (cf. Figs. A13 and A14). Our analysis further points to the enhancement of westward-propagating wave-3 and wave-4 during reflection events being related to the PT to AKR regime shift. In contrast, the lack of a statistically significant wave-1 anomalies in the spectra of weather-

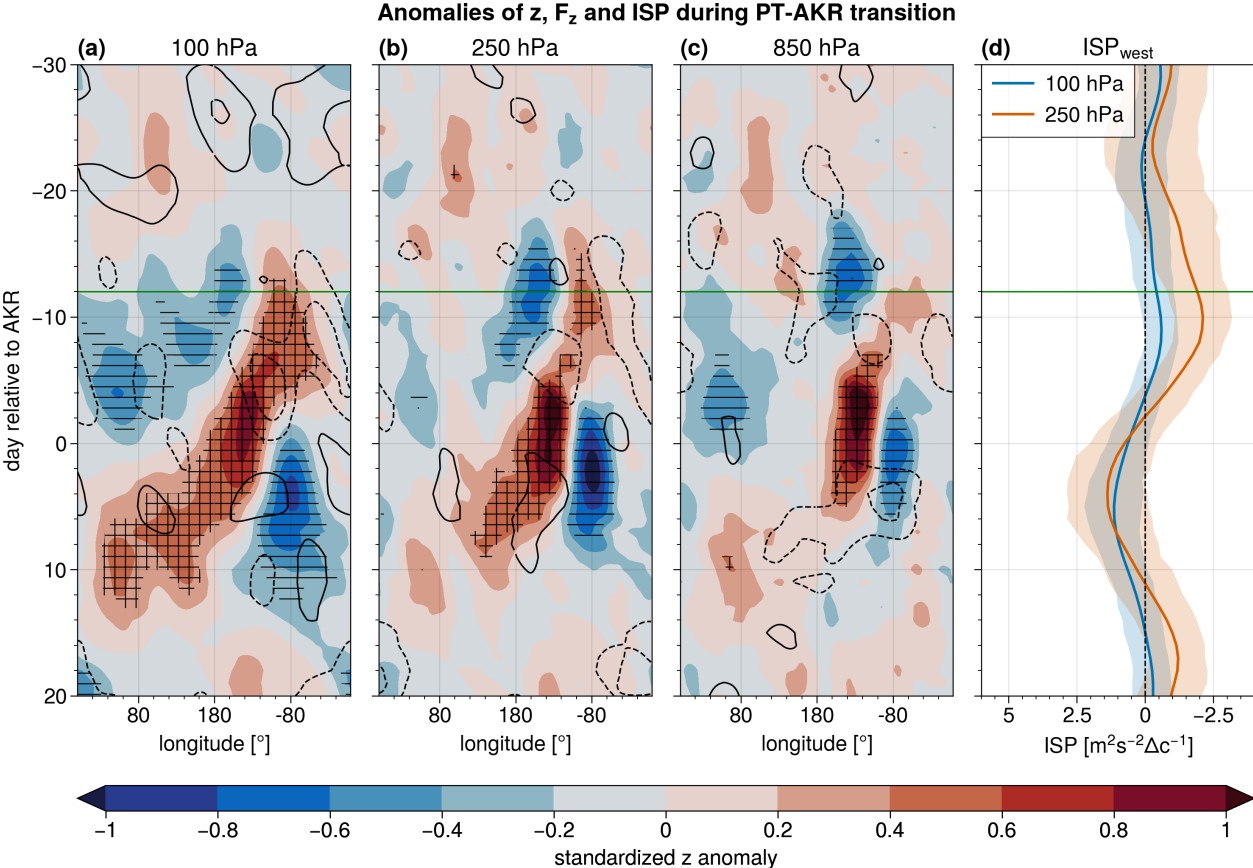

**Figure 7.** As in Fig. 3, but centred around the central date of AKR during PT–AKR transitions (Sect. 2.2).

regime transitions suggests that the latter signal arises from the stratosphere during reflection events (cf. Figs. 5 c, f and 8 c, f), and is not present in the PT-AKR transitions occurring outside stratospheric reflection events (cf. Figs. A15 and A16).

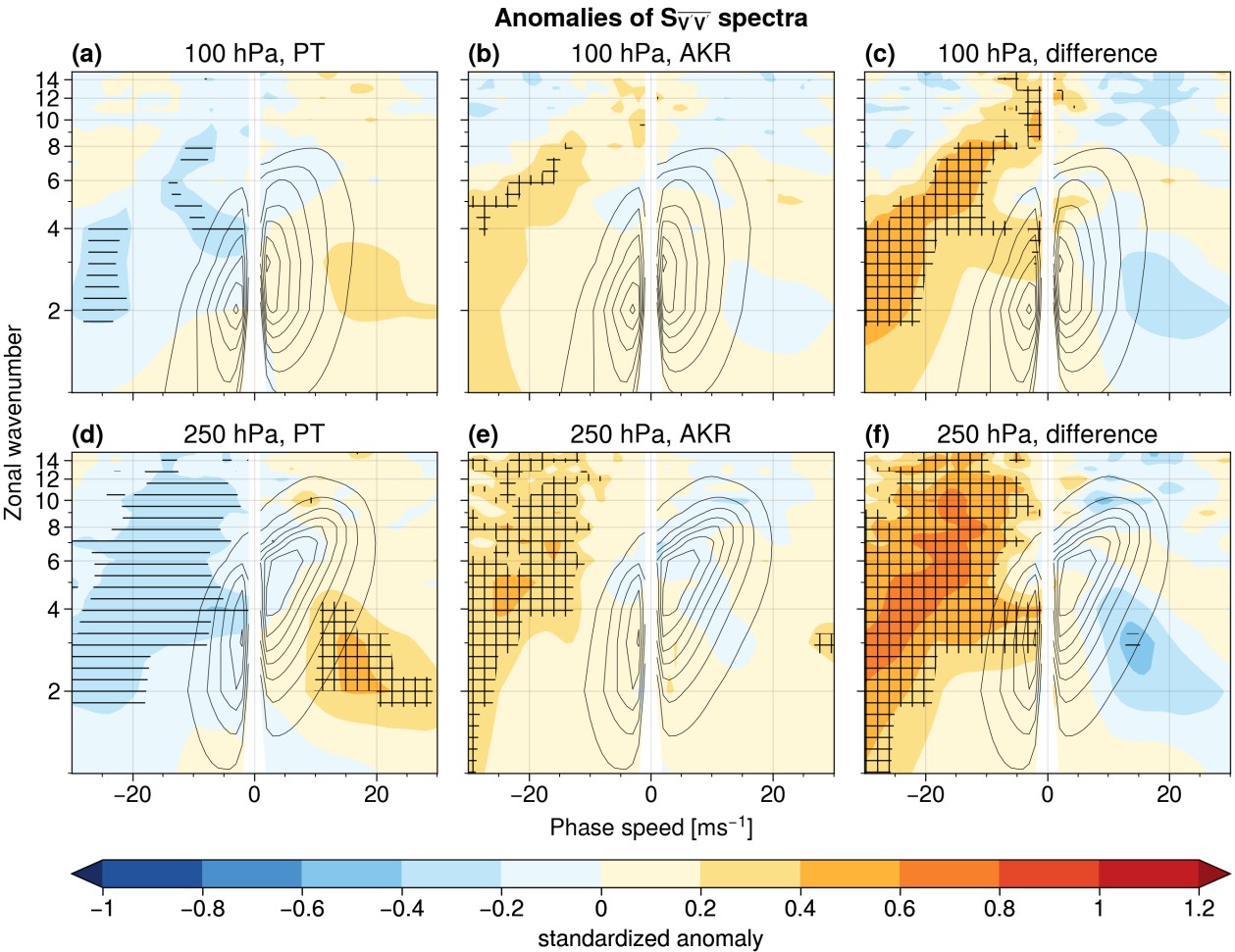

**Figure 8.** As Fig. 5, but for (a, d) PT regimes, (b, e) AKR regimes and (c, f) difference between PT and AKR regimes at 100 hPa and 250 hPa.

### 3.3 Impact on windiness over Europe

A growing literature connects cold spells in North America to extreme precipitation and/or wind in Europe, sometimes termed pan-Atlantic extremes (Messori et al., 2016; Leeding et al., 2023a, b; Riboldi et al., 2023; Messori and Faranda, 2023). In our study, we confirm previous results showing that reflection events over the North Pacific are connected to a drop in temperatures over North America (Messori et al., 2022; Kodera et al., 2016; Kretschmer et al., 2018; Matthias and Kretschmer, 2020). Additionally, we highlight the presence of hemispheric-scale anomalies in the mid-latitude circulation. Based on these results and on the literature on pan-Atlantic extremes, we investigate whether the reflection events are associated with near-surface (i.e., 10m) wind anomalies in Europe. Recently, Riboldi et al. (2023) highlighted two possible pathways connecting cold temperatures over North America and extreme wind events in Europe. The atmospheric evolution during reflection events resembles the first pathway, which involves the propagation of a Rossby wave train from the North Pacific. This results in an intensification of the North Atlantic eddy-driven jet and extreme winds over northwestern Europe a few days after the North American cold spell peak. The similar atmospheric configuration underscores the previously unreported connection between reflection events in the North Pacific and wind extremes in Europe.

As the coldest temperatures over North America typically occur a few days before the end of a reflection event (Messori et al., 2022; Millin et al., 2022), one would expect extreme winds over Europe around the end of reflection events. Confirming this expectation, one day after the end date there is a noticeable increase in average wind speeds across northwestern Europe, albeit only of the order of 1 $ms^{-1}$ (Fig. 9 a and Fig. A17). Despite this, wind speeds exceed the local 98th percentile – often used to identify damaging wind extremes – two to five times more often than on average (Fig. 9 b). Consistent with the first pathway outlined by Riboldi et al. (2023), we observe a strengthening of the jet stream over the North Atlantic, with the jet exit region over northwestern Europe (Fig. 9 c and Fig. A17). The jet stream anomalies that we find are in line with earlier findings about the connection between reflection events and atmospheric dynamics over the North Atlantic (Kodera et al., 2016) and with the impact of stratospheric downward wave events on the tropospheric circulation (Dunn-Sigouin and Shaw, 2015; Shaw et al., 2014; Shaw and Perlwitz, 2013).

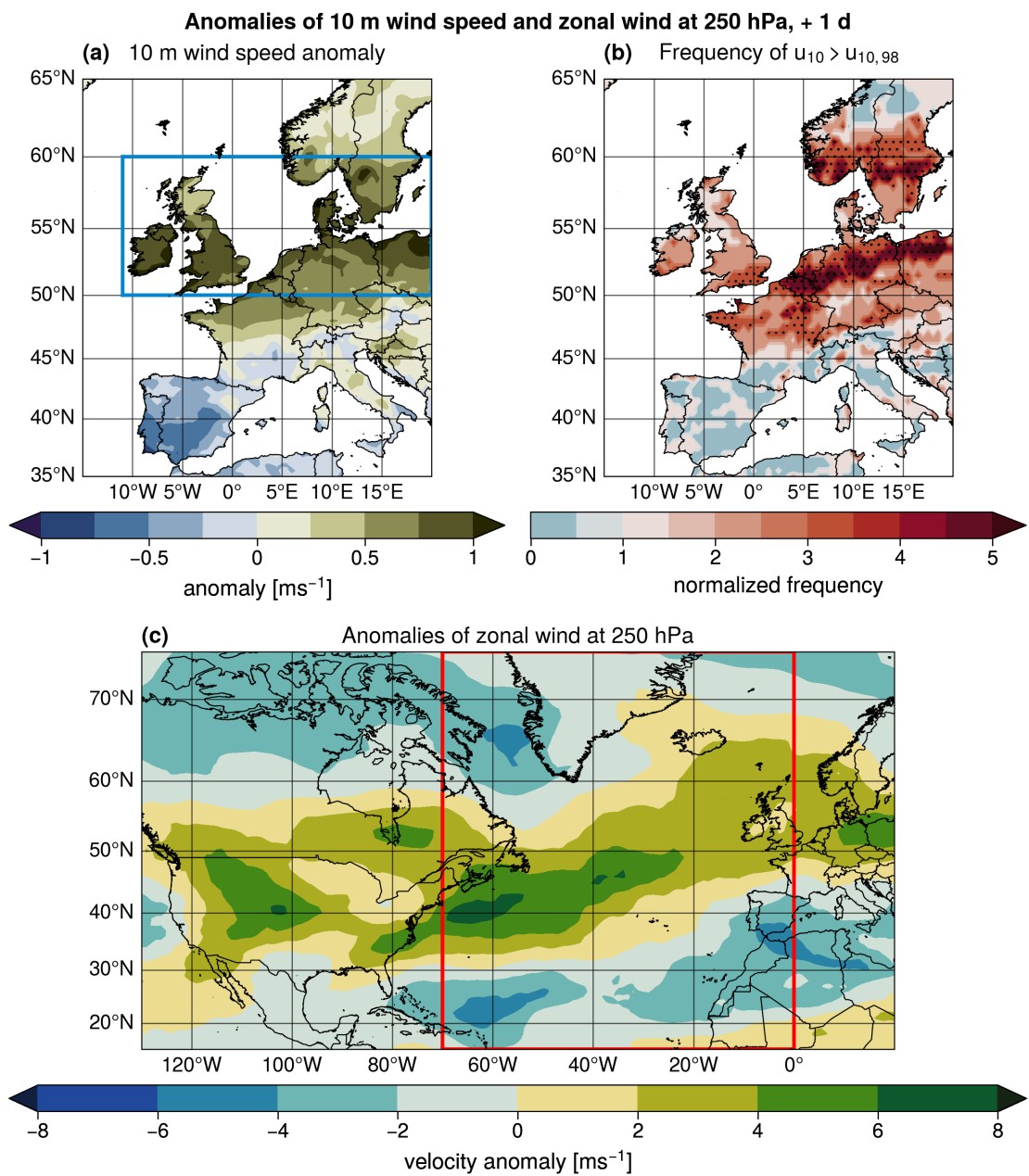

**Figure 9.** (a) Composite 10-m wind speed anomaly in Europe one day after the end of the reflection events. (b) Frequency of 10-m wind speed exceeding the local 98th percentile at each grid point, normalized with respect to the climatological frequency of 2% ($\geq 1$: above average, $\leq 1$: below average). (c) 250 hPa zonal wind anomalies. Stippling in (b) indicates regions where the 98th percentile of 10-m wind speed is exceeded at least 2.5 times more often than the DJFM average. The blue box in (a) and the red box in (c) indicate the regions used to calculate the mean 10-meter wind and jet strength, respectively, shown in Fig. A17.

## 4 Discussion and conclusions

During stratospheric wave reflection events over the North Pacific–North American sector, anomalies of meridional eddy heat flux $v'T'$ are most pronounced in the lower stratosphere, and alternate in sign before the onset and after the end of the reflection events. This alternation is connected to the formation of a large-scale ridge and its westward propagation from Canada to Siberia, resembling a shift from a Pacific Trough (PT) to an Alaskan Ridge (AKR) circulation. As already mentioned in earlier research (e.g., Messori et al., 2022; Cohen et al., 2022), around the end of reflection events, positive $v'T'$ anomalies at lower

tropospheric levels over Canada correspond to a decrease in 2-m air temperatures. In this work, we additionally show the connection of reflection events over the North Pacific to anomalously frequent wind extremes over Europe.

A space-time spectral analysis of Rossby waves indicates that stratospheric wave reflection events over the North Pacific involve different scales of waves. The change in tropospheric circulation is connected to an enhancement of westward propagating waves-3 and 4, which is equally present in the shift from PT to AKR weather regimes. The westward propagation of

medium-scale Rossby waves in the upper troposphere could be linked to internal tropospheric free Rossby wave dynamics; their interaction and superposition with planetary-scale waves from the stratosphere may contribute to the observed signature of downward reflection. Indeed, the enhancement of westward-propagating wave-1 during stratospheric wave reflection events suggests that the evolution of $v'T'$ anomalies may be influenced by the coupling between stratospheric and tropospheric waves.

The vertical alignment of Rossby waves, typical of reflection events and differing from the normal westward tilt with height

(see Perlwitz and Harnik, 2003), can be induced by the westward propagation of Rossby waves in the troposphere during reflection events. One mechanism that possibly further enhances the westward propagation of Rossby waves is the deceleration of zonal-mean zonal winds in the stratosphere (Fig. A5 in Messori et al., 2022) and upper troposphere (Fig. A18), resulting from Rossby wave absorption in the stratosphere and downward propagation of the signal. Following the period of wave reflection over the North Pacific, upward propagating Rossby waves from the Northwest Pacific can be absorbed, weakening

the stratospheric polar vortex (Matsuno, 1971; Sjoberg and Birner, 2012; Polvani and Waugh, 2004; Reichler and Jucker, 2022). In most cases, the disruption of the stratospheric polar vortex seems to be insufficient to cause an SSW, and there is only a weak correspondence between reflection events and SSWs Messori et al. (2022). On the other hand, the anomalously strong planetary-scale Rossby waves in the stratosphere are consistent with the potential 'stretching' of the stratospheric polar vortex connected to reflection events (Messori et al., 2022; Cohen et al., 2022). In line with the enhanced westward-propagating

Rossby waves, the strengthened Alaskan Ridge that characterizes reflection events propagates westward. This weakens the positive $v'T'$ anomalies over the Siberian domain and puts an end to reflection events. The key atmospheric features associated with the different stages of the reflection events are summarized in Fig. 10.

While wave amplification over Eurasia can trigger reflection events over the North Pacific (Cohen et al., 2022), the tropical Pacific could also play a role due to its link with blocking over the North Pacific and Alaska (e.g., Renwick and Wallace, 1996;

Henderson and Maloney, 2018; Carrera et al., 2004). Indeed, previous work has connected blocking patterns and stratospheric wave reflection over the North Pacific, highlighting their role in modulating 2-m air-temperature over North America (Guan et al., 2020; Kodera et al., 2013; Millin et al., 2022). In this context, the activity of wave-1 plays a crucial role (Ding et al., 2022;

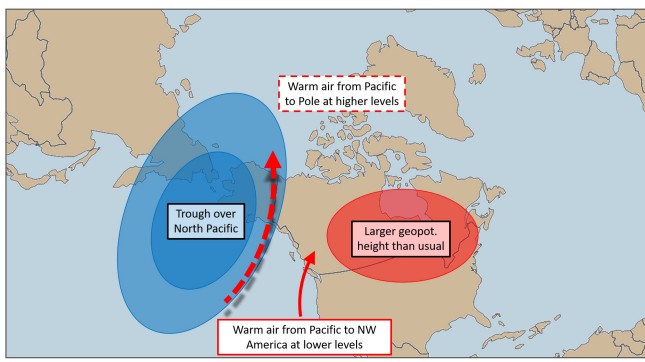

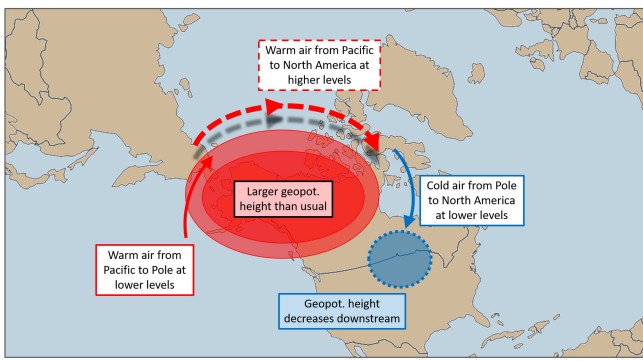

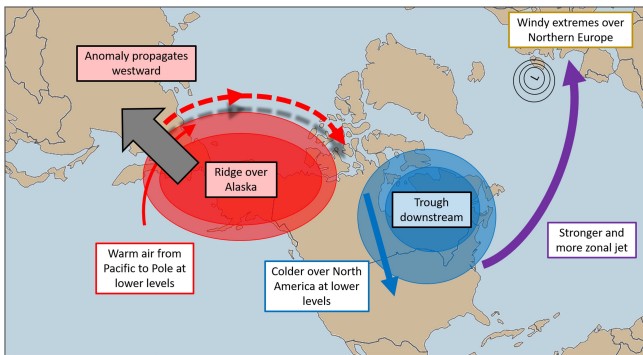

**Figure 10.** Summary figure of key atmospheric features (a) around the onset of reflection events (b) during reflection events, and (c) around the end of reflection events.

Kodera et al., 2016) by contributing to troposphere-stratosphere coupling (Dunn-Sigouin and Shaw, 2015; Shaw and Perlwitz, 2013). This agrees with our analysis highlighting the importance of wave-1 during stratospheric reflection events. Complementary to the role of wave-1, a maximum of zonal winds in the middle stratosphere has been suggested as a necessary condition

for stratospheric wave reflection (Perlwitz and Harnik, 2004; Messori et al., 2022). Another aspect deserving further investigation is the connection between reflection events and the presence of westward-propagating (retrograding) waves (Branstator, 1987; Kushnir, 1987; Madden and Speth, 1989; Raghunathan and Huang, 2019), together with the link to the recently-defined 'Stratosphere-Troposphere Oscillation' (Shen et al., 2023). The answers to these questions will increase our understanding of the interactions between tropospheric Rossby waves and North-Pacific wave reflection events, and the connection of the latter to other modes of stratosphere-troposphere coupling.

Although the exact mechanisms driving the onset or end of stratospheric reflection events over the North Pacific remain to be unravelled, our study has highlighted the presence of a systematic dynamical evolution. The connection between large-scale reflection events, characterised by inherent extended-range predictability, and surface weather conditions, such as North American cold spells and European windy extremes, provides valuable information for extending the forecast timescales of impactful weather events.

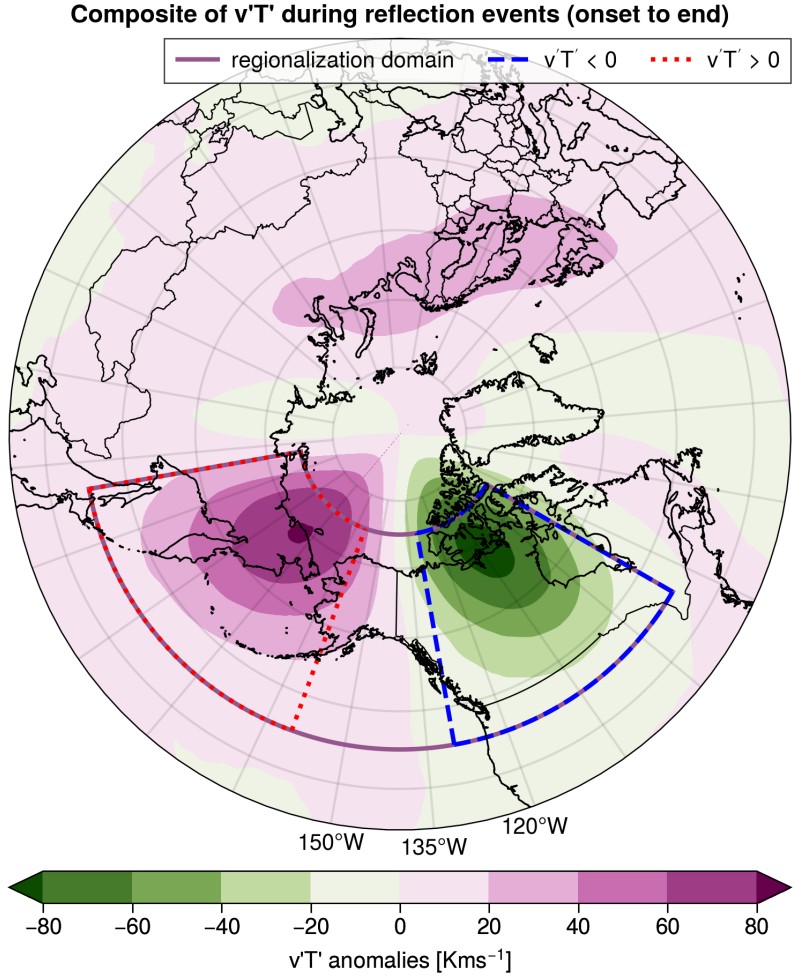

**Figure A1.** Contours: composite of $v'T'$ anomalies during all days of reflection events from their onset until the end. Boxes indicate the location of (red, dotted) the Siberian domain located between 45–75°N, 140–200°E, (blue, dashed) Canadian domain located between 45–75°N, 230–280°E, used to define reflection events, and (purple) the domain used to obtain regionalized events (Section 2.2).

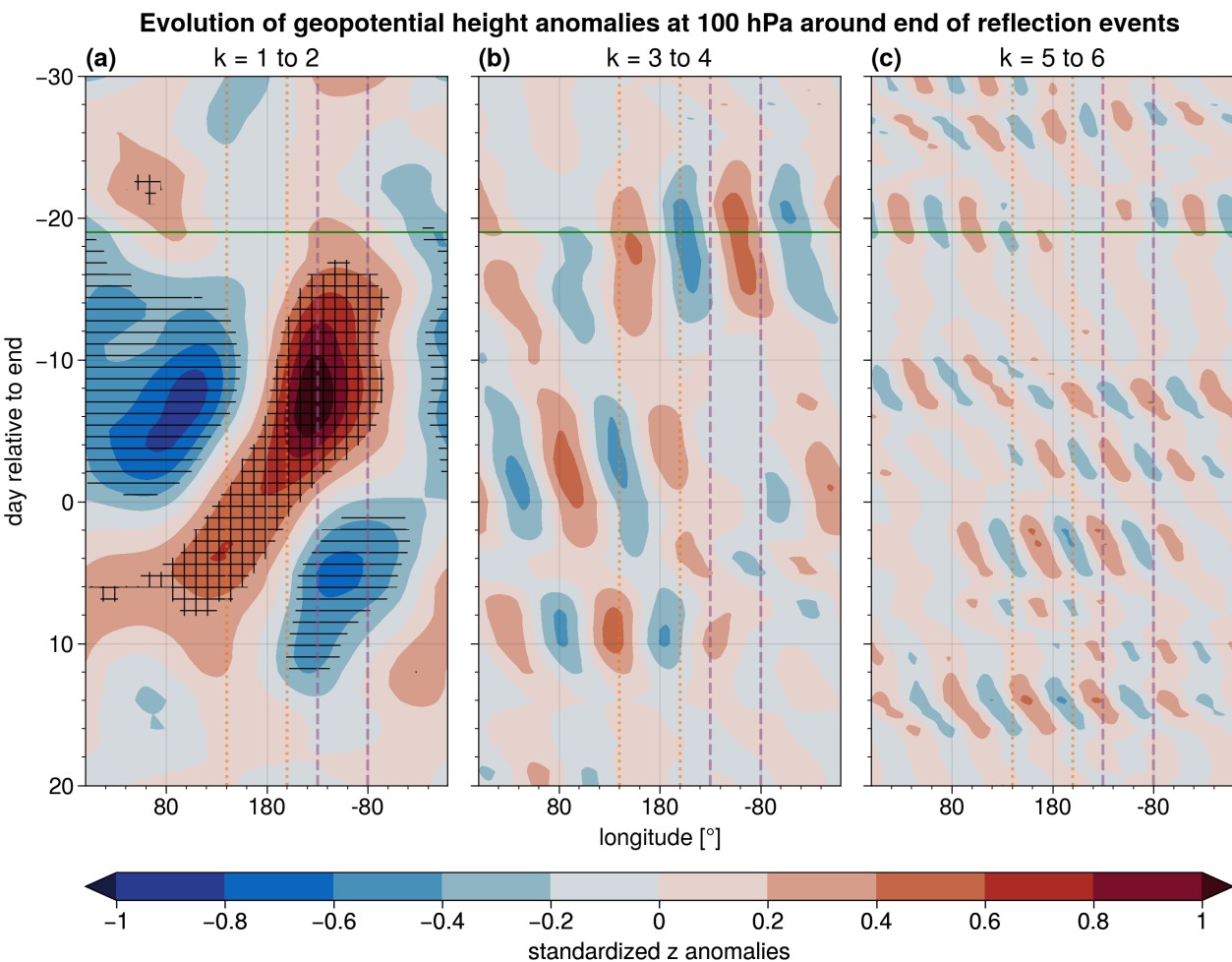

**Figure A2.** Hovmöller diagram of standardized anomalies of geopotential height averaged between $45°N - 75°N$ centred around the end date of reflection events at 100 hPa for (a) wave-1 to 2, (b) wave-3 to 4 and (c) wave-5 to 6 in shading. Horizontal hatching marks statistically significant negative anomalies and cross-hatching statistically significant positive anomalies. The continuous horizontal green line shows the median onset time of reflection events. The vertical lines mark longitudes of the Siberian (orange, dotted) and Canadian (purple, dashed) domains.

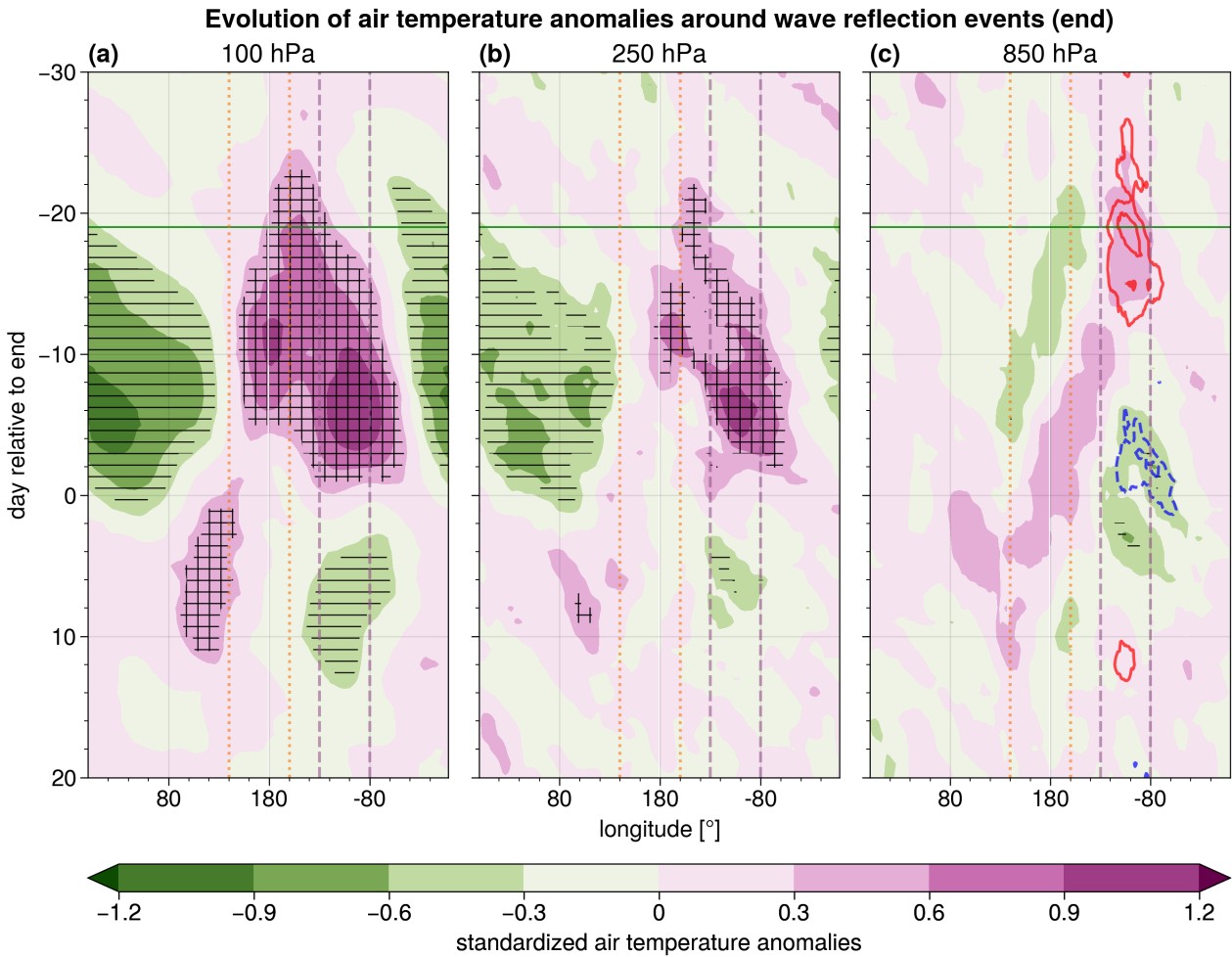

**Figure A3.** As Fig. 2, but for air temperature at (a) 100 hPa, (b) 250 hPa and (c) 850 hPa.

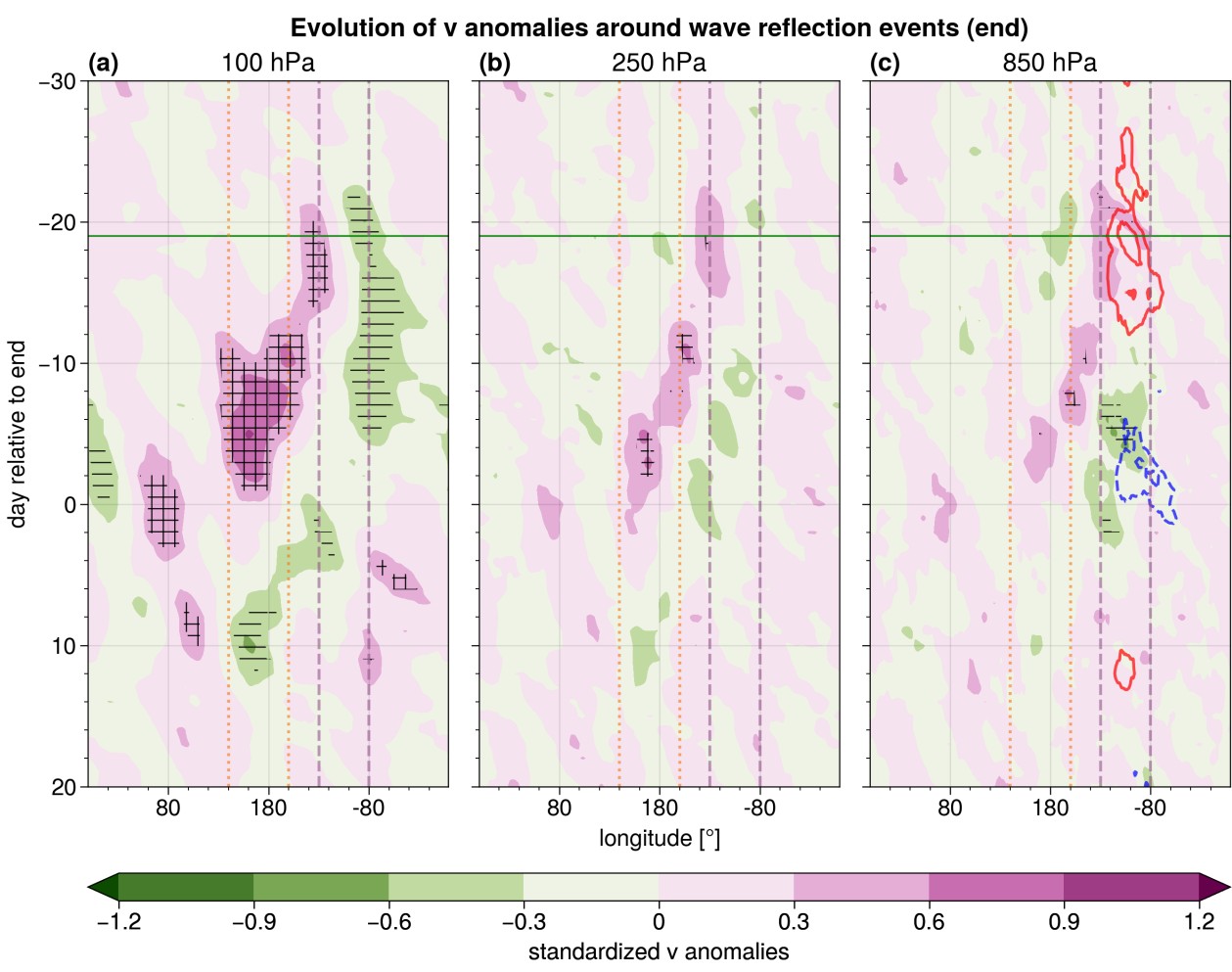

**Figure A4.** As Fig. 2, but for meridional wind speed at (a) 100 hPa, (b) 250 hPa and (c) 850 hPa.

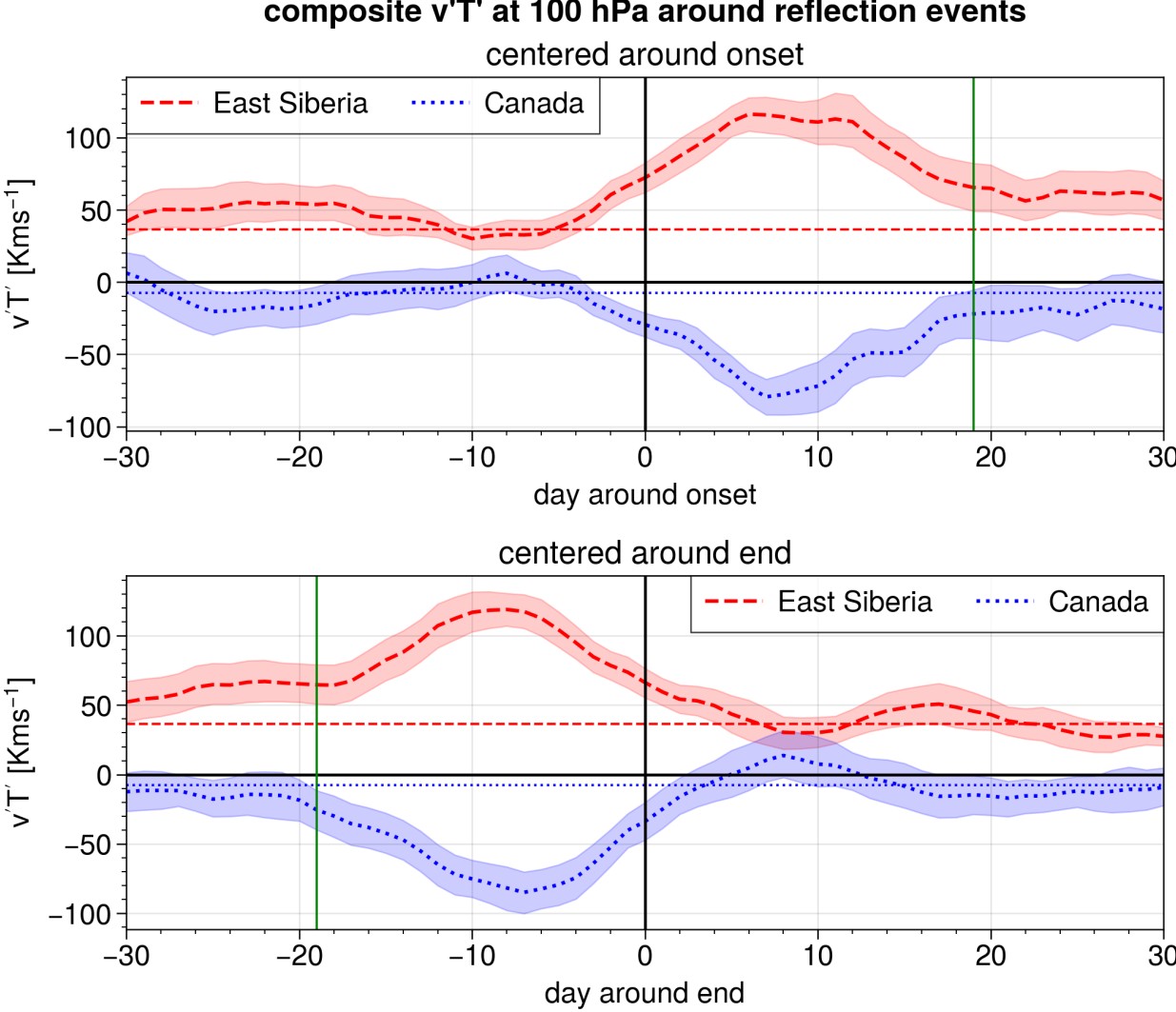

**Figure A5.** Meridional eddy heat flux $v'T'$ at $100\,\mathrm{hPa}$ over East Siberia (red, dashed) and Canada (blue, dotted), (a) centred around the onset date, (b) centred around the end date of reflection events. The thin horizontal lines show the DJFM mean in the respective region, shading indicates the 95% confidence interval on the mean, assessed with bootstrapping. The vertical green line in (a) shows the median end, and in (b) the median onset of reflection events.

**z anomaly, v'T' anomaly and temperature anomaly during wave reflection events**

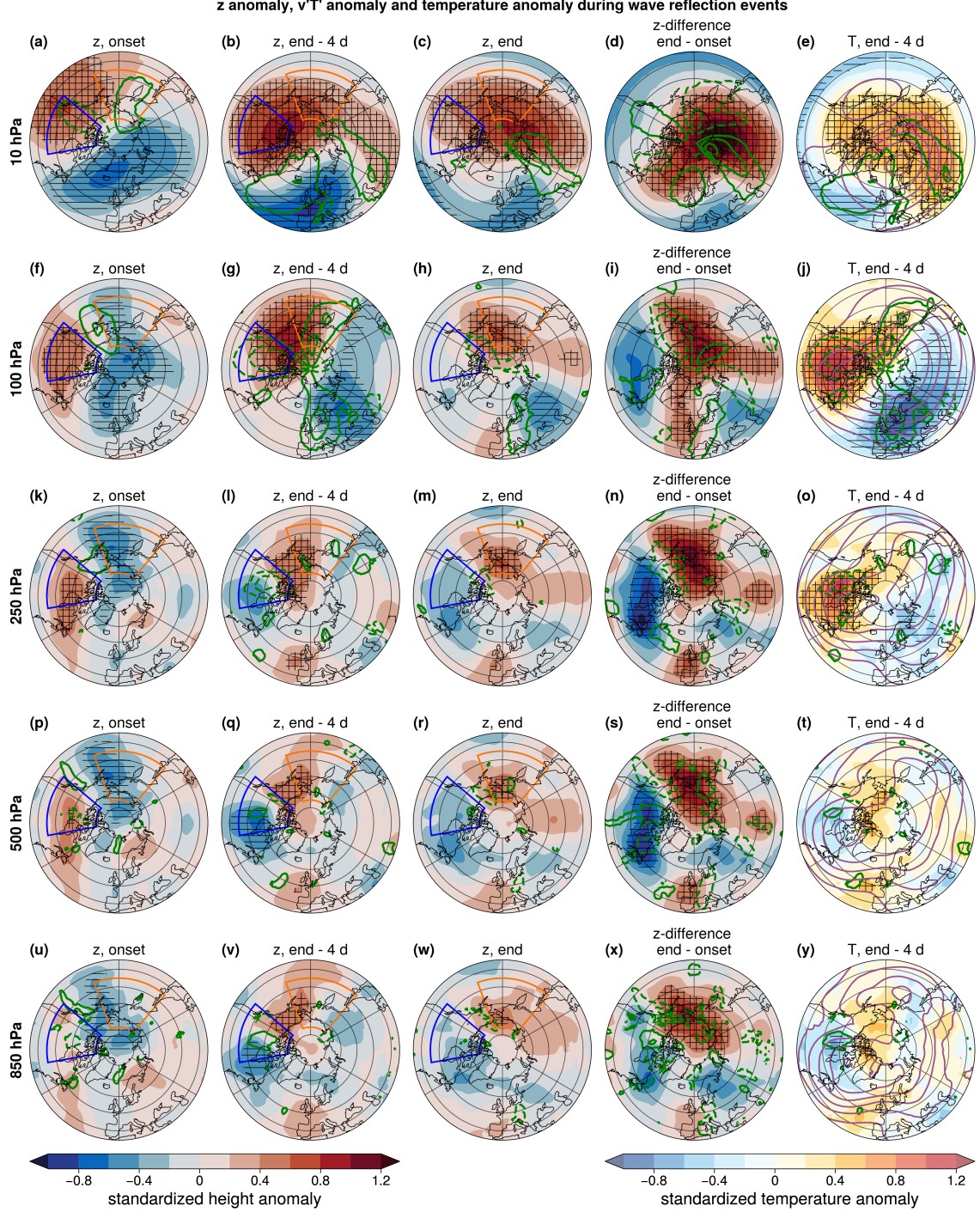

**Figure A6.** As Fig. 4, but for pressure levels 10 hPa, 100 hPa, 250 hPa, 500 hPa and 850 hPa.

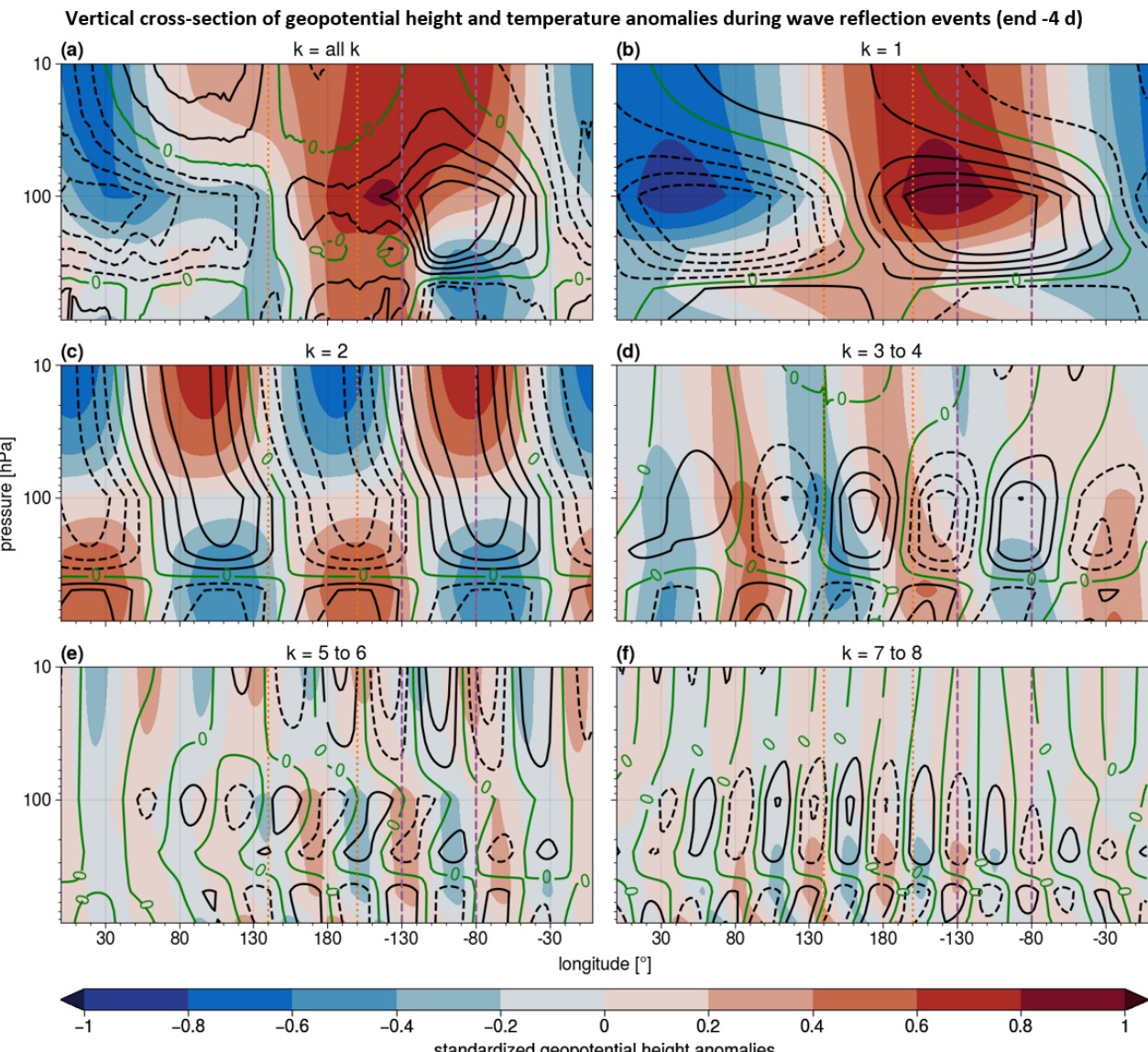

**Figure A7.** Vertical cross-section of standardized geopotential height anomalies (shading) and standardized temperature anomalies (black contours in steps of 0.2) at 60°N four days before the end of reflection events for (a) all wavenumbers, (b) wave-1 (c) wave-2, (d) wave-3 to 4, (e) wave-5 to 6 and (f) wave-7 to 8. The vertical lines mark longitudes of the Siberian (orange, dotted) and Canadian (purple, dashed) domains.

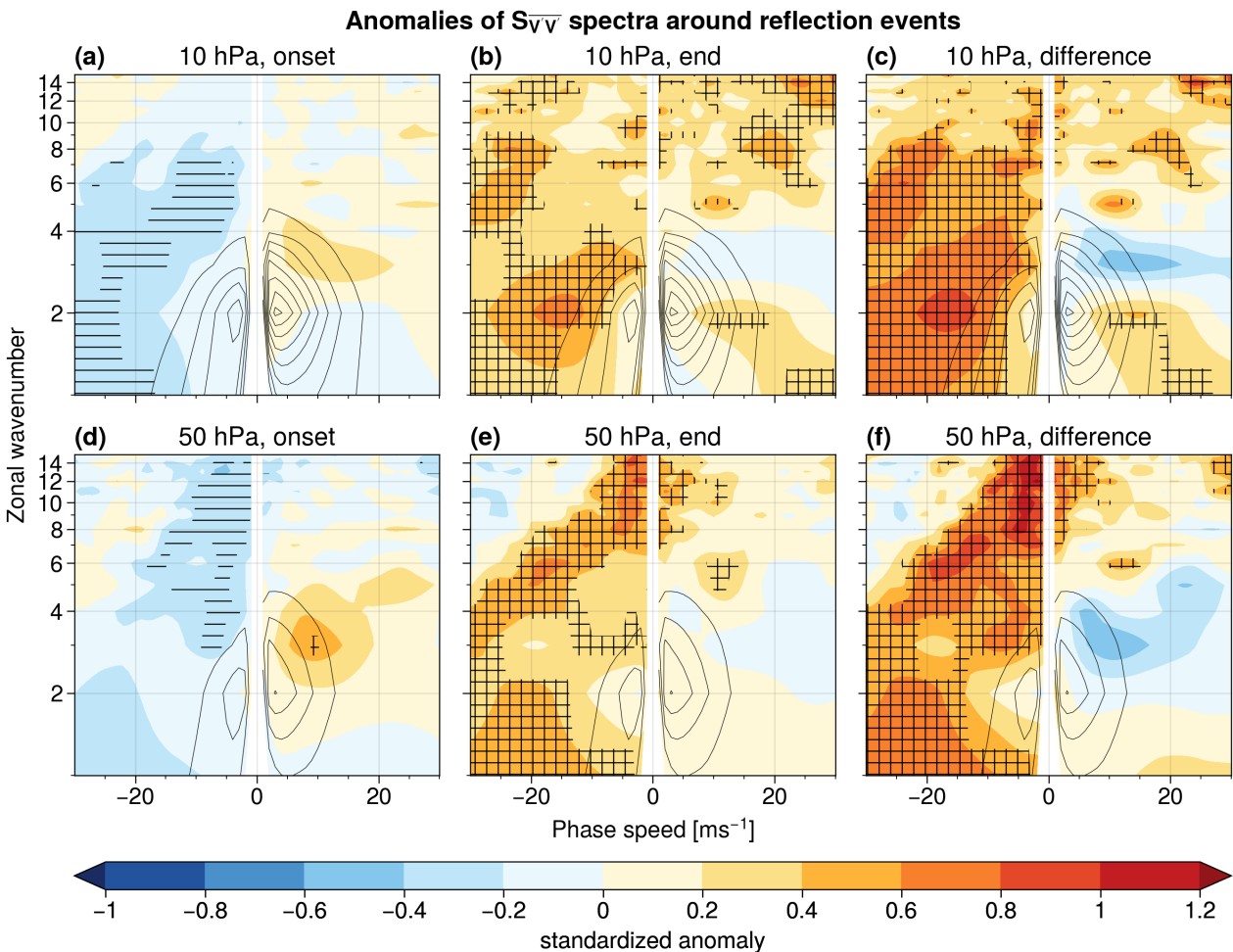

**Figure A8.** As Fig. 5, but for pressure levels 10 hPa and 50 hPa.

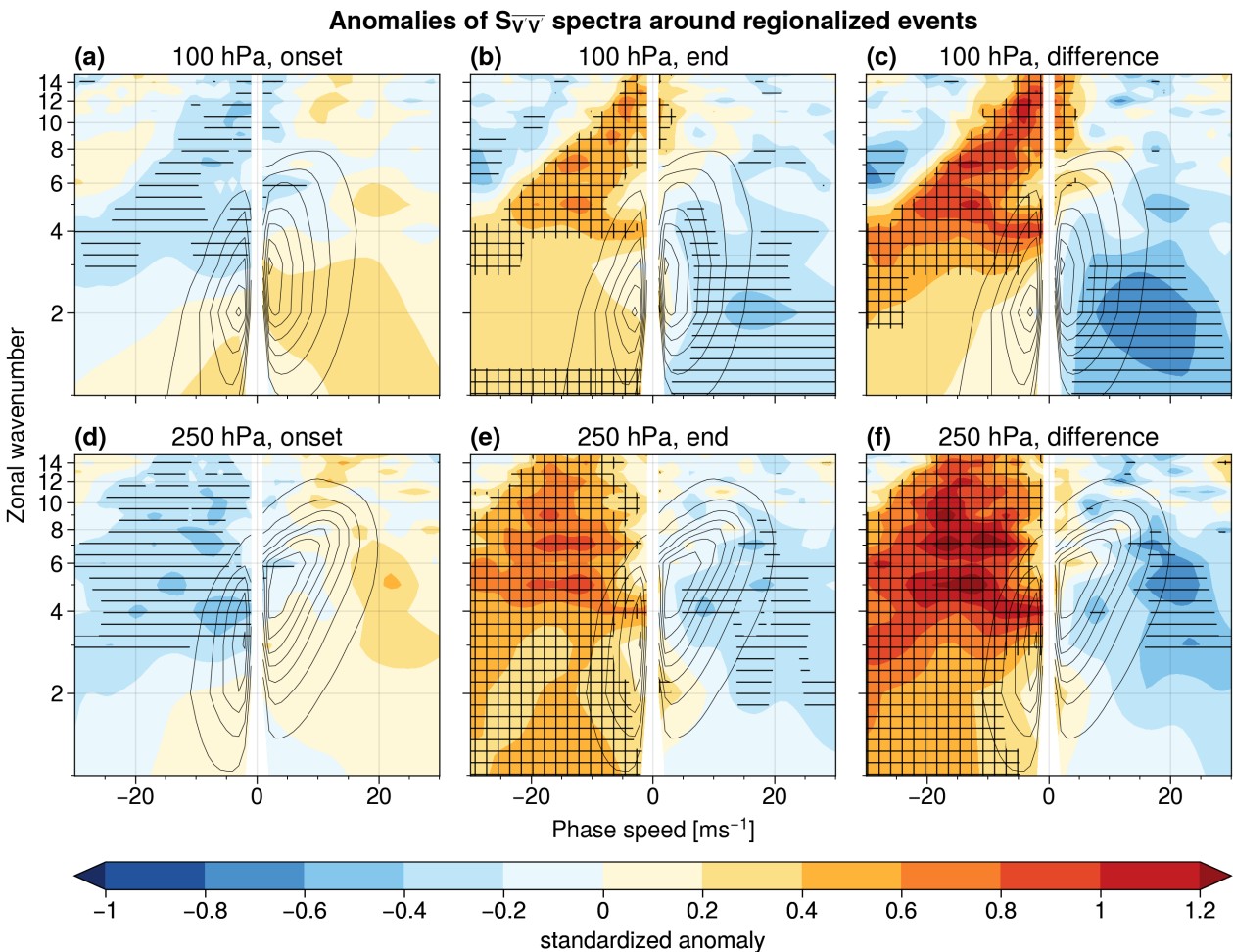

**Figure A9.** As Fig. 5, but for regionalized events (as defined in Sect. 2.2).

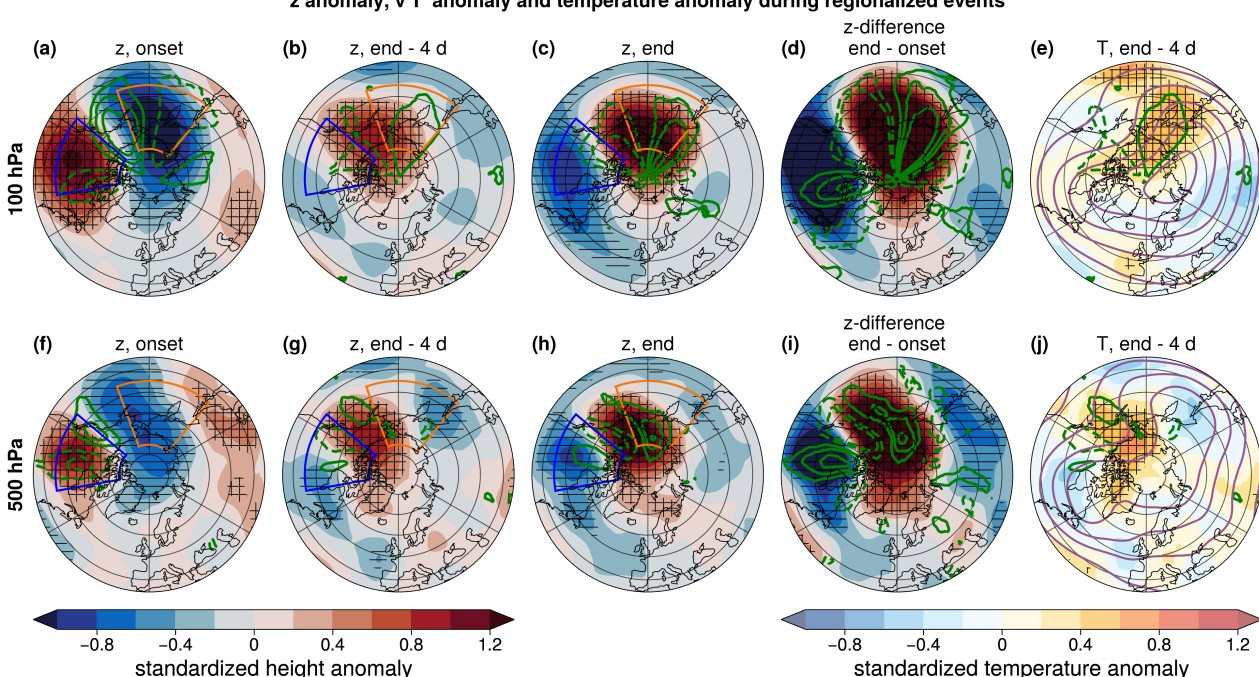

**Figure A10.** As Fig. 4, but for regionalized events (as defined in Sect. 2.2).

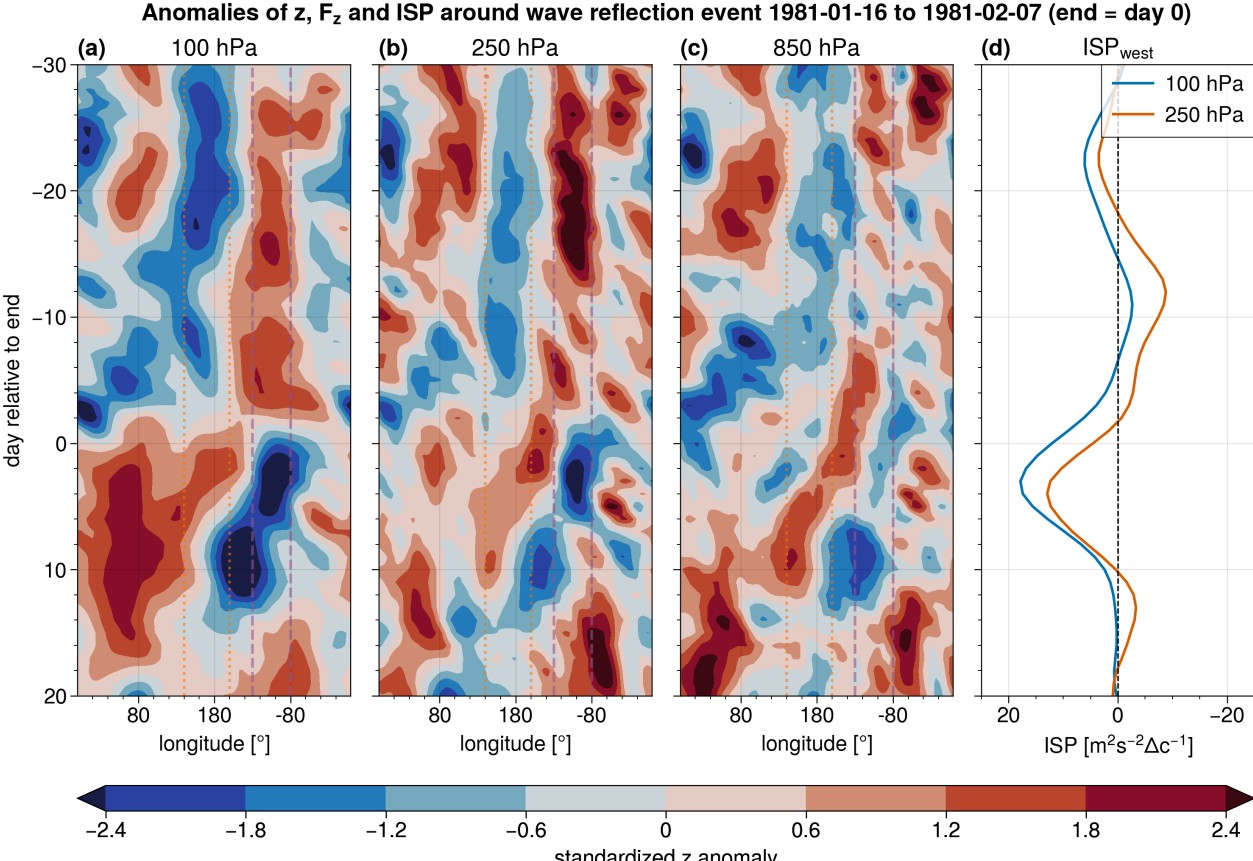

**Figure A11.** Hovmöller diagram of standardized anomalies of geopotential height averaged over 45°N – 75°N centred on the end date of the January-February 1981 reflection event. The panels show the: (a) 100 hPa, (b) 250 hPa and (c) 850 hPa levels. (d) Time series of ISP for westward-propagating Rossby waves at 100 hPa and 250 hPa with the x-axis being flipped, so that enhanced activity of westward-propagating Rossby waves lies to the left of the zero line. The vertical lines mark the longitudes of the Siberian (orange, dotted) and Canadian (purple, dashed) domains.

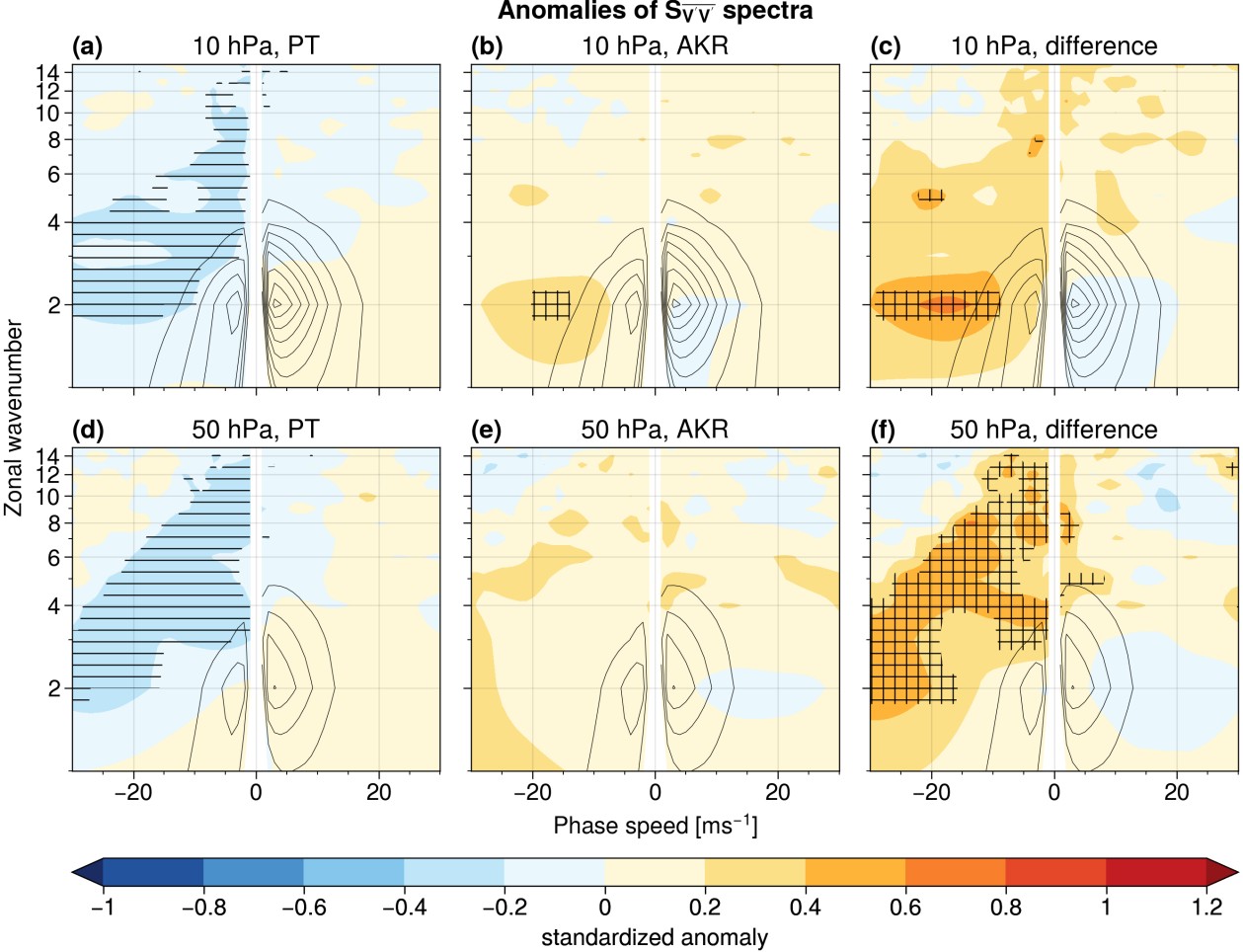

**Figure A12.** As Fig. 5, but for (a, d) PT regimes, (b, e) AKR regimes and (c, f) difference between PT and AKR regimes at 10 hPa and 50 hPa.

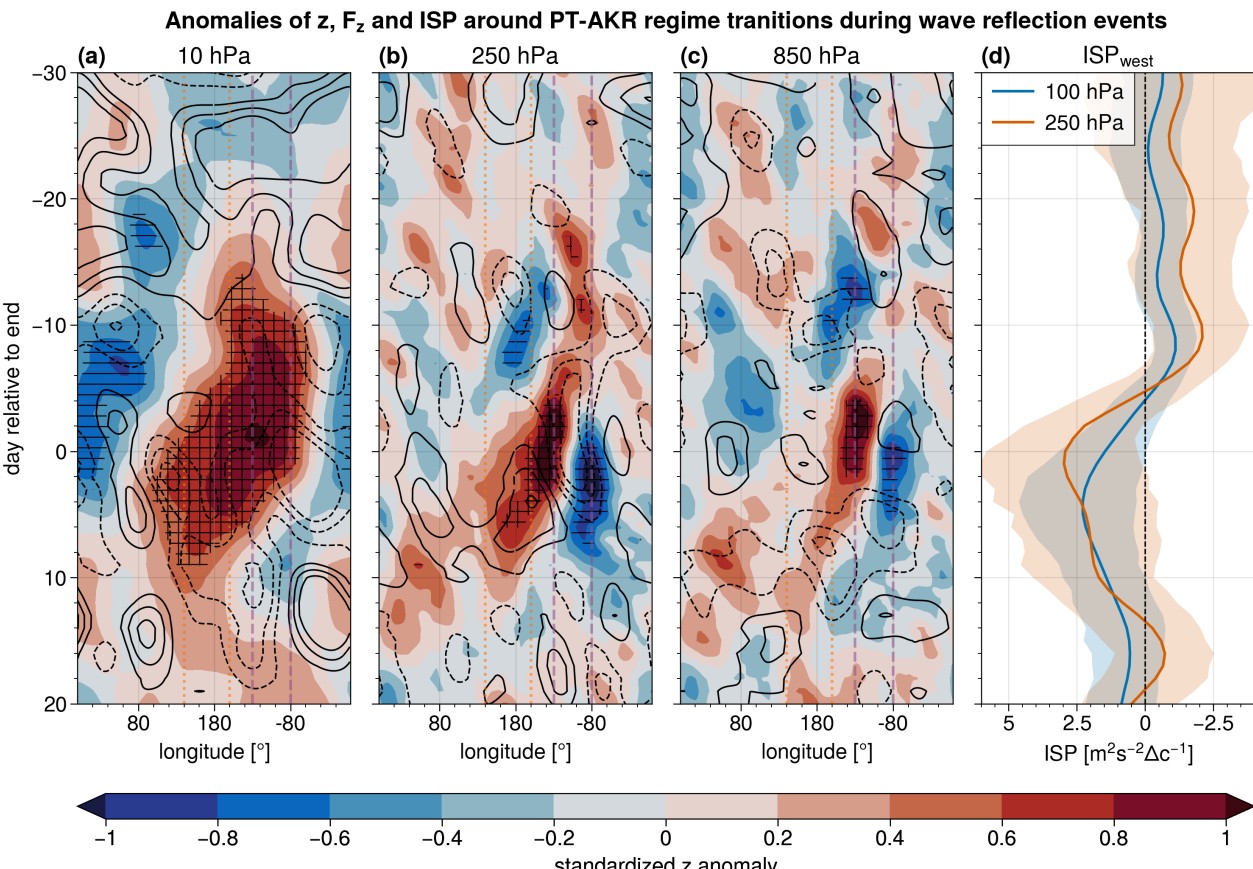

**Figure A13.** As Fig. 3, but centred on the central date of AKR for PT–AKR tranitions occurring during reflection events.

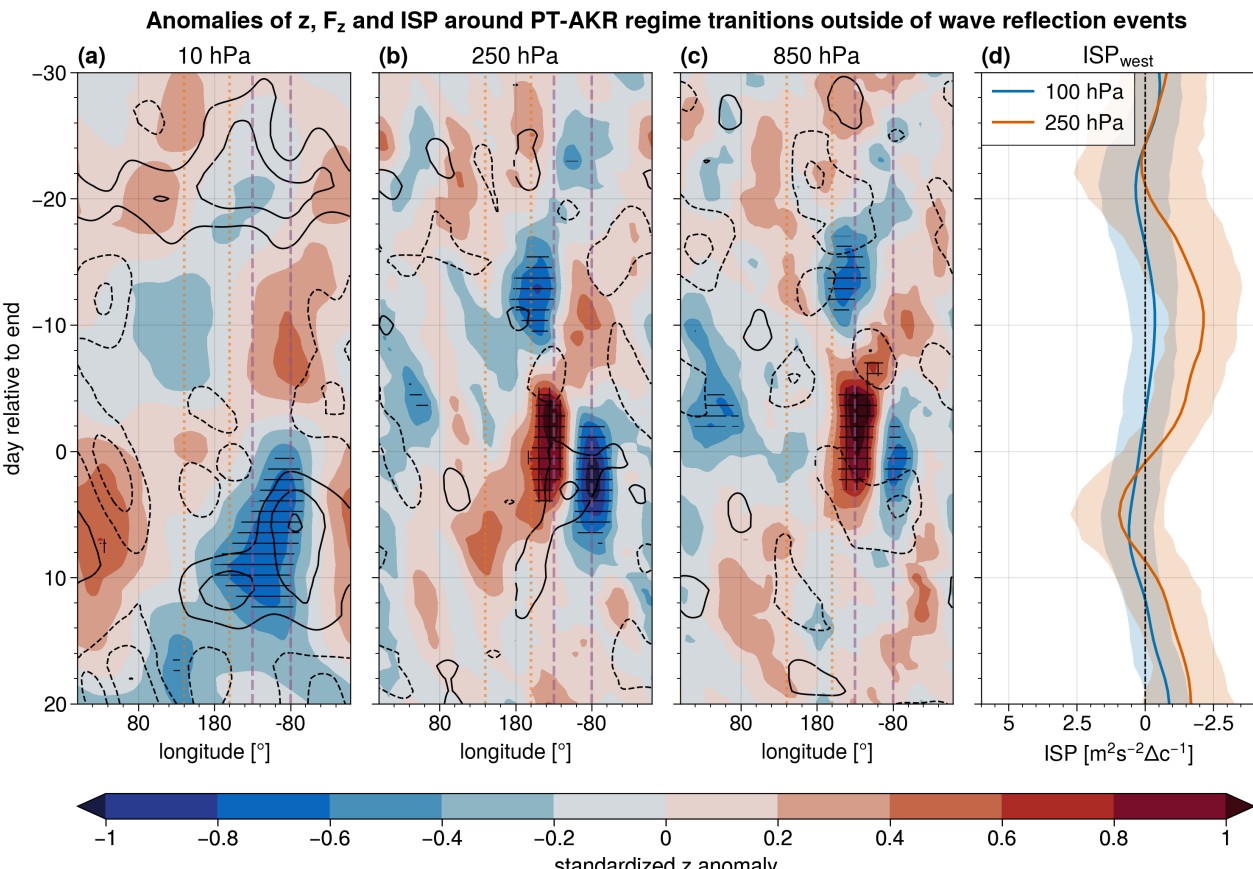

**Figure A14.** As Fig. 3, but centred on the central date of AKR for PT–AKR transitions occurring outside reflection events.

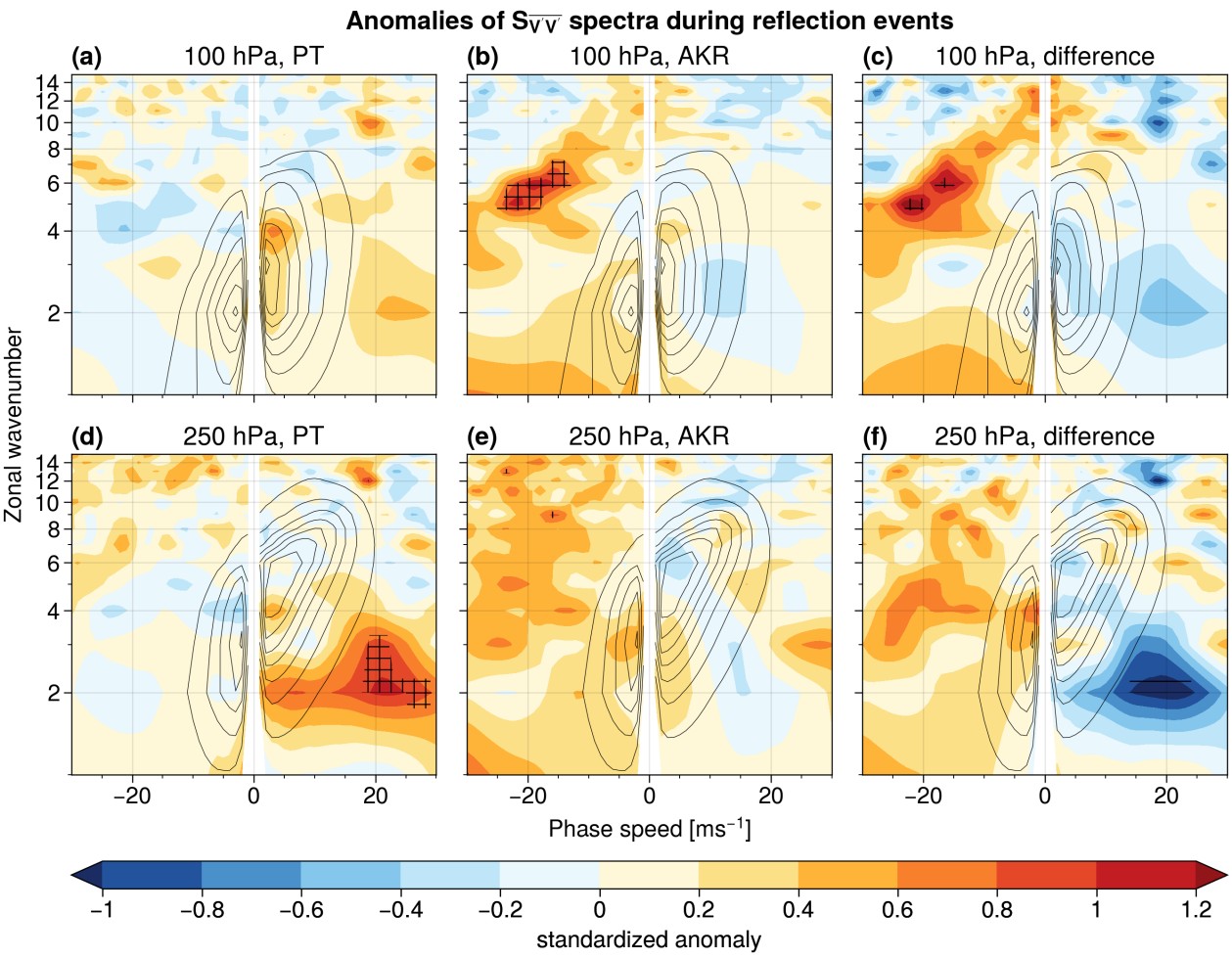

**Figure A15.** As Fig. 5, but for (a, d) PT regimes, (b, e) AKR regimes and (c, f) difference between PT and AKR regimes occurring during reflection events.

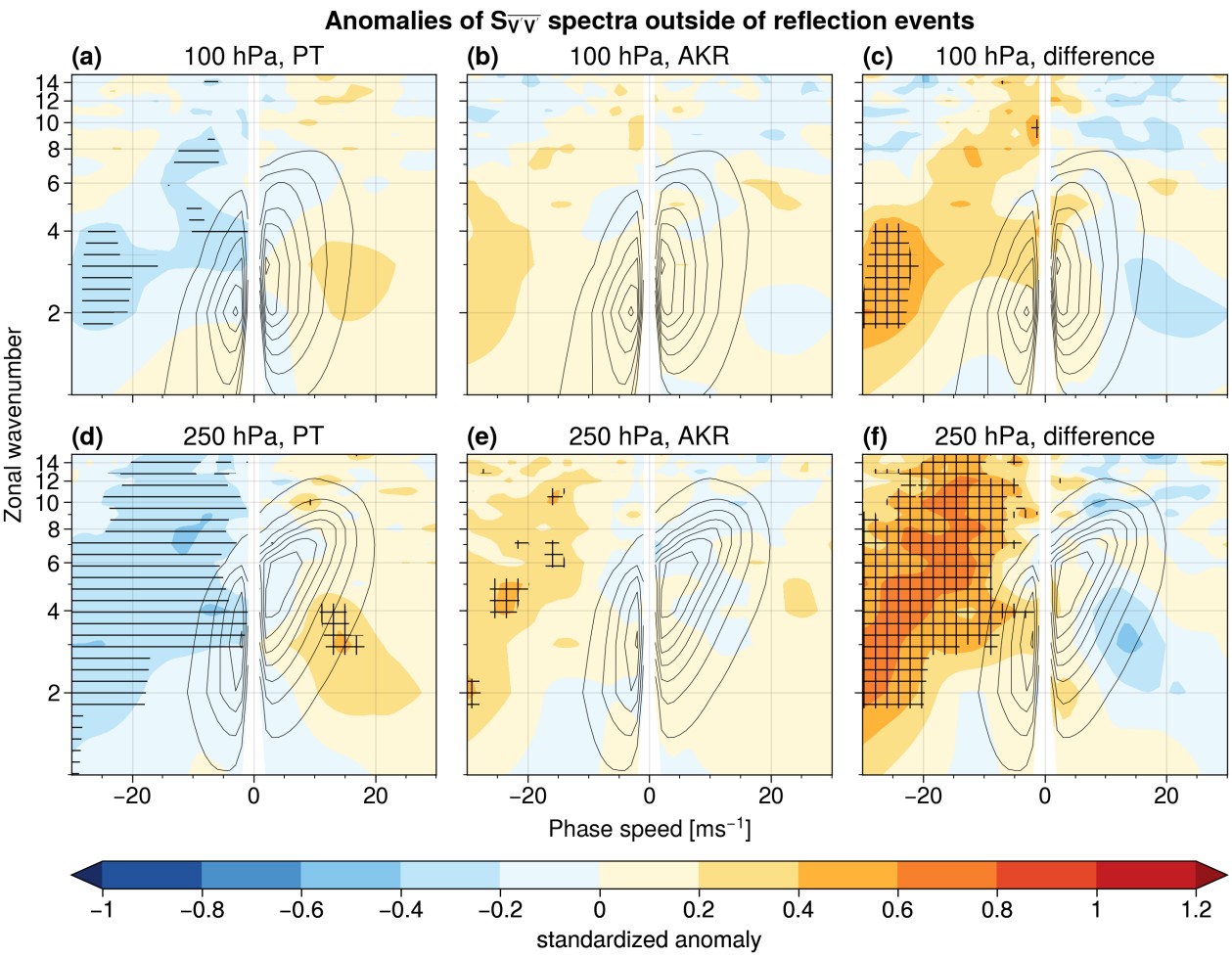

**Figure A16.** As Fig. 5, but for (a, d) PT regimes, (b, e) AKR regimes and (c, f) difference between PT and AKR regimes occurring outside reflection events.

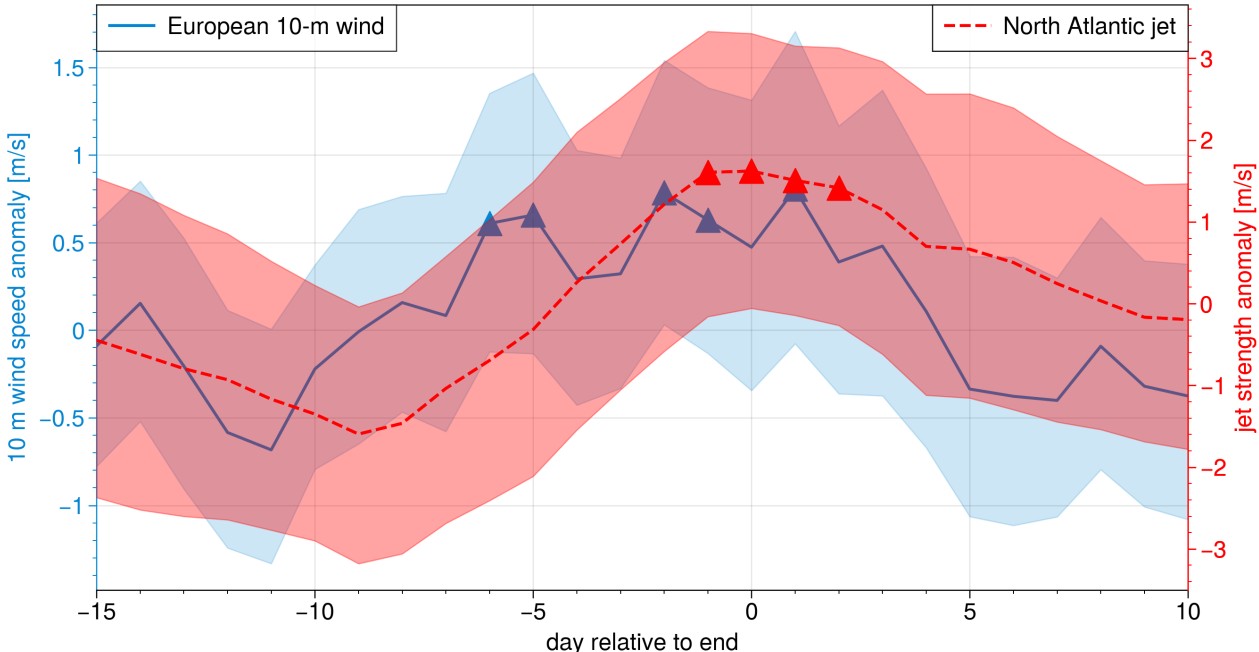

**Figure A17.** Anomaly of cosine-latitude weighted mean of 10-m wind speed over land grid points of Northern Europe (blue curve; 50-60°N, 11°W-20°E; blue box in Fig. 9 a) and anomaly of the jet strength (red, dashed curve). The jet strength is computed similarly to Woollings et al. (2010), but adapted to our study: the zonal average of the 250 hPa zonal wind over the North Atlantic (15–75°N, 70–0°W; red box in Fig. 9 c) is smoothed with a centred 10-day running mean at each latitude. Subsequently the maximum wind speed is identified to define the jet strength on a given day which is then deseasonalized following Sect. 2.1. Shading indicates the 95% confidence interval of the mean and triangles highlight dates where the composite anomaly is above the 95th percentile of a resampled distribution of 10 000 means of 45 randomly selected days from the climatology.

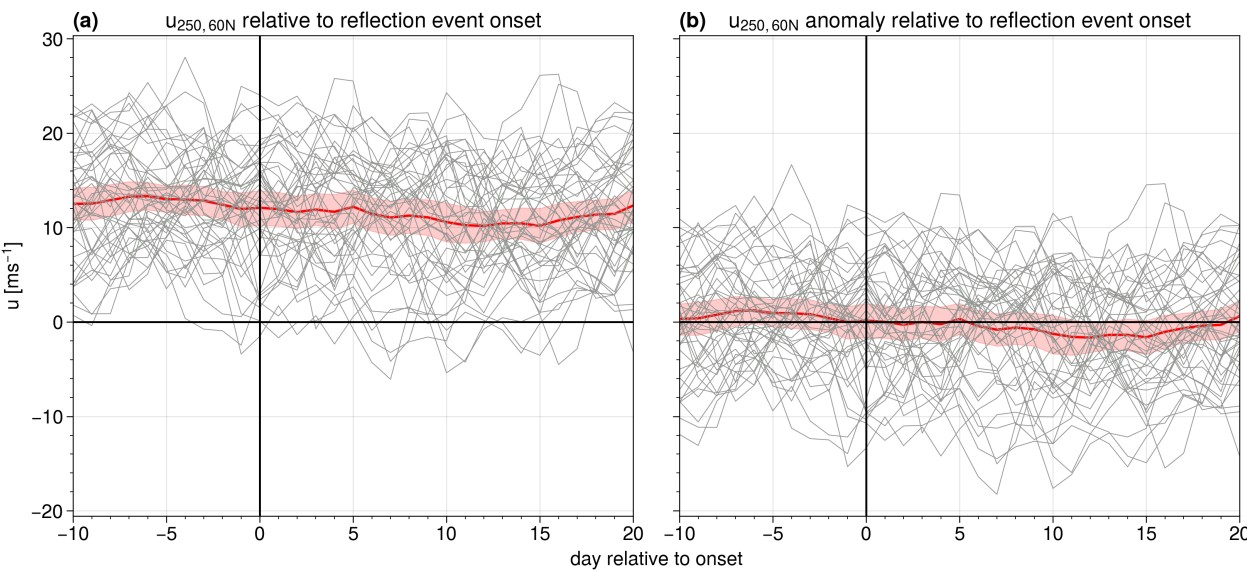

**Figure A18.** Evolution of (a) 250 hPa 60°N zonal mean zonal wind and (b) anomalies relative to seasonal cycle, centred around onset date of reflection events. The thick red line shows the average over all events, red shading indicates the 95% confidence interval on the mean, assessed with bootstrapping.

*Code and data availability.* The ERA5 data used to analyse the reflection events and compute the wave spectra are freely available from https://doi.org/10.24381/cds.bd0915c6 (Hersbach et al., 2024). The data set of space-time spectra of Rossby waves at tropospheric and

stratospheric levels is available at https://doi.org/10.57804/1dne-qk60 (Riboldi and Schutte, 2025). The data of North American weather regimes are available at: https://zenodo.org/records/8165165 (Lee et al., 2023b), and the data of the reflective index are available at: https://zenodo.org/records/10839643 (Lee et al., 2024). The code to compute the wave spectra is available from the authors upon request.

*Author contributions.* MS and GM designed the analysis. MS analyzed the data and prepared the figures with feedback from GM, AP and SL. MS drafted the first version of the manuscript and revised it in collaboration with all co-authors.

*Competing interests.* The authors declare that they have no conflict of interest.

*Acknowledgements.* The authors would like to thank Nili Harnik, one anonymous reviewer and the handling co-editor for their thorough and constructive comments, which improved the quality of the manuscript. We thank Jacopo Riboldi for computing the space-time spectra of Rossby waves and for helpful discussions. MS thanks Meriem Krouma, Iana Strigunova, Richard Leeding and Leonardo Olivetti for valuable input resulting from discussions about the results. MS and GM acknowledge funding from the European Research Council (ERC)

under the European Union's Horizon 2020 research and innovation program (grant agreement no. 948309, CENÆ project) and the Swedish Research Council Vetenskapsrådet through grant agreement no. 2022-06599. AP is funded from the Swiss National Science Foundation (SNF) Grant Number IZCOZ0_205461. The data handling was enabled by resources provided by the National Academic Infrastructure for Supercomputing in Sweden (NAISS), partially funded by the Swedish Research Council Vetenskapsrådet through grant agreement no. 2022-06725.

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
