# Peer review of "Dynamics of stratospheric wave reflection over the North Pacific"

_EGUsphere, 2024_

## Referee Comment (RC1)

Review of " Dynamics of stratospheric wave reflection over the North Pacific"  by Schutte et al.

Review by Nili Harnik

Recommendation: Major revisions

This paper examines the dynamics of North Pacific downward wave reflection events, in which the wave activity propagation is upward over the NW Pacific, zonal in the lower stratosphere and downward over E. Canada. These events have been studied in Messori et al (2022) which related them to N. American cold spells and a transition from a pacific Trough (PT) weather regime to an Alaskan Ridge (AKR) regime. In this paper the authors build upon this work and further examine the following things which Messori et al 2022 did not focus on:

The latitude-height-time evolution of meridional heat flux (alongside atmospheric temperature anomalies), Rossby wave activity and Rossby wave phase speeds based on a space-time spectral analysis, and surface wind extremes over Europe. The analysis was done with a focus on the onset and end of the Pacific reflection events. The main findings are a concentration of the heat-flux anomalies in the upper troposphere-lower stratosphere, with oppositely signed anomalies in the lower troposphere, and a clear signature of westward propagation of waves during the reflection event at all levels and for a range of zonal wavenumbers, and an increase in the occurrence of extreme wind events over Europe.

 In addition, the authors also compare the heat flux and Rossby wave evolution of these events to regional geopotential height events of similar structure and compared the onset/end of reflection events to anomalies related to PT and AKR regimes, respectively. For both the regional geopotential height anomalies and PT/AKR regimes they find the above heat flux and Rossby wave anomalies, with some differences.

The results point out to interesting dynamics and the analysis raises lots of interesting questions. At the same time, the paper can be improved significantly by better motivating the specific calculations and scientific questions, in a way that can they tie the results into a more coherent story. Following are some more specific comments and some thoughts on possible ways to improve the focus of the work.

In addition the methodology needs to be explained better, and in the case of the phase speed analysis, also better justified.

**Motivation:**

The authors discuss various ways to detect downward reflection, specifically, the detection of localized vs global events, and discuss the importance of these events for extreme event. They then do on to state on line 53, as underlying motivation to the paper, that the physical mechanisms linking stratospheric wave reflection to the subsequent tropospheric anomalies are not completely understood. As part of their aim to develop a deeper understanding of the evolution of the flow during these events, they pose 3 questions. While the questions are answered in the paper, it is not realy clear why these are chosen, beyond the fact that the analysis done happens to answer them. The paper can be greatly imporved if some of the main dynamical questions regarding these waves are more explicitly posed, and the reasons for studying heat fluxes, Rossby wave activity, and weather regimes is explained.

Question 1 specifically (*What is the vertical structure of the meridional eddy heat flux before, during and after reflection events?)* is not well motivated in my mind. Why did the authors choose to look at the vertical structure of heat fluxes and what do the results contribute to our understanding of these events?

The change in sign between lower and upper tropospheric heat fluxes, which accompanied the temporal change in sign, is interesting, but was not totally surprising to me because for the stratospheric case at least, it is related to the wave geopotential height ($\Phi$) peaking in height (at the downward reflection level), e.g. Fig. 4 of Harnik and Lindzen (2001). This peak in $\Phi$ wave amplitude leads to the temperature waves having a node at the level at which $\Phi$ peaks ($T' \sim d\Phi'/dz$). This will lead to a change in sign of v'T'.

It seems that in this case the heat flux signature comes from the medium scale waves having such a node - this can be examined easily, and can thus give us a better understanding of where the downward reflection signal is coming from. Specifically, how much of the evolution is tropospheric vs stratospheric dynamics?  Understanding this can lead to more understanding of the sources of their predictaiblity.

Another main result of the analysis is the westward phase speed of the waves during the downward reflection events. This again can be explained for the pure stratospheric wave reflection quite simply, but is more interesting in a way for the current case of the tropospheric waves propagating westward.

The following schematic shows how downard reflection natuarlly invovles westward phase propagation at lower levels (due to the vertical tilting of the phase lines of the wave). A westward phase speed on top and eastward on bottom is characteristic of a tilting of the climatologically westward tilted waves to a more vertical tilting

[Figure]

*Longitude-height wave 1 geopotential height phase lines. The + is a high and - a low. The waves at $t_0$ (in balck) tilt westward with height, and at $t_1$ the tilt vertically, due to donward reflection. They stay in place at 5mb so that below, the phase lines shift westward.*

I also thought of another option which can give a more tropospheric-induced donward reflection, as follows. The schematic below shows how a westward propagation of smaller scale tropospheric anomalies can lead to upward to the west - downward to the east wave propagation.

[Figure]

*Longitude height plots of geopotential height at two timesteps.*

*The stratospheric waves (black) do not change. Only the tropospheric wave field (green) evolves. The total field, stratosphere plus troposphere is in blue - the red arrows are the corresponding vertical EP fluxes which are up when the phase lines tilt westward with height and down when they tilt eastward.*

In this case the downward reflection occurrs due to the superposition of the waves and not due to a downward reflection of the wave 1.

Thinking of these two scenarios raises a few questions about the role of stratospheric wave 1 in these events - do these events involve both of the above scenarios or only one, or something else? See also my comments about the discussion section.

Following are some more major comments:

**Methods:**

The description of the analysis is partial and thus confusing:

In section 2.1 (line 77) you state that you deseasonalise all the variable except for the meridional wind. Why not the meridional wind? I assume you calculate v'T' from the full zonal anomalies before deseasonalising, and not using deseasonalized T' and non-deseasonalized v'..?

Section 2.3- the Space time Rossby wave spectra:

The deseasonalisation process is not explained clearly. Specifically, is ISP of equation 3 an anomaly or not. The right way is to calculate each of the quantities- S, ISP and Cp is using the full fields and only then calculate the anomaly from the climatological seasonal cycle. Maybe this is what is done but then please clarify in the text.

Another issue which is more fundamental is the ability of the spectral analysis to correctly resolve these relatively fast events, given that the spectra are calculated from 61 day time series (or shorted due to tapering). A possible way to check things is to compare just zonal spectra and then look at hovmoller diagrams of waves that dominate each time period to see if the results you get show similar changes of the wave phase speeds during the events.

**Results:**

Section 3.1.1

Figure 1 and its discussion:

As noted above, the change of sign of wave temperature anomalies, alongside no change of sign of the geopotential height, necessarily gives a change in sign of the heat fluxes, and this could be a result of the wave geopotential height structure. Have you looked at longitude-height plots of T' alongside v' or geopotential height? I think these would be very telling.

Also, regarding the negative v'T' anomaly which precedes the wave onset (fig 1a) and its corresponding positive v'T' - how much of this is due to the wave events occuring in bursts, meaning that preceding a wave event there is no heat flux, i.e. the upward flux starts after a period of "quiet", which is climatologically a negative heat flux anomaly.

Summary paragraph and specifically the statement starting on line 189: " *The oscillation of v'T' anomalies in the Siberian and Canadian domain originates from the westward propagation of negative v'T' anomalies"*

I assume this pertains to figure 2a. I find it hard to physically understand the propagation of v'T' anomalies, because it depends on two fields with a complex vertical structure (temperature). Specifically, a small change in the morphology of the plotted fields can change the appearance of the anomalies from a westward propagation to a simple standing oscillation, as in Figure A2a which shows temperature. Is this westward propagation meaningful in terms of a westward modal propagation? It could be resulting from a wave vertical structure change, rather than a modal propagation. Such questions can also motivate the space-time spectral analysis.

I think it will help to understand the phase propagation in terms of the wave geopotential height field evolution. The authors should consider switching the order of sections 3.1.1 and 3.1.2 - start with geopotential height anomaly evolution, and then examine how it manisfests in terms of v'T'.

Section 3.1.2:

Line 215 - figure 3d -

Repeating my comment from above: Is the ISP diagnostic able to capture the temporal evolution well given that it is based on spectra that are calculated from 61 day time series (or shorted due to tapering)? A possible way to check things is to compare just zonal spectra and then look at composites of the temporal changes of wave activity summed over the wavenumbers only.

Section 3.1.3

Here as well- the use of the spectral diagnostic for this specific problem needs more justification, specifically for figure 6 which shows a temporal evolution over a period which is of the same length as the date used for each of the phase speed calculations.

Here too, a zonal Fourier analysis can help - you can find the zonal wavenumbers which dominate each stage, and then look at the composited Hovmoller diagrams filtered only for the dominant zonal wavenumbers to see if the waves change their phase speed and how.

Also, anomalies in Cp shown in figures 5-6 do not necessarily indicate a westward propagation of the waves - a decrease in Cp can be due to specific zonal wavenumbers slowing down or propagating westward, but also to a weakening of the faster waves during a specific time period. How much do "background" synoptic waves affect the results of these figures?

Section 3.2:

The reference to the regimes and their transition in the context of the tropospheric waves westward propagation is interesting, and could be emphasized more in terms of the overall picture the authors have of what is going on. A more explicit motivation for this will improve the paper.

The lack of wave 1 dominance in the regime plots suggests maybe that wave 1 westward propagation is not crucial for the downward reflection signal to occur?

**Discussion:**

By better motivating the research questions, the discussion can be more focused. I am not sure how, but here is a possible flow of summarizing the results:

- The Pacific downward reflection events involve tropospheric medium scale waves, alongside planetary scale waves which dominate in the stratosphere.

- The tropospheric waves propagate westward during the events and also evolve downstream (is there also a clear westward propagation of the stratospheric wave 1?)

- This co-evolution leads to a specific evolution of heat fluxes.

- Given this picture, what is the role of the tropospheric wave evolution relative to the stratospheric wave evolution? Is it the downward reflection of stratospheric waves which causes the tropospheric waves to retrograde, or is it the westward propagation of the tropospheric waves which leads to a pattern of wave reflection? Is reflection from the stratosphere a contributing factor, or is it crucial for the occurrence of the wave reflection events defined by the regional 100mb v'T'?

Understanding the involvement of the stratosphere has implications for influences over Europe, and predictability.

Rereading the discussion, all of this is sort of mentioned but it does not come across as a coherent story, with one or two focusing questions.

Minor comments:

Lines 38-39- the reference to the wave reflection diagnostics of Harnik, Perlwitz, Shaw as

" *computationally intensive and may require data that are usually not a standard output of reanalysis products or climate models" is not quite right. The SVD time-lagged correlation*

*is not time-specific (though the space-time spectra are not as well) while the wave geometry calculation takes a few seconds on a simple laptop, but it requires a linear stationary wave model which most people don't have, and can give quite messy results which are hard to interpret especially for the troposphere.*

line 85- remove the word "daily" - it sounds like you remove the daily mean for each day..

line 86 - change "as indicated by asterisks" to "which we denote by an asterisk"

line 103-105- definition of the regional signal. Why did you choose 100hPa?

Have you tried projecting on other levels, specifically in the troposphere, or on a vertical-longitude (latitude averaged) or on the regional 3D structure in the upper troposphere-lower stratosphere?

line 108-109: How many of these onset high-correlation dates are followed by a highliy-correlated end date, meaning how many events do you find which correspond to the wave reflecting life cycle do these include?

line 225: the use of the term "barotropic" in this context is confusing, because barotropic dynamics strictly speaking do not involve heat fluxes. I think it is more exact to say that the tropospheric part of the wave is equivalent barotropic, which is consistent with the heat fluxes being concentrated in the upper troposphere and stratosphere rather than in mid troposphere (and with the heat fluxes changing sign in the vertical).

Figure 3d: Why is the x axis is flipped? A quick skim of the figure suggests there are less westward propagating waves during the event.. at the very least note in the caption that the axis is flipped..

Line 242: why is an enhancement of eastward propagating wave indicative of the presence of an accelerated stratospheric flow?

Line 244: what do you mean by *deceleration* of Rossby waves? Do you mean the increase in westward Rossby wave activity? If so it is confusing. The change of a stationary feature to westward propagation is actually an "acceleration"...

Line 269: why do you state that it is these wavenumbers? figure 5 seems to indicate wave 2 has strengthened just as much as 3,4 and larger wavenumbers

Figure 5: Is the y axis plotted in equal distances of 1/k?  So the slope is one over the frequency, or the period, right? These plots made me try and deduce the group speeds for

each period but its too complicated. Switching the x and y axes would make it easier but this is maybe not the focus of the paper, though the authors do indicate downstream development, which is essentially an eastward group propagation (alongside westward phase propagation).

Figure 7: The significant upward region in figure 7c is different from the downward reflection events. Can you explain this upward propagation and this difference?

 Lines 305-307: I think the question of the ability of the spectral analysis to represent the temporally concentrated (episodic) events, rather than spatially localized events is more of an issue, because the spatial localization will simply be manifest as a projection on a few zonal wavenumbers. It would be good to see a similar analysis focusing on time localization rather than spatial localization..

line 369: remove the word "accelerating"

Also on line 369 it is stated: "*A key result of our analysis is the accelerating westward propagation of Rossby waves during reflection events. This may be explained by the slowdown of the circulation due to Rossby wave absorption in the stratosphere.*"

I am not sure what the authors mean. I can think of two things:

1) A deceleration of the zonal mean zonal winds will cause the wave phase speed to slow down. However, the authors do not show a slowdown of the flow in the troposphere or lower stratosphere where the phase speed change is taking place (I think), even though it is an easy thing to check. Messori et al (2022) shows for these events, they are accompanied by a slowdown of zonal winds in the stratosphere, but the westward phase speed anomalies do not occur at these levels.

2) Another possible meaning has to do with the westward propagation being due to the tilting of the waves to the vertical. In this case- the slowdown of zonal mean zonal winds which Messori et al (2022) showed that occurs in the upper stratosphere, could indeed lead to downward reflection (as shown in Perlwitz and Harnik, 2003).

The authors should better state what they mean.

---

## Author Comment (AC1)

**Response to comment by Referee #1: Nili Harnik**

This paper examines the dynamics of North Pacific downward wave reflection events, in which the wave activity propagation is upward over the NW Pacific, zonal in the lower stratosphere and downward over E. Canada. These events have been studied in Messori et al (2022) which related them to N. American cold spells and a transition from a pacific Trough (PT) weather regime to an Alaskan Ridge (AKR) regime. In this paper the authors build upon this work and further examine the following things which Messori et al 2022 did not focus on:

The latitude-height-time evolution of meridional heat flux (alongside atmospheric temperature anomalies), Rossby wave activity and Rossby wave phase speeds based on a space-time spectral analysis, and surface wind extremes over Europe. The analysis was done with a focus on the onset and end of the Pacific reflection events. The main findings are a concentration of the heat-flux anomalies in the upper troposphere-lower stratosphere, with oppositely signed anomalies in the lower troposphere, and a clear signature of westward propagation of waves during the reflection event at all levels and for a range of zonal wavenumbers, and an increase in the occurrence of extreme wind events over Europe.

In addition, the authors also compare the heat flux and Rossby wave evolution of these events to regional geopotential height events of similar structure and compared the onset/end of reflection events to anomalies related to PT and AKR regimes, respectively. For both the regional geopotential height anomalies and PT/AKR regimes they find the above heat flux and Rossby wave anomalies, with some divergences.

The results point out to interesting dynamics and the analysis raises lots of interesting questions. At the same time, the paper can be improved significantly by better motivating the specific calculations and scientific questions, in a way that can they tie the results into a more coherent story. Following are some more specific comments and some thoughts on possible ways to improve the focus of the work.

In addition, the methodology needs to be explained better, and in the case of the phase speed analysis, also better justified.

*Thank you very much for your very thoughtful review of our manuscript. We greatly appreciate the time and effort put into providing such helpful comments. We agree with your suggestion that the paper can be improved by a clearer motivation of the scientific questions, and explaining or justifying parts of the methodology better. In the following, we will address each of your comments in more detail.*

The authors discuss various ways to detect downward reflection, specifically, the detection of localized vs global events, and discuss the importance of these events for extreme event. They then do on to state on line 53, as underlying motivation to the paper, that the physical mechanisms linking stratospheric wave reflection to the subsequent tropospheric anomalies are not completely understood. As part of their aim to develop a deeper understanding of the evolution of the flow during these events, they pose 3 questions. While the questions are answered in the paper, it is not really clear why these are chosen, beyond the fact that the analysis done happens to answer them. The paper can be greatly improved if some of the main dynamical questions regarding these waves are more explicitly posed, and the reasons for studying heat fluxes, Rossby wave activity, and weather regimes is explained.

Question 1 specifically (*What is the vertical structure of the meridional eddy heat flux before, during and after reflection events?*) is not well motivated in my mind. Why did the authors choose to look at

the vertical structure of heat fluxes and what do the results contribute to our understanding of these events?

The change in sign between lower and upper tropospheric heat fluxes, which accompanied the temporal change in sign, is interesting, but was not totally surprising to me because for the stratospheric case at least, it is related to the wave geopotential height (F) peaking in height (at the downward reflection level), e.g. Fig. 4 of Harnik and Lindzen (2001). This peak in F wave amplitude leads to the temperature waves having a node at the level at which F peaks (T'~dF'/dz). This will lead to a change in sign of v'T'.

*The first question, investigating the vertical structure of meridional heat flux, v'T', is motivated by the definition of reflection events: Using v'T' at 100 hPa to detect reflection events raised the question, how this signal is represented at other levels. With respect to the atmospheric behavior, we were also expecting a sign-change over Canada, since Messori et al. (2022) have shown that colder-than-usual air is advected southward at lower levels (positive v'T' anomalies).*

*Additionally, we were curious about the vertical extent of positive and negative v'T' anomalies and at which level the sign would change. The proportionality between v'T' and vertical wave propagation motivated us to investigate at which height we can observe a strong upward/downward signal. This contributes to understanding the vertical structure, i.e., at which height we can observe wave reflection, and how it connects to the advection of colder-than-usual air close to the surface.*

*Your explanation of sign-change makes sense and could complement our discussion. Your explanation, as also shown in Fig. 4 in Harnik and Lindzen (2001), applies in our context, as well, with the only difference that we find the reversed structure in the upper troposphere. This includes a local minimum of geopotential height over Canada, as temperature anomalies change vertically from colder than normal at lower levels to warmer than normal in the upper troposphere and lower stratosphere. In line with your expectation, we can observe the vertical temperature node over Canada at approximately 400 hPa, just above the minimum of a geopotential height anomaly at approximately 500 hPa (Fig. R1a).*

*We will add a short motivation to each question in lines 58-61, e.g., for question 1: "As reflection events are defined by v'T' anomalies at 100 hPa, we would like to investigate the vertical extent of these anomalies. Furthermore, Messori et al. (2022) have pointed out, that colder-than-usual air is advected southward at lower levels over Canada, which leads to opposite anomalies closer to the surface than at 100 hPa. This motivates to investigate additionally at which vertical level v'T' anomalies are changing sign." Furthermore, we will extend the discussion of geopotential height in the manuscript by the result about the vertical temperature node with respect to Figure R1.*

[Figure]

**Figure R1**. *Vertical cross-section of standardized geopotential height anomalies (shading) and temperature anomalies (lines in steps of 0.2) at 60°N four days before the end of reflection events (a) for all wavenumbers, (b) for wave-1 (c) for wave-2, (d) for wave-3 to 4, (e) for wave-5 to 6 and (f) for wave-7 to 8. The vertical lines mark longitudes of the Siberian (orange, dotted) and Canadian domains (purple, dashed).*

It seems that in this case the heat flux signature comes from the medium scale waves having such a node - this can be examined easily, and can thus give us a better understanding of where the downward reflection signal is coming from. Specifically, how much of the evolution is tropospheric vs stratospheric dynamics? Understanding this can lead to more understanding of the sources of their predictability.

*We share your impression that medium scale waves in the troposphere (~wave-2 to 4) play an important role in the context of reflection events over the North Pacific, which can also be seen in the space-time spectra (Fig. 5 in the manuscript). As shown in Fig. R1, the vertical temperature node in the troposphere over Canada is mostly linked to waves-2 to 4 in temperature and geopotential height anomalies four days before the end of reflection events.*

*In the stratosphere, wave-1 to 2 contribute most to the observed anomalies during reflection events with wave-2 having the opposite phase between stratosphere and troposphere. Thus, wave-1 is likely most important for the stratospheric setup during reflection events, while wave-2 is relevant in stratosphere and troposphere. Conversely, wave-3 to 4 contribute more to the overall anomalies in the troposphere than in the stratosphere.*

*We will address your comment by mentioning the relevance of wave-3 to 4 in the troposphere in discussion about Rossby wave spectra (e.g., line 270), e.g., "The vertical structure of geopotential anomalies confirms the importance of waves-3 and 4 in the troposphere and wave-1 in the stratosphere (Fig. R1).", and by adding Fig. R1 to the appendix.*

Another main result of the analysis is the westward phase speed of the waves during the downward reflection events. This again can be explained for the pure stratospheric wave reflection quite simply, but is more interesting in a way for the current case of the tropospheric waves propagating westward.

The following schematic shows how downward reflection naturally involves westward phase propagation at lower levels (due to the vertical tilting of the phase lines of the wave). A westward phase speed on top and eastward on bottom is characteristic of a tilting of the climatologically westward tilted waves to a more vertical tilting.

I also thought of another option which can give a more tropospheric-induced downward reflection, as follows. The schematic below shows how a westward propagation of smaller scale tropospheric anomalies can lead to upward to the west - downward to the east wave propagation.

In this case the downward reflection occurs due to the superposition of the waves and not due to a downward reflection of the wave 1 (*see drawings in original answer*).

Thinking of these two scenarios raises a few questions about the role of stratospheric wave 1 in these events - do these events involve both of the above scenarios or only one, or something else? See also my comments about the discussion section.

*Thank you for elaborating on these two mechanisms to potentially explain the westward phase speed. Both could be relevant for the observed westward-propagation of Rossby waves. A discussion on this could benefit our manuscript.*

*The first mechanism (phase lines tilting vertically during reflection) seems like a very reasonable explanation for the observed enhancement of westward-propagating Rossby waves in the lower and middle stratosphere. In line with you first schematic, we see enhanced phase speed in the middle stratosphere (~10 hPa), and positive anomalies with a lower amplitude in lower stratosphere (~100 hPa) during reflection events (Fig. R3a). We also observe the vertical alignment of the positive geopotential height anomaly during reflection events, supporting your suggestion (Fig. R2).*

*The second mechanism (constant stratospheric field, evolving tropospheric field) likely also plays a role, as synoptic-scale disturbances in the troposphere are replaced by their opposite anomaly (e.g., Fig. 4f to i in the manuscript). During the course of reflection events this would result in the proposed evolution, as in your second schematic. At the same time, larger-scale anomalies have also a greater influence on the upper troposphere (see later comments connected to Fig. R4) and the stratospheric field is not constant during reflection events (e.g., westward-propagation of positive anomaly in Fig. A3 in the manuscript). Thus, the first mechanism likely plays a larger role in the stratospheric evolution, while the second scenario could still be of importance for the tropospheric evolution. It might also be less clear to assess the role of the second mechanism due to a larger variability in the troposphere.*

*We will mention that the westward propagation of anomalies in the lower stratosphere can be understood by the mechanism of phase lines aligning vertically in the discussion of geopotential height anomalies.*

[Figure]

**Figure R2.** *Time series of standardized geopotential height anomalies (shading) and temperature anomalies (lines in steps of 0.2) at 60°N, ranging from (a) 16 days before the end of reflection events to (f) four days after the end of reflection events. The vertical lines mark longitudes of the Siberian (orange, dotted) and Canadian domains (purple, dashed).*

**Methods**:

The description of the analysis is partial and thus confusing: In section 2.1 (line 77) you state that you deseasonalize all the variable except for the meridional wind. Why not the meridional wind? I assume you calculate v'T' from the full zonal anomalies before deseasonalizing, and not using deseasonalized T' and non-deseasonalized v'..?

*Thank you for highlighting this unclarity. We calculate v'T' from the full zonal anomalies and deseasonalize only afterwards, as you stated. We will refine this description the manuscript by modifying the data description in lines 77-78: "Unless stated differently, we deseasonalized each field shown in the following analysis by subtracting the daily seasonal cycle, computed…", and by adding a sentence to end of paragraph: "The meridional eddy heat flux, WAF and space-time Rossby wave spectra are computed from the full fields and deseasonalized afterwards."*

Section 2.3- the Space time Rossby wave spectra: The deseasonalization process is not explained clearly. Specifically, is ISP of equation 3 an anomaly or not. The right way is to calculate each of the

quantities- S, ISP and Cp is using the full fields and only then calculate the anomaly from the climatological seasonal cycle. Maybe this is what is done but then please clarify in the text.

*We have computed the Space-time Rossby wave spectra on the full fields of meridional wind and deseasonalized the spectra afterwards. Thank you for highlighting this aspect, we will improve the explanation here.*

*Regarding the spectrally derived metrics, ISP and $\bar{C}p$, we computed them from the full spectra and from the deseasonalized spectra separately, but did not deseasonalize them. For ISP, this makes no big difference, as this is just the sum of spectral power, and hence the sum of anomalies is approximately the same. However, the mean $\bar{C}p$, that is weighted with the phase speed of each harmonic individually (Eq. 4), shows a slightly different behavior, depending on if one computes it from the deseasonalized spectra or if one computes $\bar{C}p$ from the full spectra and deseasonalizes the mean phase speed $\bar{C}p$ afterwards (cf. Fig. R3a to the manuscript's Fig. 6). The general behavior of a drop of mean phase speed $\bar{C}p$ towards the end of reflection events is still very pronounced, albeit with stronger positive anomalies during the event and weaker negative anomalies around the end. Thank you for spotting this inconsistency, we will update the manuscript on that aspect in line 140: "Both integral metrics (ISP and $\bar{C}p$) are computed from the full space-time spectra and deseasonalized afterwards.", and replace Fig. 6 in the manuscript by Fig. R3 and update the discussion of Fig. 6 in lines 251-271 accordingly.*

[Figure]

**Figure R3.** *Vertical cross-section of standardized anomalies of (a) phase speed, (b) ISP, (c) ISPwest and (d) ISPeast derived from spectral power SV'V' between 850 hPa and 10 hPa centered around the end date of*

*reflection events. Horizontal hatching marks significant negative anomalies, cross-hatching significant positive anomalies. ISP and phase speed anomalies were obtained by deseasonalizing their full time series and not from deseasonalized space-time spectra, as in Fig. 6 in the manuscript.*

Another issue which is more fundamental is the ability of the spectral analysis to correctly resolve these relatively fast events, given that the spectra are calculated from 61-day time series (or shorted due to tapering). A possible way to check things is to compare just zonal spectra and then look at Hovmöller diagrams of waves that dominate each time period to see if the results you get show similar changes of the wave phase speeds during the events.

*We acknowledge that the relatively large time window of 61 days can potentially smooth out the signal of events with a short duration. However, reflection events still happen on weekly time scales (median duration of 19 days, as the green line in the Hovmöller plots, e.g., Fig. 3 in the manuscript), which is long enough for space-time spectral analysis with the underlying 61-day time window to pick up on that signal. Due to the additional tapering of the first and last 12 days, only 37 days count fully into the space-time spectra of each day (section 2.3 in the manuscript). Furthermore, variability before and after events likely averages out other signals, so that only changes connected to reflection events become apparent in the composite space-time spectra.*

*The motivation for this time window was to resolve also slowly propagating Rossby waves with low absolute phase speeds, even though standing waves can still not be resolved. For example, in the case of your first scenario proposed earlier (phase lines tilting vertically during reflection), low phase speeds become important to describe the wave behavior in the middle stratosphere. Additionally, earlier results on extreme states of the stratospheric polar vortex suggest that sudden changes can be captured by space-time spectra computed this way (Schutte et al., 2024). See also the later comment on Section 3.1.2: Line 215 - figure 3d.*

*Following your suggestion, Hovmöller diagrams of specific wavenumbers reveal a similar behavior, as the space-time spectra. For example, at 100 hPa, we observe mainly large-scale anomalies propagating westward (Fig. R4). Comparing the wave propagation during median onset (day -19) with end (day 0) shows even some of the changes explored with help of space-time Rossby wave spectra, e.g., a change of slowly eastward-propagating waves-1 and 2 during the median onset to a more rapid westward-propagation during reflection events and around the end (Fig. R4a).*

*This behavior can even be seen in a similar way in the Hovmöller diagrams of geopotential height without zonal wave decomposition (Fig. 3a in the manuscript). In the lower stratosphere and upper troposphere, large-scale waves dominate and exhibit a tendency to propagate westward around the end of reflection events.*

*We will address your comment by adding in line 128: "Despite the time filtering applied during the computation of Rossby wave space-time spectra, we note that this metric can also capture daily to weekly changes in Rossby wave behavior (Fig. R4).", and we will add Fig. R4 to the appendix.*

[Figure]

**Figure R4.** *Hovmöller diagram of standardized anomalies of geopotential height averaged between 45N - 75N centered around the end date of reflection events at 100 hPa for (a) wave-1 to 2, (b) wave-3 to 4 and (c) wave-5 to 6 in shading. Horizontal hatching marks significant negative anomalies and cross-hatching significant positive anomalies. The continuous horizontal green line shows the median onset time of reflection events. The vertical lines mark longitudes of the Siberian (orange, dotted) and Canadian domains (purple, dashed).*

**Results**:

Section 3.1.1 Figure 1 and its discussion: As noted above, the change of sign of wave temperature anomalies, alongside no change of sign of the geopotential height, necessarily gives a change in sign of the heat fluxes, and this could be a result of the wave geopotential height structure. Have you looked at longitude-height plots of T' alongside v' or geopotential height? I think these would be very telling.

*Thank you for the idea to investigate longitude-height plots of T', v' or geopotential height. We have looked at longitude height plots of geopotential height and temperature with respect to your earlier comments (see answers referring to Fig. R1).*

Also, regarding the negative v'T' anomaly which precedes the wave onset (fig 1a) and its corresponding positive v'T' - how much of this is due to the wave events occuring in bursts, meaning that preceding a wave event there is no heat flux, i.e. the upward flux starts after a period of "quiet", which is climatologically a negative heat flux anomaly

*Thank you for your suggestion about seeing wave reflection events as bursts in heat flux. This 'quiet' period before could be an additional hint at some preconditioning needed for reflection events. We*

*observe a more uniform, positive meridional eddy heat flux before and after the burst connected to reflection events (Fig. R5). However, v'T' is not completely suppressed, but remains rather close to climatology during these 'quiet' periods.*

*In line with your comment, we will add in line 190: "Furthermore, the opposite v'T' anomalies before the onset and after the end of reflection events relate to times with more uniform, weakly positive meridional eddy heat fluxes over both regions (Fig. R5).", and add Fig. R5 to the appendix.*

[Figure]

*__Figure R5.__ Meridional eddy heat flux v'T' at 100 hPa over East Siberia (red, dashed) and Canada (blue, dotted), (a) centered around the onset date, (b) centered around the end date of reflection events. The thin horizontal lines show the DJFM mean in the respective region, shading indicates the 95% confidence interval on the mean, assessed with bootstrapping. The vertical green line in (a) shows the median end, and in (b) the median onset of reflection events.*

Summary paragraph and specifically the statement starting on line 189: " *The oscillation of v'T' anomalies in the Siberian and Canadian domain originates from the westward propagation of negative v'T' anomalies*"

I assume this pertains to figure 2a. I find it hard to physically understand the propagation of v'T' anomalies, because it depends on two fields with a complex vertical structure (temperature). Specifically, a small change in the morphology of the plotted fields can change the appearance of the anomalies from a westward propagation to a simple standing oscillation, as in Figure A2a which shows temperature. Is this westward propagation meaningful in terms of a westward modal propagation? It could be resulting from a wave vertical structure change, rather than a modal propagation. Such questions can also motivate the space-time spectral analysis.

I think it will help to understand the phase propagation in terms of the wave geopotential height field evolution. The authors should consider switching the order of sections 3.1.1 and 3.1.2 - start with geopotential height anomaly evolution, and then examine how it manifests in terms of v'T'.

*As you assumed, the cited statement refers mostly to the lower stratosphere. We will clarify this in the manuscript. Furthermore, we share your impression that the propagation of v'T' anomalies is not so clear to understand or to interpret. In line with the westward propagation of geopotential height anomalies we understand the resulting evolution of v'T' anomalies being mostly driven by a westward propagation of v' (see also Fig. R6). Thank you for the hint that this can be an additional motivation to motivate space-time spectral analysis.*

*We understand the logic of elaborating on the evolution of geopotential height anomalies before discussing v'T' anomalies. Our motivation for the current order of sections is to begin with the investigation of v'T' anomalies, as they are used to define reflection events, and explain this behavior in the following section through the evolution of geopotential height anomalies. Thus, we would like to keep the current order of sections 3.1.1 and 3.1.2.*

*With respect to your comment, we will modify sentence in line 189 to: "The oscillation of v 'T ' anomalies in the stratosphere over the Siberian and Canadian domain ...". Further, we will add in line 190: "As this evolution of v'T' anomalies is mostly linked to the meridional wind (Fig. R6), this motivates to investigate the corresponding geopotential height anomalies and Rossby wave spectra of meridional wind, focusing also on the emerging westward propagation in the following sections.", and add figure R6 to the appendix.*

[Figure]

**Figure R6.** *Hovmöller diagram of standardized anomalies of composite meridional wind v' averaged between 45N - 75N centered around the end date of reflection events at (a) 100 hPa, (b) 250 hPa and (c) 850 hPa. Horizontal hatching marks significant negative anomalies and cross-hatching significant positive anomalies. Continuous red and dashed blue contours in (c) show standardized positive and negative 2m air temperature anomalies, respectively, averaged between 40N - 55N (1 std ≈ 3.8 K). The contours start at ±0.5std with steps of 0.3std. The continuous horizontal green line shows the median onset day of reflection events. The vertical lines mark longitudes of the Siberian (orange, dotted) and Canadian domains (purple, dashed)*

Section 3.1.2: Line 215 - figure 3d –

Repeating my comment from above: Is the ISP diagnostic able to capture the temporal evolution well given that it is based on spectra that are calculated from 61-day time series (or shorted due to tapering)? A possible way to check things is to compare just zonal spectra and then look at composites of the temporal changes of wave activity summed over the wavenumbers only.

*The ISP diagnostic is able to capture the temporal evolution of reflection events, in line with our answers to your earlier comment about Rossby wave spectra capturing reflection events. Furthermore, ISP can also capture other stratospheric events, such as stratospheric warmings or strong polar vortex events (Fig. 8 in Schutte et al, 2024), so we are convinced that it can capture the temporal evolution of reflection events, as well.*

*Since Fig. 3d in the manuscript shows only ISP for westward-propagating waves, detecting Rossby wave behavior with a specific propagation direction requires some sort of decomposition in time, which we obtained from the time/frequency component of space-time spectral analysis. However, comparing ISP for all wavenumbers with the sum of spectral power obtained from zonal decomposition reveals a very similar behavior, confirming that space-time spectra are able to capture the relevant changes of Rossby wave behavior during reflection events (e.g., compare zonal spectral power at 250 hPa in Fig. R7 to ISP in Fig. R2b). One potential reason for differences is the smoothing*

*in the frequency direction during the computation of space-time spectra, as described in section 2.3. in the manuscript.*

*We will adjust the manuscript in line with the earlier with respect to figure R4.*

[Figure]

[Figure]

*Figure R7. Standardized anomalies of zonal spectral power of meridional wind at 50 hPa, obtained from the sum over wavenumbers 1 to 15. The thick orange line shows the average over all events, orange shading indicates the 95% confidence interval of the mean, assessed with bootstrapping.*

Section 3.1.3

Here as well- the use of the spectral diagnostic for this specific problem needs more justification, specifically for figure 6 which shows a temporal evolution over a period which is of the same length as the date used for each of the phase speed calculations. Here too, a zonal Fourier analysis can help - you can find the zonal wavenumbers which dominate each stage, and then look at the composited Hovmöller diagrams filtered only for the dominant zonal wavenumbers to see if the waves change their phase speed and how.

*Thank you for your suggestion. As highlighted with respect to earlier comments, we believe that the method of space-time spectral analysis can be applied to analyze this type of reflection events. The decrease in phase speed during the end of reflection events, as observed in Fig. R3, is also discernible in Hovmöller diagrams for large-scale waves (e.g., at 100 hPa in Fig. R4).*

*We will adjust the manuscript in line with the earlier with respect to figure R4.*

Also, anomalies in Cp shown in figures 5-6 do not necessarily indicate a westward propagation of the waves - a decrease in Cp can be due to specific zonal wavenumbers slowing down or propagating westward, but also to a weakening of the faster waves during a specific time period. How much do "background" synoptic waves affect the results of these figures?

*Thank you for highlighting this caveat of the averaged phase speed $\bar{C}p$. As it is weighted by the spectral power of each harmonic, one cannot obtain the detailed change in Rossby wave behavior, but rather some dominant effect. Since synoptic-scale waves dominate in the troposphere and large-scale waves dominate in the stratosphere, the different scales affect the phase speed depending on the specific level. Differences in phase speed can also be driven by large-scale waves affecting synoptic-scale waves which propagate on the large-wave background. Furthermore, the shift of wave-3 to 4 from an eastward to a westward propagation is likely relevant for the observed decrease of phase speed (Fig. 5 and 6 a in the manuscript).*

Section 3.2:

The reference to the regimes and their transition in the context of the tropospheric waves' westward propagation is interesting, and could be emphasized more in terms of the overall picture the authors have of what is going on. A more explicit motivation for this will improve the paper.

The lack of wave 1 dominance in the regime plots suggests maybe that wave 1 westward propagation is not crucial for the downward reflection signal to occur?

*Thank you for the suggestion to emphasize the transition of regimes more with respect to the overall picture. Since Messori et al. (2022) had suggested that reflection events are dominated by the transition from Pacific trough (PT) to Alaskan Ridge (AKR), we were curious if the emerging AKR connects also to the westward propagation of the positive anomaly during reflection events. This would confirm earlier research linking the end of blocking over Alaska to a westward-propagation of geopotential height anomalies (Carrera et al., 2004). In line with the earlier comment about the two different mechanisms linking reflection events to a decrease of phase speed during reflection events, the comparison between reflection events and PT-AKR weather regime transitions can even be used to assess the role of tropospheric anomalies during reflection events.*

*However, we would like to emphasize that this regime transition can also happen without the stratospheric counterpart. Thus, only the difference between PT and AKR events happening during reflection events shows an enhancement in westward propagating wave-1 (17 of 107 PT and 23 of 91 AKR events during reflection), while those regimes occurring outside of reflection events (90 of 107 PT and 68 of 91 AKR) don't show this signal (Fig. R8 and R9). This makes us conclude that the wave-1 signal is a feature that differentiates reflection events from the normal PT-AKR regime transition, which we emphasize in the manuscript (e.g., line 300-302).*

*We will address you comment by adding in line 297: "This includes the replacement of the negative geopotential height anomaly around the onset by a ridge before the end of reflection events, which subsequently propagates westward and facilitates the formation of a trough downstream over North America.", and we will add figures R8 and R9 to the appendix. Additionally, we will add in line 302 "... differentiating PT-AKR transitions from reflection events. Separating between PT-AKR transitions happening during reflection events and those outside of reflection events, also highlights the relevance of wave-1 for reflection events (see R8 and R9)."*

[Figure]

**Figure R8.** *Standardized anomalies of spectral power SV'V' of each harmonic at 100 hPa during reflection events with (a) PT, (b) AKR (c) difference between PT and AKR weather regimes in shading. Sub-panels (d) - (e) show the same, but for 250 hPa. Black contours show the DJFM mean for all years (100 hPa: steps of 0.05 m2 s −2 Δc−1 from 0.05 m2 s −2 Δc−1 to 0.35 m2 s −2 Δc−1; 250 hPa: steps of 0.1 m2 s −2 Δc−1 from 0.1 m2 s −2 Δc−1 to 0.7 m2 s −2 Δc−1). Horizontal hatching indicates significant negative anomalies, cross-hatching indicates significant positive anomalies.*

[Figure]

**Figure R9.** *Same as Fig. R6, but for PT and AKR regimes outside of reflection events.*

**Discussion**:

By better motivating the research questions, the discussion can be more focused. I am not sure how, but here is a possible flow of summarizing the results:

- The Pacific downward reflection events involve tropospheric medium scale waves, alongside planetary scale waves which dominate in the stratosphere.

- The tropospheric waves propagate westward during the events and also evolve downstream (is there also a clear westward propagation of the stratospheric wave 1?)

- This co-evolution leads to a specific evolution of heat fluxes.

- Given this picture, what is the role of the tropospheric wave evolution relative to the stratospheric wave evolution? Is it the downward reflection of stratospheric waves which causes the tropospheric waves to retrograde, or is it the westward propagation of the tropospheric waves which leads to a pattern of wave reflection? Is reflection from the stratosphere a contributing factor, or is it crucial for the occurrence of the wave reflection events defined by the regional 100mb v'T"?

Understanding the involvement of the stratosphere has implications for influences over Europe, and predictability. Rereading the discussion, all of this is sort of mentioned but it does not come across as a coherent story, with one or two focusing questions.

*Thank you for providing this suggestion of discussion flow. We will edit this chapter with respect to your ideas. Furthermore, we will extend the discussion about the stratospheric role with respect to your two suggested mechanisms, highlighting the question for future investigation about the role of tropospheric and stratospheric dynamics during reflection events.*

**Minor comments:**

Lines 38-39- the reference to the wave reflection diagnostics of Harnik, Perlwitz, Shaw as *"computationally intensive and may require data that are usually not a standard output of reanalysis products or climate models"* is not quite right. The SVD time-lagged correlation is not time-specific (though the space-time spectra are not as well) while the wave geometry calculation takes a few seconds on a simple laptop, but it requires a linear stationary wave model which most people don't have, and can give quite messy results which are hard to interpret especially for the troposphere.

*Thank you for highlighting this inconsistency. We were referring here to the computation of wave activity fluxes, which may be computationally intensive if one is interested in a larger region and several levels. To avoid this misunderstanding, we will adapt the introduction with respect to your input.*

*With respect to your comment, we will edit the sentence to "...which are insightful but may require specific models or data that are usually not a standard output of reanalyses or climate models."*

line 85- remove the word "daily" - it sounds like you remove the daily mean for each day.

*Thank you, we will clarify the description by replacing "... the daily mean" with "the mean for each calendar day" in line 85.*

line 86 - change "as indicated by asterisks" to "which we denote by an asterisk"

*Good suggestion, thank you. We will change the wording in line 86 to "which we denote by an asterisk".*

line 103-105- definition of the regional signal. Why did you choose 100hPa? Have you tried projecting on other levels, specifically in the troposphere, or on a vertical-longitude (latitude averaged) or on the regional 3D structure in the upper troposphere-lower stratosphere?

*We chose 100 hPa to be consistent with the definition of reflection events using v'T' anomalies at 100 hPa. Nevertheless, we performed this type of analysis also for other levels, e.g., 10 hPa and 250 hPa simultaneously instead of 100 hPa. This led to qualitatively similar results with lower amplitudes than for the chosen 100 hPa, which is likely due to conditioning on a similar pattern at two levels simultaneously (Fig. R10).*

*With respect to your comment, we will mention in line 105: "Projecting this method on other levels results in qualitatively similar results (not shown)."*

[Figure]

**z anomaly, v'T' anomaly and temperature anomaly during wave reflection events**

***Figure R10**. Geopotential height anomalies in shading and v'T' anomalies in green contours (first column) during regionalized onset, (second column) 7 days after regionalized onset, (third column) at regionalized end, and (fourth column) difference between regionalized onset and end events. Horizontal hatching marks significant negative geopotential height anomalies and cross-hatching significant positive anomalies. (last column) Temperature anomalies in shading, geopotential height field in purple and v'T' anomalies in green 7 days after regionalized onset. Regionalized onset and end events were computed with respect to high spatial correlation at 10 hPa and 250 hPa simultaneously.*

line 108-109: How many of these onset high-correlation dates are followed by a highly-correlated end date, meaning how many events do you find which correspond to the wave reflecting life cycle do these include?

*Thank you for that question. With respect to a 69-day time window (longest duration of reflection events), we found 39 of 54 instances with regionalized onset events being followed by regionalized end events. As the median duration is with 30 days longer than that of reflection events (19 days), we checked also with a shorter maximum duration of 58 days (median duration of reflection events + 3 standard deviations). In this case, 31 of the 54 regionalized onset events are followed by an end, with a median duration of 24 days.*

*We will address your comment by mentioning in line 109: "With respect to a 69-day time window (longest duration of reflection events), we found 39 of 54 instances with regionalized onset events being followed by regionalized end events."*

line 225: the use of the term "barotropic" in this context is confusing, because barotropic dynamics strictly speaking do not involve heat fluxes. I think it is more exact to say that the tropospheric part of the wave is equivalent barotropic, which is consistent with the heat fluxes being concentrated in the upper troposphere and stratosphere rather than in mid troposphere (and with the heat fluxes changing sign in the vertical).

*We agree with your objection against the term 'barotropic' and will reformulate it to "equivalent barotropic", also in line 277.*

Figure 3d: Why is the x axis is flipped? A quick skim of the figure suggests there are less westward propagating waves during the event. At the very least note in the caption that the axis is flipped.

*Thank you very much for pointing out the potential confusion with the flipped x-axis in Figure 3d. As ISP of westward propagating waves is shown there, the motivation was that positive anomalies, would go to the left, indicating enhanced activity of Rossby waves moving to the west, which is the same direction as the anomalies. Nevertheless, we agree that one should at least mention the flipped x-axis in the caption.*

*We will add to caption of Fig. 3 in the manuscript: "The x-axis of panel (d) is flipped, so that enhanced activity of westward-propagating Rossby waves shows anomalies to the left".*

Line 242: why is an enhancement of eastward propagating wave indicative of the presence of an accelerated stratospheric flow?

*We followed here the rule of thumb that was discussed in Schutte et al. (2024), that the background flow contribution equally affects to first order all wavenumbers by shifting the spectra towards lower/higher phase speeds than climatology. Of course, this is a simplification, but showed agreement in space-time spectra during states of a very strong or weak stratospheric polar vortex. Thus, the presence of both suppressed westward propagating waves and enhanced propagating waves (Fig. 5 a and d in the manuscript) motivated this statement.*

*We will address your comment by adding in line 242: "... [presence of an accelerated stratospheric flow] under the assumption that the background flow equally affects all wavenumbers by shifting the spectra towards lower/higher phase speeds than climatology."*

Line 244: what do you mean by *deceleration* of Rossby waves? Do you mean the increase in westward Rossby wave activity? If so it is confusing. The change of a stationary feature to westward propagation is actually an "acceleration"...

*Thank you for highlighting this formulation. In line with the previous comment we meant here the deceleration of Rossby wave propagation. With respect to your comment, we will change "Rossby waves" to "Rossby wave propagation" in line 244.*

Line 269: why do you state that it is these wavenumbers? figure 5 seems to indicate wave 2 has strengthened just as much as 3,4 and larger wavenumbers

*You are absolutely right with respect to the standardized anomalies in the space-time spectra. However, considering the range of climatologically relevant harmonics, indicated by the black contours, only these specific waves seemed to have also some energetically meaningful contribution. While westward-propagating wave-2 showed some indication of enhancement, this signal is weaker in the range of climatologically relevant harmonics than for the other waves and, moreover, non-significant. Thus, the role of wave-2 seems to be less clear during reflection events. The same applies for larger wavenumbers within climatologically relevant harmonics (within the black lines in Fig. 5 c and f in the manuscript).*

Figure 5: Is the y axis plotted in equal distances of 1/k? So the slope is one over the frequency, or the period, right? These plots made me try and deduce the group speeds for each period but its too complicated. Switching the x and y axes would make it easier but this is maybe not the focus of the paper, though the authors do indicate downstream development, which is essentially an eastward group propagation (alongside westward phase propagation).

*The y-axis in the space-time spectra is plotted in distances of ln(k) in line with earlier visualizations of space-time spectra (e.g., Schutte et al., 2024; Riboldi et al., 2022). For more details, we would like to refer to these and the supplement of Riboldi et al. (2022). For our discussion about west-and eastward propagating Rossby waves we would like to keep the phase speed on the x-axis.*

Figure 7: The significant upward region in figure 7c is different from the downward reflection events. Can you explain this upward propagation and this diverence?

*As the positive geopotential height anomaly is more pronounced at 850 hPa during the weather regime shift than during reflection events, we suspect that the stronger upward wave activity flux is connected to the higher amplitude. Furthermore, the level of 850 hPa is close enough to the surface that topography, i.e., the Rocky Mountains could play a role, as well.*

*With respect to RC2's comments, we will use the wave activity flux proposed by Takaya and Nakamura (2001) instead of the one suggested by Plumb (1985). In that case, the WAF signal of the PT-AKR transition resembles that of reflection events more closely.*

Lines 305-307: I think the question of the ability of the spectral analysis to represent the temporally concentrated (episodic) events, rather than spatially localized events is more of an issue, because the spatial localization will simply be manifest as a projection on a few zonal wavenumbers. It would be good to see a similar analysis focusing on time localization rather than spatial localization.

*As elaborated earlier, the time window would only result in a smoothing of the signal, thus making it harder to detect changes in Rossby wave behavior. With respect to our previous replies on that topic we would like to point out that we can detect changes in Rossby wave spectra during shorter time scales than the averaging window, including reflection events. Nevertheless, the question about temporally representation could be elaborated in the discussion section.*

*In line with the second reviewer's comment on this part, we will skip section 3.3 and discuss both spatial and temporal resolution in section 4.*

line 369: remove the word "accelerating"

*Thank you, we will remove "accelerating" in line 369.*

Also on line 369 it is stated: "A key result of our analysis is the accelerating westward propagation of Rossby waves during reflection events. This may be explained by the slowdown of the circulation due to Rossby wave absorption in the stratosphere." I am not sure what the authors mean. I can think of two things:

1) A deceleration of the zonal mean zonal winds will cause the wave phase speed to slow down. However, the authors do not show a slowdown of the flow in the troposphere or lower stratosphere where the phase speed change is taking place (I think), even though it is an easy thing to check. Messori et al (2022) shows for these events, they are accompanied by a slowdown of zonal winds in the stratosphere, but the westward phase speed anomalies do not occur at these levels.

2) Another possible meaning has to do with the westward propagation being due to the tilting of the waves to the vertical. In this case- the slowdown of zonal mean zonal winds which Messori et al (2022) showed that occurs in the upper stratosphere, could indeed lead to downward reflection (as shown in Perlwitz and Harnik, 2003).

The authors should better state what they mean.

*Thank you for highlighting the potential for misunderstanding in this statement. While the westward propagation could be associated with the vertical tilting of the waves, our intention was to convey that the increased westward propagation of Rossby waves could be linked to the deceleration of zonal mean zonal winds. In line with the slowdown of zonal mean zonal winds at 10 hPa shown by Messori et al. (2022), we observe this tendency also at lower levels, albeit with much lower amplitudes making it hardly noticeable (Fig. R11). This could also be linked to the higher variability in zonal winds across the entire hemisphere at 250 hPa compared to 10 hPa, as reflection events have a more regionalized impact at tropospheric levels (e.g., Fig. A3 k to x in the manusript).*

*With respect to your comment, we will replace the second sentence in line 369-370 with: "This may be linked to waves tilting to the vertical and subsequent downward reflection as shown in Perlwitz and Harnik (2003). Another possible mechanism is the deceleration of zonal mean zonal winds (Fig. R10), resulting from Rossby wave absorption in the stratosphere." and add Fig. R11 to the appendix.*

[Figure]

*Figure R11. Evolution of (a) 250 hPa 60°N zonal mean zonal wind and (b) anomalies relative to seasonal cycle, centered around onset date of reflection events. The thick red line shows the average over all events, red shading indicates the 95% confidence interval on the mean, assessed with bootstrapping.*

**References**

Carrera, M. L., Higgins, R. W., and Kousky, V. E.: Downstream Weather Impacts Associated with Atmospheric Blocking over the Northeast Pacific, J. Climate, 17, 4823 – 4839, https://doi.org/10.1175/JCLI-3237.1, 2004.

Messori, G., Kretschmer, M., Lee, S. H., and Wendt, V.: Stratospheric downward wave reflection events modulate North American weather regimes and cold spells, Weather Clim. Dynam., 3, 1215–1236, https://doi.org/10.5194/wcd-3-1215-2022, 2022.

Plumb, R. A.: On the Three-Dimensional Propagation of Stationary Waves, J. Atmos. Sci., 42, 217 – 229, https://doi.org/10.1175/1520-0469(1985)042<0217:OTTDPO>2.0.CO;2, 1985.

Riboldi, J., Rousi, E., D'Andrea, F., Rivière, G., and Lott, F.: Circumglobal Rossby wave patterns during boreal winter highlighted by space–time spectral analysis, Weather Clim. Dynam., 3, 449–469, https://doi.org/10.5194/wcd-3-449-2022, 2022.

Schutte, M. K., Domeisen, D. I. V., and Riboldi, J.: Opposite spectral properties of Rossby waves during weak and strong stratospheric polar vortex events, Weather Clim. Dynam., 5, 733–752, https://doi.org/10.5194/wcd-5-733-2024, 2024.

Takaya, K. and Nakamura, H.: A Formulation of a Phase-Independent Wave-Activity Flux for Stationary and Migratory Quasi-geostrophic Eddies on a Zonally Varying Basic Flow, J. the Atmos. Sci., 58, 608 – 627, https://doi.org/10.1175/1520-0469(2001)058<0608:AFOAPI>2.0.CO;2, 2001.

---

## Author Comment (AC2)

**Response to comment by Anonymous Referee #2**

The authors present an evaluation of the composite evolution of wave reflection events in the North Pacific, which previous studies had connected with cold spells in North America. The identification of reflection events is based on an regional index of eddy heat flux, and the dynamics of these events are characterized using geopotential height, eddy heat flux, three-dimensional wave activity flux, and the wavenumber - phase speed spectra. The main results show a westward propagating ridge and the development of a trough downstream during the evolution of reflection events, which is associated with a change of sign in the meridional eddy heat flux. In turn these eddy heat fluxes can explain the meridional transport of colder air from the pole to lower latitudes, leading to a decrease of temperature over North America.

The paper is well-written and the presentation of the results is clear. I appreciate the level of detail given in the description and interpretation of each figure. I recommend publication after consideration of the comments given below.

*We would like to thank you for your helpful comments and will update our manuscript accordingly. Please, see our detailed answers below.*

**Comments:**

- The three-dimensional wave activity flux (WAF) used in this study (Plumb 1985) is a diagnostic for the stationary component of the wave field. However, the results presented reveal the presence of migrating eddies during the evolution of reflection events, especially at the end of their life cycle. Therefore, I would suggest to use a diagnostic of the WAF suitable to the problem at hand, for example the one proposed by Takaya and Nakamura (2001). On the other hand, the authors may reconsider the need of including a WAF analysis in the paper, given the detailed analysis of the geopotential and eddy heat fluxes.

*Thank you for your comment about the use of the Plumb WAF (Plumb, 1985). We agree, that the assumption of a stationary wave field does not hold when considering the entire evolution of reflection events. Our motivation was, that the wave field can be seen as approximately stationary for each individual day, which could make the use of Plumb WAF still valid. Furthermore, it has been argued that the Plumb WAF can be applied to investigate reflection events after filtering to keep planetary waves (wave-1 to 3) (Messori et al., 2022).*

*In line with your later comment, we also acknowledge the potential confusion arising from the way we display the vertical component of WAF in Figures 3 and 7. Thus, we suggest to display only the vertical component of the wave activity flux proposed by Takaya and Nakamura (2001). Despite a weaker signal than the Plumb (1985) WAF, the Takaya and Nakamura (2001) WAF exhibits a similar evolution during times of enhanced v'T' anomalies (cf. black lines in Fig. R1 to shading in Fig. 2 in the manuscript). Due to this similarity, we would like to include the discussion of vertical wave activity flux, as this also connects the findings of section 3.1.1 and 3.1.2 in the manuscript. We will change the arrows in figure 3 and 7 in the manuscript to black lines showing Takaya and Nakamura (2001) WAF, i.e., to match figure R1.*

[Figure]

**Figure R1.** *Hovmöller diagram of standardized anomalies of geopotential height averaged between 45N - 75N centered around the end date of reflection events at (a) 100 hPa, (b) 250 hPa and (c) 850 hPa in shading. Horizontal hatching marks significant negative anomalies and cross-hatching significant positive anomalies. Black lines indicate standardized anomalies of the vertical component of WAF filtered to wavenumbers-1 to 4 (Takaya and Nakamura, 2001), in steps of 0.1, excluding the zero-line. (d) Time series of ISP for westward-propagating Rossby waves and the 95% confidence interval (shaded area) at 100 hPa and 250 hPa. The continuous horizontal green line shows the median onset time of reflection events. The vertical lines mark longitudes of the Siberian (orange, dotted) and Canadian domains (purple, dashed).*

- lines 41-43 and 48-50. The eddy heat flux is part of the vertical component of different 3D wave activity flux diagnostics (e.g. Plumb 1985, 1986, Takaya and Nakamura 2001), but strictly speaking it is proportional to the vertical group velocity only in a zonal mean framework, where the EP flux represents the wave activity flux in the meridional-vertical plane.

*Thank you for highlighting this fact. We will adjust the description in lines 41-43 and 48-50, by adding: "... in a zonal mean framework".*

- Lines 77-78. Why is the meridional wind component not deseasonalized?

*Thank you for your question. Our description was unclear in this context. Neither the meridional wind nor the temperature used to compute the meridional eddy heat flux (v'T') are deseasonalized beforehand; only the v'T' itself is deseasonalized. Similarly, the meridional wind used to compute space-time spectra isn't deseasounalized beforehand either, as the spectra are deseasonalized afterwards. We will clarify this in the data description in lines 77-78: "Unless stated differently, we deseasonalized each field shown in the following analysis by subtracting the daily seasonal cycle,*

*computed…". Furthermore, we will add a sentence to the end of the paragraph: "The meridional eddy heat flux, WAF and space-time Rossby wave spectra are computed from the full fields and deseasonalized afterwards."*

- Eq. 1. Please specify the pressure level at which RI is defined (is it 100 hPa?).

*Yes, the reflective index RI is defined at 100 hPa, following Messori et al. (2022) and Matthias and Kretschmer (2020). We will clarify this in the manuscript by adding "at 100 hPa" in the description of RI.*

- Line 92. Are the weather regimes computed using year-round data? Why not only winter data? Is there any sensitivity there?

*Yes, the weather regimes are computed using year-round data, as described in Lee et al. (2023). Since the seasonal cycle is normalized using empirical orthogonal function analysis combined with k-means clustering, the weather regimes are insensitive to seasonality. Lee et al. (2023) also note that the year-round regimes differ only marginally from winter-only regimes, while the classification was also more extensively tested for significance and robustness than existing seasonally-dependent classifications. Further, we note that the year-round classification of Grams et al. (2017) has been extensively used for season-specific analysis, such as in the stratosphere-troposphere analysis of Domeisen et al. (2020). We see no reason why a season-specific analysis should use season-specific regimes; the purpose of year-round regimes is that they can be used in any season. With respect to your comment, we will mention in line 92: "…, [and no regime (N)], which are insensitive to seasonality.")*

- The arrows in Fig. 3 are confusing. The vertical component represents the vertical component of the WAF(?), but the vertical axis in the figure represent time. If the authors decide to keep the analysis of the WAF (after switching to Takaya and Nakamura 2001), then I would suggest to plot it on a map like Fig. 4.

*Thank you for your suggestion. We agree that this way of displaying the vertical component of WAF is confusing due to the y-axis displaying time and not vertical height levels. In line with your earlier comment, we decided to switch to the WAF proposed by Takaya and Nakamura (2001), and change the arrows in figure 3 and 7 in the manuscript to black lines showing Takaya and Nakamura (2001) WAF, i.e., to match figure R1.*

- Lines 297-298. How many reflective events are also regime transitions? Are the composites of Figs. 7-8 dominated by the signal of those transitions that are also reflective events? Does the composites of transitions that are NOT reflective events show a similar qualitative evolution to those that are?

*If one includes regime transitions that have the central date of AKR events after the end of reflection events, 24 of 45 reflection events with this overlap can be detected. Figure 11 in Messori et al. (2022) can provide more details on the weather regimes during reflection events.*

*PT-AKR regime transitions during and outside of reflection events, are both similar to the composite of all PT-AKR regime transitions in the troposphere (cf. Fig. R2 b, c and R3 b, c to Fig. 7 b, c in the manuscript), while the 21 regime transitions happening during reflection events resemble also the composite of all reflection events in the lower stratosphere (Fig. R2 a to Fig. R1 a). Conversely, the 41*

*regime transitions happening outside of reflection events exhibit a signal different to reflection events in the lower stratosphere (Fig. R3 a to R1 a).*

*Furthermore, only the difference between PT and AKR events occurring during reflection events shows an enhancement in westward propagating wave-1 (17 of 107 PT and 23 of 91 AKR events during reflection), while those regimes occurring outside of reflection events (90 of 107 PT and 68 of 91 AKR) don't show this signal (Fig. R4 and R5). Due to the larger number of PT and AKR events happening outside of reflection events, their spectral behavior resembles the composite of all PT and AKR events more closely than those regimes happening during reflection events. With respect to your comment, we plan to add figures R2 to R5 to the appendix and mention briefly the characteristic similarities and differences between PT-AKR regime transitions happening during reflection events and outside of reflection events.*

[Figure]

**Figure R2.** *Same as Fig. R1, but for PR-AKR regime transitions occurring during reflection events, with AKR central date as day 0.*

[Figure]

**Figure R3.** *Same as Fig. R1, but for PR-AKR regime transitions occurring outside of reflection events, with AKR central date as day 0.*

[Figure]

**Figure R4.** *Standardized anomalies of spectral power SV'V' of each harmonic at 100 hPa during reflection events with (a) PT, (b) AKR (c) difference between PT and AKR weather regimes in shading. Sub-panels (d) - (e) show the same, but for 250 hPa. Black contours show the DJFM mean for all years (100 hPa: steps of 0.05 m2 s −2 Δc−1 from 0.05 m2 s −2 Δc−1 to 0.35 m2 s −2 Δc−1; 250 hPa: steps of 0.1 m2 s −2 Δc−1 from 0.1 m2 s −2 Δc−1 to 0.7 m2 s −2 Δc−1). Horizontal hatching indicates significant negative anomalies, cross-hatching indicates significant positive anomalies.*

[Figure]

**Figure R5.** *Same as Fig. R6, but for PT and AKR regimes outside of reflection events.*

- I find section 3.3 unnecessary for the goals of the study. The summary at the end of the section would be a sufficient remark to make in the text without the need of showing the figures.

*Thank you for your suggestion to shorten the discussion about regionalized events. In accordance with Nili Harnik's comments, we would like to highlight in the manuscript that space-time spectral analysis is applicable to investigate reflection events, despite potential limitations of regional or short-lived events. Yet, we understand that one can put less weight on that discussion in the manuscript, than we have right now. We will consider your suggestion and might only put a summary of the regionalization results to the discussion and move figures 9 and 10 and to the appendix, instead of the detailed description in section 3.3.*

References:

Takaya, K., and H. Nakamura, 2001: A Formulation of a Phase-Independent Wave-Activity Flux for Stationary and Migratory Quasigeostrophic Eddies on a Zonally Varying Basic Flow. J. Atmos. Sci., 58, 608–627

**References**

Domeisen, D. I. V., Grams, C. M., and Papritz, L.: The role of North Atlantic–European weather regimes in the surface impact of sudden stratospheric warming events, Weather Clim. Dynam., 1, 373–388, https://doi.org/10.5194/wcd-1-373-2020, 2020.

Grams, C., Beerli, R., Pfenninger, S., Staffell. I. and Wernli, H.: Balancing Europe's wind-power output through spatial deployment informed by weather regimes. Nature Clim. Change 7, 557–562, https://doi.org/10.1038/nclimate3338, 2017.

Lee, S. H., Tippett, M. K., and Polvani, L. M.: A New Year-Round Weather Regime Classification for North America, J. Climate, 36, 7091 – 7108, https://doi.org/10.1175/JCLI-D-23-0214.1, 2023.

Matthias, V. and Kretschmer, M.: The Influence of Stratospheric Wave Reflection on North American Cold Spells, Mon. Weather Rev., 148, 1675 – 1690, https://doi.org/10.1175/MWR-D-19-0339.1, 2020.

Messori, G., Kretschmer, M., Lee, S. H., and Wendt, V.: Stratospheric downward wave reflection events modulate North American weather regimes and cold spells, Weather Clim. Dynam., 3, 1215–1236, https://doi.org/10.5194/wcd-3-1215-2022, 2022.

Plumb, R. A.: On the Three-Dimensional Propagation of Stationary Waves, J. Atmos. Sci., 42, 217 – 229, https://doi.org/10.1175/1520-0469(1985)042<0217:OTTDPO>2.0.CO;2, 1985.

Takaya, K. and Nakamura, H.: A Formulation of a Phase-Independent Wave-Activity Flux for Stationary and Migratory Quasi-geostrophic Eddies on a Zonally Varying Basic Flow, J. the Atmos. Sci., 58, 608 – 627, https://doi.org/10.1175/1520-0469(2001)058<0608:AFOAPI>2.0.CO;2, 2001.

---

## Referee Report (RR1)

Review of revised version of "Dynamics of stratospheric wave reflection over the North Pacific" by Schutte et al.

The authors have addressed many of my comments and the paper reads tighter and the motivation is clearer now. However, there are a few points that are still unclear, and in some ways misleading, that should be improved before publication.

Main points:

1) The emphasis on the role of the stratosphere in the introduction and abstract is a bit misleading. The focus of the analysis is on understanding the tropospheric and lower stratospheric evolution of the positive-negative heat flux dipole.

The abstract refers to negative v'T' events as Stratospheric Wave Reflection events which *"involve the upward propagation of planetary waves, which are subsequently reflected downward by the stratospheric polar vortex."* (1st sentence of the abstract).

The first few paragraphs of the introduction discuss a downward effect of the stratosphere on the troposphere, specifically, via downward wave reflection, and including how to diagnose downward reflection events. I am not sure, however, that the negative v'T' events discussed here are classical downward reflection events in the sense of an upward wave pulse propagating to the stratosphere, then reflecting back down from a reflecting surface. At least, the data shown does not necessarily support this. Rather, it shows that the combination of a stratospheric wave 1 and a westward propagating tropospheric medium scale wave packet which is also developing downstream, combine to give a dynamical evolution of upward wave activity flux over Siberia and downward over Canada.

In addition some of the results do not fit this chain of events - for example the sentence on lines 281-282 : "*The overall Rossby wave activity mostly remains close to its climatological average and only increases in the stratosphere after the end of reflection events (Fig. 6 b)."* I would expect the wave activity to increase in the stratosphere before reflection events, and to increase in the troposphere after the waves are reflected downwards, if the secenario assumed apriori in the introduction is true. Rather, this result (though see comment on figure 6 below) maybe supports the notion that the negative v'T' is due to a superposition of stratospheric and tropospheric waves which have different sources and phase speeds.

The only suggestion for a classical evolution of upward propagation to the stratosphere followed by a downward propagation downstream is found, to my mind, in figure A8 which shows the stratospheric phase tilt changes from westward with height to vertical with height.

2) The discussion of the validity of the spectral diagnostics (lines 304-313) is convincing to some level but not for all the points it is used for. Specifically, it is not convincing enough to justify using it for figure 6.

More explicitly:

Line 145 refers to figures 5 and A2 but these figures do not really show the diagnostic is able to capture daily time scale variations. On the other hand, figure 3 shows the Hovmöller diagram alongside ISP_west, and it does seem like ISP captures the temporal evolution of phase speed, though maybe the change from stationary to westward phase speed is sharper in the Hovmöller diagram than in ISP. Figure 7 also makes such a comparison, and the spectral diagnostic captures the broad features but maybe not as nicely as in figure 3.

A better test maybe would be to look at specific events and compare the time evolving spectral diagnostics with a corresponding Hovmöller diagram (which is not a composite), and show that these match for a few events.

- Figure 6: I find the use of the diagnostic potentially problematic for this figure, especially conclusions about detailed daily-timescale evolution, c.f. comments like that made on line 297: "and in the stratosphere a few days later". I am not convinced the temporal resolution is good enough. Moreover, as fig A7 clearly shows, that stratosphere is dominated by smaller zonal wavenumbers than the troposphere, meaning that the significant ISP_west enhancement is occurring for larger frequencies in the troposphere than the stratosphere. Can't this, by itself, lead to differences in the timing of statistical significance of the signal?

I expect the spectral diagnostics can only point out changes on coarser time scales, thus it is more convincing to use for onset - end comparison plots like figure 5, given that the phase propagation lasts for quite a few days, as does the stationarity of the anomalies during the first part of the events.

Minor comments

line 188 - how do you deduce Rossby wave breaking from thr figures?

line 251 - I would check references by Randel and co-authors from the 1980s which studies medium scale waves in the southern hemisphere for a tropospheric example which is maybe more relevant to the case studies here. Specifically there is: Randel, W. J., and J. L.

Stanford, 1985: An Observational Study of Medium-Scale Wave Dynamics in the Southern Hemisphere Summer. Part II: Stationary-Transient Wave Interference. *J. Atmos. Sci.*, **42**, 1189–1197, https://doi.org/10.1175/1520-0469(1985)042<1189:AOSOMS>2.0.CO;2.

Figure 4- hard to see the green contours

line 268- the reference to waves 2-5 in figure 5c is confusing because there changes in these wavenumbers are not statistically significant

There are too many appendix figures - it is hard to follow the paper when the reader is referred to these figures so much. This impression is strengthened by the fact that the appendix figures are not referred to in the order of appearance. specifically, Fig A9, A16, A17 appear earlier than the reference to figures before them.

Line 281: the following statement -" *and only increases in the stratosphere after the end of reflection events (Fig. 6 b)."* I am puzzled by this- I would expect the wave activity to increase in the stratosphere before reflection events, and to increase in the troposphere after the waves are reflected downwards. This maybe supports the notion that the negative v'T' is due to a superposition of stratospheric and tropospheric waves and not wave activity actually going down from the stratosphere to the troposphere... see major comment 1

Figure 6, The discussion of this figure on lines 291-295 is confusing. Lines 291-295 took several re-reads for me to figure out what the authors are trying to say, specifically due to some of the implied causality words (can be understood by, a result of). Are you implying that the westward phase propagation in the lower stratosphere upper troposphere **is consistent with** the stratospheric waves vertical phase tilt changing from a westward to a vertical one (as is expected in downward wave reflection), with the phase of the wave not changing too much at 10mb?

In addition, the total ISP subplot (6b) does not seem to match the eastward and westward ISP plots 6c and 6d, especially at positive time lags - shouldn't it be some sort of sum of the two? At the same time, I am not sure how the information in figure 6 adds to the discussion, thus I would consider dropping this analysis altogether.

Section 3.2 - I think a plot like figures 4 or A17 for the PT-AKR transition would be useful to see.

Lines 342-343 - discussion of figures A11-A12 refers only to differences in the stratosphere, but I would say that also the tropospheric evolutions are different- there is essentially no westward phase speed for the regime transitions in these figures outside of reflection events (A12), and only a very weak westward phase propagation for those with a reflection event (A11).

Sentence starting at the end of line 343, to 345: I am confused. I assume the authors mean to point out the small region of significant difference in the region of 250mb waves 3-4 phase speeds that protrudes into the contoured region of phase space that dominates climatology in figure A14 which does not appear in A13? I would state this reference to A13 vs A14 explicitly, but this is contrary to what A11 vs A12 show for the 250mb level...

Discussion

The reasoning in the two sentences on lines 381-385 is not clear. Specifically the way this sentence is phrased implies that the westward propagation results from downward reflection. However, it is possible that the westward propagation results from internal tropospheric free Rossby wave dynamics, and its superposition with stratospheric stationary waves could give a signature of downward reflection.

I think maybe a schematic might help summarize the results more clearly.

Appendix figures:

Figure A7 - Title should be changed - this is not an evolution, rather a longitude height section breakup to different wavenumbers.

Figure A18 - I don't find the signal strong enough to be convincing. The variability is so much stronger...

---

## Author Response (AR2)

**Response to comment by Referee #1: Nili Harnik**

*Review of revised version of "Dynamics of stratospheric wave reflection over the North Pacific" by Schutte et al.*

*The authors have addressed many of my comments and the paper reads tighter and the motivation is clearer now. However, there are a few points that are still unclear, and in some ways misleading, that should be improved before publication.*

**We would like to thank you for your detailed and thoughtful review. We greatly appreciate the time and effort you have invested in providing constructive feedback. Below, we copied your comments in** *italic* **and address your comments point by point in bold blue.**

*Main points:*

*1) The emphasis on the role of the stratosphere in the introduction and abstract is a bit misleading. The focus of the analysis is on understanding the tropospheric and lower stratospheric evolution of the positive-negative heat flux dipole.*

*The abstract refers to negative v'T' events as Stratospheric Wave Reflection events which "involve the upward propagation of planetary waves, which are subsequently reflected downward by the stratospheric polar vortex." (1st sentence of the abstract).*

*The first few paragraphs of the introduction discuss a downward effect of the stratosphere on the troposphere, specifically, via downward wave reflection, and including how to diagnose downward reflection events. I am not sure, however, that the negative v'T' events discussed here are classical downward reflection events in the sense of an upward wave pulse propagating to the stratosphere, then reflecting back down from a reflecting surface. At least, the data shown does not necessarily support this. Rather, it shows that the combination of a stratospheric wave 1 and a westward propagating tropospheric medium scale wave packet which is also developing downstream, combine to give a dynamical evolution of upward wave activity flux over Siberia and downward over Canada.*

*In addition some of the results do not fit this chain of events - for example the sentence on lines 281-282 : "The overall Rossby wave activity mostly remains close to its climatological average and only increases in the stratosphere after the end of reflection events (Fig. 6 b)." I would expect the wave activity to increase in the stratosphere before reflection events, and to increase in the troposphere after the waves are reflected downwards, if the scenario assumed a priori in the introduction is true. Rather, this result (though see comment on figure 6 below) maybe supports the notion that the negative v'T' is due to a superposition of stratospheric and tropospheric waves which have different sources and phase speeds.*

*The only suggestion for a classical evolution of upward propagation to the stratosphere followed by a downward propagation downstream is found, to my mind, in figure A8 which shows the stratospheric phase tilt changes from westward with height to vertical with height.*

**Reflection events, as defined in our study via meridional eddy heat flux anomalies over Siberia and North Canada, have been strongly connected to stratospheric dynamics in earlier work (e.g., Matthias and Kretschmer, 2020; Messori et al., 2022; Millin et al., 2022). While the events are defined at 100 hPa, the dynamics of the stratospheric polar vortex at higher levels play a crucial role in enabling reflection, for example by creating a vertical reflective surface, which is also present during the events defined via the reflective index (Fig. 2 and its discussion in Messori et al., 2022). Simultaneously, the lower stratosphere remains essential for diagnosing upward and downward propagating Rossby waves using our reflective index. Unlike indices that solely diagnose anomalous downward wave activity flux, the reflective index used here captures both upward and downward wave flux (Matthias and Kretschmer, 2020).**

As you noted, Figure A8 provides evidence of the stratospheric phase tilt transitioning from westward to vertical with height, which is characteristic for downward wave reflection. Our analysis also suggests that the stratospheric wave-1 anomaly acts as a mediator, while the smaller-scale waves (e.g., wave-3 and 4) from the troposphere are reflected. The role of wave-1 aligns with the concept of vortex stretching, as discussed in Cohen et al. (2022). This connection between wave-1 and wave-3 and 4 highlights the role of the stratosphere in organizing and reflecting waves back to the troposphere, supporting our interpretation of these events as stratospheric downward wave reflection. Nevertheless, we will acknowledge in lines 381-385 in the discussion that the observed structure in v'T' anomalies potentially arises from the superposition of stratospheric and tropospheric waves.

Regarding the comment on lines 281–282, we would like to thank you for spotting this inconsistency. We agree that Rossby wave activity increases in the stratosphere before the onset and in the troposphere during reflection events, which is apparent in Fig. 6b, as well. We will adjust the description in lines 281-282 accordingly. Nevertheless, it is important to differentiate between the overall Rossby wave activity and the subset of waves actively undergoing reflection. Waves that are reflected downward may not significantly increase overall stratospheric wave activity because they might be only a small fraction of the entire wave spectrum and potentially don't remain in the stratosphere long enough to be pronounced or accumulate there. Instead, these waves return to the troposphere, contributing to surface anomalies rather than persistent stratospheric signals. This contrasts with the period around the end of reflection events, where the blocking signal of the AKR regime results in enhanced wave activity and waves might remain longer in the stratosphere, possibly breaking or dissipating instead of being reflected downwards.

We appreciate your observations and adjust the discussion of Figure 6 also with respect to later comments.

*2) The discussion of the validity of the spectral diagnostics (lines 304-313) is convincing to some level but not for all the points it is used for. Specifically, it is not convincing enough to justify using it for figure 6.*

*More explicitly:*

*Line 145 refers to figures 5 and A2 but these figures do not really show the diagnostic is able to capture daily time scale variations. On the other hand, figure 3 shows the Hovmöller diagram alongside ISP_west, and it does seem like ISP captures the temporal evolution of phase speed, though maybe the change from stationary to westward phase speed is sharper in the Hovmöller diagram than in ISP. Figure 7 also makes such a comparison, and the spectral diagnostic captures the broad features but maybe not as nicely as in figure 3.*

*A better test maybe would be to look at specific events and compare the time evolving spectral diagnostics with a corresponding Hovmöller diagram (which is not a composite), and show that these match for a few events.*

*- Figure 6: I find the use of the diagnostic potentially problematic for this figure, especially conclusions about detailed daily-timescale evolution, c.f. comments like that made on line 297: "and in the stratosphere a few days later". I am not convinced the temporal resolution is good enough. Moreover, as fig A7 clearly shows, that stratosphere is dominated by smaller zonal wavenumbers than the troposphere, meaning that the significant ISP_west enhancement is occurring for larger frequencies in the troposphere than the stratosphere. Can't this, by itself, lead to differences in the timing of statistical significance of the signal?*

*I expect the spectral diagnostics can only point out changes on coarser time scales, thus it is more convincing to use for onset - end comparison plots like figure 5, given that the phase propagation lasts for quite a few days, as does the stationarity of the anomalies during the first part of the events.*

**Thank you for pointing out the need to clarify our comparison. In line 145, we intended to compare the spectral characteristics shown in Figure 5a–c with the behavior of specific waves depicted in Figure A2. Specifically, the large-scale waves exhibit distinct differences in behavior between the onset and end of reflection events, as highlighted in the harmonics of large-scale waves in Figure 5a–c. We will highlight that these figures complement each other, with Figure A2 providing additional insights into wave propagation during the onset and end. Following your suggestion, we will also refer to Figures 3 and 7, which show that the spectral diagnostics broadly capture the temporal evolution of phase speed. Instead of discussing this relation in further detail in section 2.3., we decided to elaborate on it at the end of section 3.1.3.**

**We acknowledge the concern regarding the ability of the diagnostic to capture daily timescale variations. Thus, we computed a Hovmöller diagram for one specific reflection event (Fig. R1). While interpreting daily changes in spectral diagnostics may have some limitations, the agreement between the Hovmöller diagrams and ISP_west supports our conclusion that the space-time spectra can capture meaningful changes in Rossby wave behavior, even on shorter timescales.**

[Figure]

***Figure R1.*** *Hovmöller diagram of standardized anomalies of geopotential height averaged between 45°N - 75°N centred around the end date of one reflection event at (a) 100 hPa, (b) 250 hPa and (c) 850 hPa in shading. (d) Time series of ISP for westward-propagating Rossby waves at 100 hPa and 250 hPa with the x-axis being flipped, so that enhanced activity of westward-propagating Rossby waves lies to the left of the zero line. The vertical lines mark longitudes of the Siberian (orange, dotted) and Canadian (purple, dashed) domains.*

**We appreciate your observation regarding the temporal resolution and the potential influence of different wave frequencies between the troposphere and stratosphere. While the spectral diagnostics shown in Figure 6 may not fully support detailed daily-timescale conclusions, their ability to capture the broader temporal evolution of wave activity remains robust. We will adjust the statement on line 295-297 to reflect a more cautious interpretation to: "Additionally, westward-propagating Rossby waves become more active in the troposphere and stratosphere around 1 week before the end of reflection events. This matches temporally with …" Furthermore, we will adjust the discussion on lines 284-285.**

*Minor comments*

*line 188 - how do you deduce Rossby wave breaking from the figures?*

**We did not directly deduce Rossby wave breaking from the figures, but rather speculate on a potential mechanism that could decrease the downward pulse. Specifically, it is possible that upward-propagating Rossby waves may either break or become absorbed after some time, instead of being reflected. This mechanism could potentially contribute to the end of reflection events, and we will clarify this distinction in the manuscript.**

*line 251 - I would check references by Randel and co-authors from the 1980s which studies medium scale waves in the southern hemisphere for a tropospheric example which is maybe more relevant to the case studies here. Specifically, there is: Randel, W. J., and J. L. Stanford, 1985: An Observational Study of Medium-Scale Wave Dynamics in the Southern Hemisphere Summer. Part II: Stationary-Transient Wave Interference. J. Atmos. Sci., 42, 1189–1197, https://doi.org/10.1175/1520-0469(1985)0422.0.CO;2.*

**Thank you for suggesting the reference by Randel and Stanford (1985), which is a relevant addition. We will include it in line 251.**

*Figure 4- hard to see the green contours*

**Thank you for pointing this out. We will increase the size of the green contours for better clarity (see example in Figure R2). This adjustment will be applied to Figures 4, A6, and A17.**

*line 268- the reference to waves 2-5 in figure 5c is confusing because there changes in these wavenumbers are not statistically significant.*

**While the reduction of spectral power in wavenumbers 2-5 is not statistically significant according to our relatively strict testing procedure, which accounts for the false discovery rate, we believe it is still worth mentioning this well-pronounced tendency. We will clarify in the text that this part of the signal is not statistically significant.**

*There are too many appendix figures - it is hard to follow the paper when the reader is referred to these figures so much. This impression is strengthened by the fact that the appendix figures are not referred to in the order of appearance. specifically, Fig A9, A16, A17 appear earlier than the reference to figures before them.*

**We will reorganize the appendix figures to ensure they are referred to in the order of appearance, by switching positions of Fig. A9 and A8 and by placing Fig. A16 and A17 in the correct position. While the appendix contains a large number of figures, many were included in response to reviewer comments to provide additional clarity and support for the analysis. To**

**support our methodology and analysis, we would like to keep the appendix figures available for future readers.**

*Line 281: the following statement -" and only increases in the stratosphere after the end of reflection events (Fig. 6 b)." I am puzzled by this- I would expect the wave activity to increase in the stratosphere before reflection events, and to increase in the troposphere after the waves are reflected downwards. This maybe supports the notion that the negative v'T' is due to a superposition of stratospheric and tropospheric waves and not wave activity actually going down from the stratosphere to the troposphere... see major comment 1.*

**If waves are reflected back downward, it is possible that we do not observe anomalously high Rossby wave activity in the stratosphere because the waves do not remain there long enough. In contrast, after the end of reflection events, waves may instead propagate through a greater depth of the stratosphere, leading to the observed increase in wave activity. Additionally, reflection events are associated with higher wave activity near the surface about one week before their end, which coincides with decreasing surface temperatures over North America. We have also addressed similar concerns in response to the first major comment in more detail.**

*Figure 6, The discussion of this figure on lines 291-295 is confusing. Lines 291-295 took several re-reads for me to figure out what the authors are trying to say, specifically due to some of the implied causality words (can be understood by, a result of). Are you implying that the westward phase propagation in the lower stratosphere upper troposphere is consistent with the stratospheric waves vertical phase tilt changing from a westward to a vertical one (as is expected in downward wave reflection), with the phase of the wave not changing too much at 10mb?*

**Thank you for pointing this out. We apologize for the lack of clarity in our discussion. We appreciate your careful reading and will rewrite these sentences to: "The westward propagation of large-scale anomalies in the lower stratosphere and upper troposphere coincides with a shift of Rossby wave phase lines, that typically tilt westward with height, to a more vertical orientation over the North Pacific during stratospheric wave reflection (Fig. A7). This change in vertical phase tilt is consistent with a more pronounced enhancement of eastward phase speeds at higher stratospheric levels compared to those below (Fig. 6 a)." This way, the discussion should better convey that the westward phase propagation in the lower stratosphere and upper troposphere aligns with the stratospheric wave's vertical phase tilt changing from westward to vertical, as expected during downward wave reflection. In Figure A8, one can also see that the phase of the wave doesn't change too much at 10 hPa during reflection events.**

*In addition, the total ISP subplot (6b) does not seem to match the eastward and westward ISP plots 6c and 6d, especially at positive time lags - shouldn't it be some sort of sum of the two? At the same time, I am not sure how the information in figure 6 adds to the discussion, thus I would consider dropping this analysis altogether.*

**We would argue that Figure 6b aligns reasonably well with the average of 6c and 6d, particularly when considering that ISP_east typically has more weight due to the climatological prevalence of eastward-propagating Rossby waves. Since this figure displays the temporal evolution of spectral metrics and provides an overview of their vertical structure, we would like to keep Figure 6 in the manuscript to complement the results from Figure 5.**

*Section 3.2*

*- I think a plot like figures 4 or A17 for the PT-AKR transition would be useful to see.*

**The overview of PT-AKR transition in form of a plot like Figure 4 also highlights the similarity between reflection events and the regime transition (Fig. R2). To avoid adding additional**

**appendix figures, we believe Fig R2 does not add substantial additional information beyond that contained in the Hovmöller plots (e.g., Fig. 7).**

[Figure]

**z anomaly, v'T' anomaly and temperature anomaly during regime transition (PT-AKR)**

*Figure R2.* Geopotential height anomalies in shading and v'T' anomalies in green contours (a, f) during onset, (b, g) 4 days before end, (c, h) at the end date and (d, i) difference between onset and end of PT-AKR regime transition events at 100 hPa (first row) and 500 hPa (second row).The central date of PT is denoted as onset and the central date of AKR as end of the regime transition events. Horizontal hatching marks significant negative geopotential height anomalies and cross-hatching significant positive anomalies. (e, j) Temperature anomalies in shading, geopotential height field in purple and v'T' anomalies in green 4 days before end.

*Lines 342-343 - discussion of figures A11-A12 refers only to differences in the stratosphere, but I would say that also the tropospheric evolutions are different- there is essentially no westward phase speed for the regime transitions in these figures outside of reflection events (A12), and only a very weak westward phase propagation for those with a reflection event (A11).*

**We agree that there are also some differences between reflective and non-reflective regime transitions in the tropospheric evolutions, specifically with respect to the phase speed, but the most pronounced differences are found in the stratosphere. This is connected to weather regimes being defined on the tropospheric level of 500 hPa, and expanding on the tropospheric differences here would go beyond the scope of the discussion in lines 342-343, which we have therefore decided not to extend. While the PT-AKR regime shift outside of reflection events could be more related to the superposition and interference of waves, it is likely, that we observe the weak westward-propagation during reflection events due to the additional influence of the stratosphere. Nevertheless, there is also a weak tendency of the ridge to propagate westward outside of reflection events at 250 hPa (Fig. A12).**

*Sentence starting at the end of line 343, to 345: I am confused. I assume the authors mean to point out the small region of significant difference in the region of 250mb waves 3-4 phase speeds that protrudes into the contoured region of phase space that dominates climatology in figure A14 which does not appear in A13? I would state this reference to A13 vs A14 explicitly, but this is contrary to what A11 vs A12 show for the 250mb level...*

**Yes, we are referring to the small region of significant difference for westward-propagating waves 3 and 4, as seen in Figure A14 f. Even though this signal is non-significant for regimes**

**during reflection events in Figure A13 f, the tendency is still evident for PT-AKR regime transitions during and outside of reflection events. Therefore, we attribute this part of the anomalies in the Rossby wave spectra during reflection events to the PT-to-AKR weather regime shift. Despite the more stationary setup outside of reflection events, we can still observe the tendency of the ridge to propagate westward (Fig. A12 b). The westward propagation is even more pronounced for PT-AKR regime transitions during reflection events (albeit being still slow in Fig. A11), which is also reflected in the higher amplitude of anomalies for westward-propagating waves-3 and 4 (Fig. A13 f). Even though these anomalies are not significant, the signal is also represented in the respective Hovmöller diagram (Fig. A11 b).**

*Discussion*

*The reasoning in the two sentences on lines 381-385 is not clear. Specifically, the way this sentence is phrased implies that the westward propagation results from downward reflection. However, it is possible that the westward propagation results from internal tropospheric free Rossby wave dynamics, and its superposition with stratospheric stationary waves could give a signature of downward reflection.*

**We agree that the phrasing is misleading and could imply that the westward propagation is solely a result of downward reflection. We will rephrase the sentences to clarify that the westward propagation could also arise from internal tropospheric free Rossby wave dynamics, with its interaction and superposition with stratospheric stationary waves contributing to the observed signature. They read now: "The westward propagation of medium-scale Rossby waves in the upper troposphere could be linked to internal tropospheric free Rossby wave dynamics. Their interaction and superposition with planetary-scale waves from the stratosphere may contribute to the observed signature of downward reflection. Indeed, we observe the enhancement of westward-propagating wave-1 during stratospheric wave reflection events, which suggests that the evolution of v'T' anomalies may be influenced by this coupling between stratospheric and tropospheric waves."**

*I think maybe a schematic might help summarize the results more clearly.*

**Thank you for the suggestion! We have created a summary figure (Figure R3) that we will include in the manuscript.**

*Appendix figures:*

*Figure A7 - Title should be changed - this is not an evolution, rather a longitude height section breakup to different wavenumbers.*

**Thank you for pointing out this mistake. We will correct the title.**

*Figure A18 - I don't find the signal strong enough to be convincing. The variability is so much stronger…*

**We agree that there is a large variability in 100 hPa zonal wind, but there is also large variability at upper-level levels (Figure A5 in Messori et al., 2022). Thus, there are probably some cases, where wave breaking and the subsequent vortex deceleration occurs, but there is likely also a large number of cases, where other dynamics dominate the behavior of the stratospheric polar vortex, and we don't see the slow-down of 100 hPa zonal wind. As mentioned in lines 388 -390, this is just one potential mechanism.**

**a    Onset of reflection events**

[Figure]

**b    During reflection events**

[Figure]

**c    End of reflection events**

[Figure]

***Figure R3.*** *Summary figure of reflection events.*

**References**

Cohen, J., Agel, L., Barlow, M., Furtado, J. C., Kretschmer, M., and Wendt, V.: The "Polar Vortex" Winter of 2013/2014, J. Geophy. Res. Atmospheres, 127, e2022JD036 493, https://doi.org/10.1029/2022JD036493, 2022.

Matthias, V. and Kretschmer, M.: The Influence of Stratospheric Wave Reflection on North American Cold Spells, Mon. Weather Rev., 148, 1675 – 1690, https://doi.org/10.1175/MWR-D-19-0339.1, 2020.

Messori, G., Kretschmer, M., Lee, S. H., and Wendt, V.: Stratospheric downward wave reflection events modulate North American weather regimes and cold spells, Weather Clim. Dynam., 3, 1215–1236, https://doi.org/10.5194/wcd-3-1215-2022, 2022.

Millin, O. T., Furtado, J. C., and Basara, J. B.: Characteristics, Evolution, and Formation of Cold Air Outbreaks in the Great Plains of the United States, J. Climate, 35, 4585 – 4602, https://doi.org/10.1175/JCLI-D-21-0772.1, 2022.

Randel, W. J. and Stanford, J. L.: An Observational Study of Medium-Scale Wave Dynamics in the Southern Hemisphere Summer. Part II: Stationary-Transient Wave Interference, J. Atmos. Sci., 42, 1189 – 1197, https://doi.org/10.1175/1520-0469(1985)042<1189:AOSOMS>2.0.CO;2, 1985.

---

## Author Response (AR3)

**Response to comments by the co-editor**

Dear Michael Schutte et al.,

thank you for the additional revisions in response to the additional comments by referee 1 (Nili Harnik). I have thoroughly assessed all points and think that your applied changes to the manuscript are mostly adequate, but I do have a few minor suggestions that I hope will further improve the presentation.

*We would like to thank the co-editor for handling the manuscript and for the following feedback. This has helped us to highlight the main messages and to clarify several points more precisely. We greatly appreciate your thorough and constructive assessment and provide our answers below, addressing each point individually. Line numbers and figures refer to the track-changes version.*

1. Regarding main point 1) by the referee (~misleading portrayal of role of stratosphere in the abstract and introduction): I share the sentiment by the referee that the abstract and introduction don't spell out sufficiently that the reflection events under consideration here are different from the "classical" reflection events where the full (zonal-mean) wave activity flux changes from upward to downward. I encourage the authors to emphasize this distinction a bit more in the abstract and introduction. Specific places are:

- near the beginning of the abstract (the first sentence seems to refer to "classical" events, while the third sentence at first seems to suggest a direct reference to these "classical" events but then switches to partial wave reflection over a region)
- lines 47-50 (line numbers here and in the following refer to the track-changes version) where the regional index is first mentioned, but the distinction of the resulting events from the "classical" events is not spelled out
- perhaps below the event definition (Eq. 1)

BTW, since Fig. A9 (current version, previous Fig. A8) shows anomalies and not full fields, it cannot be used as evidence for changes in the phase tilt. However, regional changes in phase tilt can be seen in Fig. 12 of Messori et al. (2022) and perhaps this could be pointed out with some of the suggested clarification regarding regional versus full wave reflection.

*We agree with your suggestion and revised the abstract and introduction to more clearly distinguish our analysis from "classical" reflection events:*

- *The third sentence in the abstract reads now:'Here, we investigate **a set of** wave reflection events characterised by an enhanced difference between poleward eddy heat flux over the Northwest Pacific and equatorward eddy heat flux over Canada.'*
- *The beginning of the paragraph in lines 46-49 reads now: 'To overcome this challenge, Matthias and Kretschmer (2020) proposed an index based on regional averages of the meridional eddy heat flux that is able to capture wave reflection events over the North Pacific. **This new index, unlike more traditional reflection events based on zonal-mean diagnostics, builds on** the climatological pattern of the meridional eddy heat flux **that displays upward** wave activity flux over [...]'*

- *In line 97, we have added the following sentence after the definition of reflection events: 'This definition of reflection events differs from traditional approaches relying on zonal-mean metrics.'*

*Additionally, we removed Fig. A9 and instead reference Fig. 12 from Messori et al. (2022) as evidence for changes in the phase tilt in line 306.*

2. Regarding main point 2) by the referee (daily timescales implied in Fig. 6): I appreciate the example Hovmöller diagram (Fig. R1) and encourage the authors to include a statement in the text related to the caveat about temporal resolution and why you find Fig. 6 to offer robust insights into the broader temporal wave activity evolution.

*Figure 6 provides a comprehensive perspective on the broader temporal evolution of wave activity, illustrating how Rossby wave activity differs between the troposphere and stratosphere during reflection events. While space-time spectra, such as those in Fig. 5 and Fig. A8, capture detailed changes in specific wave-phase speed harmonics, Fig. 6 serves as a complementary tool by emphasizing temporal scales and vertical variations in wave activity. We further clarified its role after its discussion in line 298.*

*We further adjusted the discussion about temporal resolution in lines 318–328 at the end of Section 3.1.3 and included Fig. R1 (a Hovmöller diagram of the January-February 1981 reflection event) in the appendix as Fig. A11 to demonstrate the ability of space-time spectra to capture relevant changes on shorter time scales.*

3. Statistical significance: please specifically refer to "statistical significance" (as opposed to "physical significance") wherever this is the intended meaning of "significance" (incl. instances of "[something] is/is not significant").

*Thank you for highlighting the distinction between statistical and physical significance. We revised the manuscript to explicitly refer to "statistical significance" where appropriate.*

4. Large number of additional figures in appendix: since it could indeed be distracting to readers to frequently be referred to additional figures, I would encourage a related statement (about the role of the appendix) at the end of the introduction (where you outline the structure of the paper), perhaps emphasizing that these figures are not instrumental to the main points in the paper.

*We incorporated this clarification at the end of the introduction as recommended: 'Additional figures in Appendix A provide further context for and verification of our findings, but all core results are presented in the main body of the paper.'*

5. line 407: "A mechanism ..." - suggest to change to "One mechanism ..." to highlight that there could be others (as stated in your response to the referee)

*Good point, thank you. We changed the wording here.*